# Fetal-restricted hematopoietic progenitors arise from hemogenic endothelium in vitelline and umbilical arteries

Cristiana Barone [1,15], Giulia Quattrini[1,15], Alessandro Muratore[1], Giorgio Anselmi [2], Yurim Park[2], Naeema T. Mehmood[2], Elena Morganti [3], Roberto Orsenigo [1,11], Filipa Timóteo-Ferreira [1], Anna Cazzola[1,12], Arianna Patelli[1], Thea Milanesi[1], Bianca Nesti [3,4], Francisca Soares-da-Silva [5,6,7,13], Matthew Nicholls [2], Gloria Zambelli [1], Mario Mauri[1], Silvia Bombelli[1,8], Sofia De Marco[1], Deborah D'Aliberti[1], Silvia Spinelli[1], Veronica Bonalume[9], Alison Domingues[9,14], Mahdieh Naghavi Alhosseini[1], Gianluca Sala[1], Arianna Colonna[1], Elisabetta D'Errico [1], Cristina D'Orlando[1], Cristina Bianchi[1], Roberto A. Perego[1], Raffaella Meneveri[1], Ana Cumano[5,6,7], Silvia Brunelli [1,10], Marella F. T. R. De Bruijn[2], Andrea Ditadi[3], Alessandro Fantin [9], Rocco Piazza[1,10] & Emanuele Azzoni [1,10] ✉

Embryonic hematopoiesis involves successive waves of progenitors from distinct anatomical sites, but the origins and contributions of early hematopoietic stem and progenitor cells (HSPCs) remain incompletely defined. Here we use genetic fate mapping in mice to temporally label hemogenic endothelium (HE) subsets and track their progeny. We show that a wave of fetal-restricted HSPCs arises from HE in the vitelline and umbilical arteries between embryonic days 8.5 and 9.5, preceding the emergence of definitive hematopoietic stem cells. Lineage tracing, single-cell transcriptomic analyses and functional assays revealed that these progenitors are transient and distinct from erythro-myeloid progenitors, contribute extensively to fetal lympho-myelopoiesis but decline postnatally. Our findings reveal a previously unrecognized early HE wave as a key source of fetal-restricted HSPCs, refining the spatial–temporal understanding of layered hematopoiesis and informing developmental origins of blood cell diversity.

The vertebrate embryonic hematopoietic system develops through a series of overlapping waves of blood progenitors, each with progressively broader lineage potential[1,2]. Although the phenomenon of 'layered hematopoiesis'—the sequential emergence of distinct blood cell populations—is highly conserved across species, its detailed analysis in model organisms has been challenging owing to the temporal and spatial overlap of these waves and their multiple anatomical sources. Furthermore, the identification of embryonic blood progenitor cells has been complicated by the extensive sharing of surface markers among these populations, and only recently has their heterogeneity begun to be unraveled through advances in single-cell methodologies[3].

Adult repopulating hematopoietic stem cells (HSCs) are firstly and autonomously generated in the aorta–gonad–mesonephros (AGM) region starting from embryonic day (E)10.5 in the mouse[4]. HSCs originate from a specialized population of endothelial cells termed hemogenic endothelium (HE) in the major embryonic arteries[5–8] and mature through a hierarchy of pro- and pre-HSC intermediates[9,10]. Starting from E12, HSCs colonize the fetal liver (FL) where they are thought to

expand in numbers[11], and toward the end of gestation relocate to the bone marrow (BM) where they will reside throughout adult life. The BM niche, however, does not acquire robust HSC support capability until after birth[12].

Before HSC generation and following the early onset of primitive hematopoiesis, several waves of oligopotent progenitors begin emerging at E8.25 (refs. 2,13), initially from HE in the yolk sac (YS)[14]. Among HSC-independent progenitors, erythro-myeloid progenitors (EMPs) generate tissue-resident macrophages persisting until adult life[15,16]. EMPs were also reported to contribute to fetal erythropoiesis[17] and fetal innate lymphoid cells[18]. Although EMPs can generate multiple myeloid lineages[19], their physiological contribution to fetal and postnatal hemopoiesis is still unclear. Immune-restricted lympho-myeloid progenitors (LMPs) emerge in the YS slightly later than EMPs and were shown to take part in fetal lympho-myelopoiesis, even though for a limited time window and contributing less than 20% of myeloid cells at E14.5 (ref. 20). Despite conclusive evidence that YS-derived HSC-independent B and T cell progenitors exist and persist to adulthood[21–23], some controversy still remains regarding the identity of the first progenitors responsible for the colonization of the fetal thymus and, in particular, whether they originate from HSCs or not[24,25].

HSC-independent progenitors are necessary and sufficient to sustain fetal life until the end of gestation[26]. Indeed, recent work showed that HSCs exert a limited contribution to prenatal hematopoiesis[27,28]. Lineage tracing identified embryonic multipotent progenitors (eMPPs) appearing concomitantly to definitive HSCs, which markedly contribute to fetal and adult multi-lineage hematopoiesis[28–30]. Although the origin of eMPPs appears to be HSC independent[28,31], it is not clear when and where they emerge during development. The existence of fetal HSCs with characteristics distinct from those of adult HSCs has also been suggested[32,33]; however, as for eMPPs, a prospective identification of these progenitors, which would allow localization in their niche of emergence, is currently not possible[34]. Moreover, the true extent of their contribution to fetal and adult hematopoiesis needs further clarification.

To genetically label and trace discrete subsets of HE, we took advantage of well-established conditional fate-mapping strategies in mice. We found that a wave of fetal-restricted hematopoietic stem and progenitor cells (HSPCs) emerges from HE between E8.5 and E9.5, before the onset of adult-type HSCs, and acts as a major driver of fetal lympho-myelopoiesis. Through a combination of whole-mount imaging and single-cell transcriptomics, we localized the initial emergence of HSPCs belonging to this wave to the hematopoietic clusters of vitelline and umbilical arteries (VU). Moreover, we show that fetal-restricted HSPCs are not endowed with long-term multi-lineage engraftment potential but instead represent a heterogeneous subset of hematopoietic progenitors poised for differentiation, probably including eMPPs.

## Results

### HE lineage tracing identifies a population of fetal-restricted hematopoietic progenitors

All hematopoietic cells, with the possible exception of some primitive erythrocytes, originate from $Cdh5^+$ HE (ref. 1). To differentially label embryonic hematopoietic waves and systematically trace their contribution during fetal and adult hematopoiesis, we used a well validated pulse-chase approach using tamoxifen-inducible *Cdh5-CreER[T2]* mice[35], together with reporter lines selected for their suitability to different applications (*R26[tdTomato]/R26[zsGreen]* or *R26[EYFP]*) (Extended Data Fig. 1a). This strategy benefits from the use of a single Cre line, thereby avoiding bias from cell-type-specific promoters. To achieve precise temporal control, we used (Z)-4-hydroxytamoxifen (4-OHT), which has a short in vivo half-life (<3 h), reaches peak serum levels rapidly after administration and is cleared to undetectable within 12 h, as shown by mass spectrometry[36]. By contrast, tamoxifen requires hepatic metabolization into 4-hydroxytamoxifen (4-OHT), a process that takes approximately 6–12 h in vivo; tamoxifen itself remains detectable in serum for up to 48 h after administration[37].

We first evaluated the labeling of YS EMPs (Ter119⁻Kit⁺CD41^low CD16/32⁺)[19]. In *Cdh5-CreER[T2]::R26[zsGreen]* embryos, E9.5 and E10.5 YS EMPs were found labeled at high efficiency with 4-OHT activation at both E7.5 and E8.5 (Extended Data Fig. 1b,c). EMP labeling was also confirmed by whole-mount imaging of E9.5 YS (Extended Data Fig. 1d), which identified no difference in the number of labeled Kit⁺ clusters when traced at the two activation time points (Extended Data Fig. 1e). Consistent with this, brain microglia, which originates from YS EMPs[38], was found highly labeled with both 4-OHT at E7.5 and E8.5, in the E16.5 fetus and in the adult (Extended Data Fig. 1f,g). By contrast, as expected, 4-OHT at E10.5 did not label microglia (Extended Data Fig. 1g).

We analyzed the labeling of LMPs (CD31⁺Kit⁺CD45⁺) in the E10.5 AGM and YS including VU connecting the yolk sac to the embryo proper (YS + VU). While in the AGM LMP labeling was partial with both activation time points, 4-OHT at E8.5 traced the majority of LMPs in the YS + VU (Extended Data Fig. 2a,b), consistent with LMP emergence at or around E9.5 in the YS[20]. In the E11.5 FL, the LMP (Lin⁻Kit⁺CD45⁺Flt3⁺IL7Rα⁺) recombination frequency was low with 4-OHT at E7.5 and increased with activation at E8.5 (Extended Data Fig. 2c,d). Interestingly, the percentage of traced CD45⁺Kit⁺ hematopoietic progenitors in the E11.5 FL doubled when 4-OHT was delivered at E8.5 compared with E7.5 (Extended Data Fig. 2e), suggesting that part of them originate independently from EMPs.

Next, we investigated the extent of labeling of immunophenotypic type 1 (CD31⁺Kit⁺CD41^low CD45⁻CD43⁺CD201⁺) and type 2 pre-HSCs (CD31⁺Kit⁺CD41^low CD45⁺CD43⁺CD201⁺)[9,39] in AGM and YS + VU of *Cdh5-CreER[T2]::R26[tdTomato]/R26[zsGreen]* E11.5 embryos (Fig. 1a). 4-OHT at E7.5 labeled a minority of type 1 and type 2 pre-HSCs in both AGM (Fig. 1b,c) and YS + VU (Fig. 1d). Type 1 pre-HSC labeling was also low at E10.5 at this activation time (Extended Data Fig. 2a,b). By contrast,

**Fig. 1 | Lineage tracing of HE between E8.5 and E9.5 labels a subset of phenotypic hematopoietic progenitors and HSCs. a**, Visual schematic of lineage tracing experiments in *Cdh5-CreER[T2]::R26[tdTomato]/R26[zsGreen]* E11.5–E14.5–E16.5 embryos and postnatal mice, related to Figs. 1 and 2 and Extended Data Figs. 1f,g, 2c–m and 3. **b**, Representative flow cytometric analysis of AGM region hematopoietic progenitors (Ter119⁻Kit⁺CD31⁺CD41^low CD43⁺CD201⁻), type 1 pre-HSCs (Ter119⁻Kit⁺CD31⁺CD45⁻CD41^low CD43⁺CD201⁺) and type 2 pre-HSCs (Ter119⁻Kit⁺CD31⁺CD45⁺CD41^low CD43⁺CD201⁺) in E11.5 *Cdh5-CreER[T2]::R26[zsGreen]* embryos, quantified in **c**. **c,d**, Quantification of flow cytometric analysis shown in **b**. The percentage (%) of recombination is represented as the percentage of zsGreen⁺ cells within progenitors and type 1 and type 2 pre-HSCs in E11.5 *Cdh5-CreER[T2]::R26[zsGreen]* AGM (**c**) and YS + VU (**d**). 4-OHT at E7.5 (n = 15), 4-OHT at E8.5 (n = 23) (shown in **b**), 4-OHT at E10.5 (n = 12) AGM (**c**) and 4-OHT at E7.5 (n = 15), 4-OHT at E8.5 (n = 14), 4-OHT at E10.5 (n = 12). (**d**) were analyzed individually across six independent experiments. Error bars represent the mean ± s.d. P values

are as indicated in the figure (two-way ANOVA followed by Tukey's multiple comparisons test). **e**, Representative flow cytometric analysis of FL HSCs (Lin⁻Kit⁺Sca1⁺CD48⁻CD150⁺), LK (Lin⁻Kit⁺Sca1⁻) and LSK (Lin⁻Kit⁺Sca1⁺) and relative labeling frequencies in E14.5 *Cdh5-CreER[T2]::R26[tdTomato]* embryos. Lineage cocktail: B220, CD19, CD3e, F4/80, Gr1, Nk1.1, Ter119, 7-AAD. The related quantification is shown in **f**. **f–h**, Recombination within phenotypic HSCs in E14.5 FL (**f**), E16.5 FL (**g**) and adult BM (**h**). Data are shown as the percentage of tdTomato⁺ or zsGreen⁺ cells within HSCs. E14.5 FL: 4-OHT at E7.5 (n = 14), E8.5 (n = 26) or E10.5 (n = 16) (shown in **e**), 6 independent experiments. E16.5 FL: 4-OHT at E7.5 (n = 13), E8.5 (n = 18), E10.5 (n = 17), 4 independent experiments. Adult BM: 4-OHT at E7.5 (n = 5), E8.5 (n = 13), E10.5 (n = 13), 7 independent experiments. Error bars represent the mean ± s.d. P values are indicated in the figure (one-way ANOVA followed by Tukey's multiple comparisons test). Illustration in **a** created in BioRender; Brunelli, S. https://biorender.com/sb5tlt5 (2026).

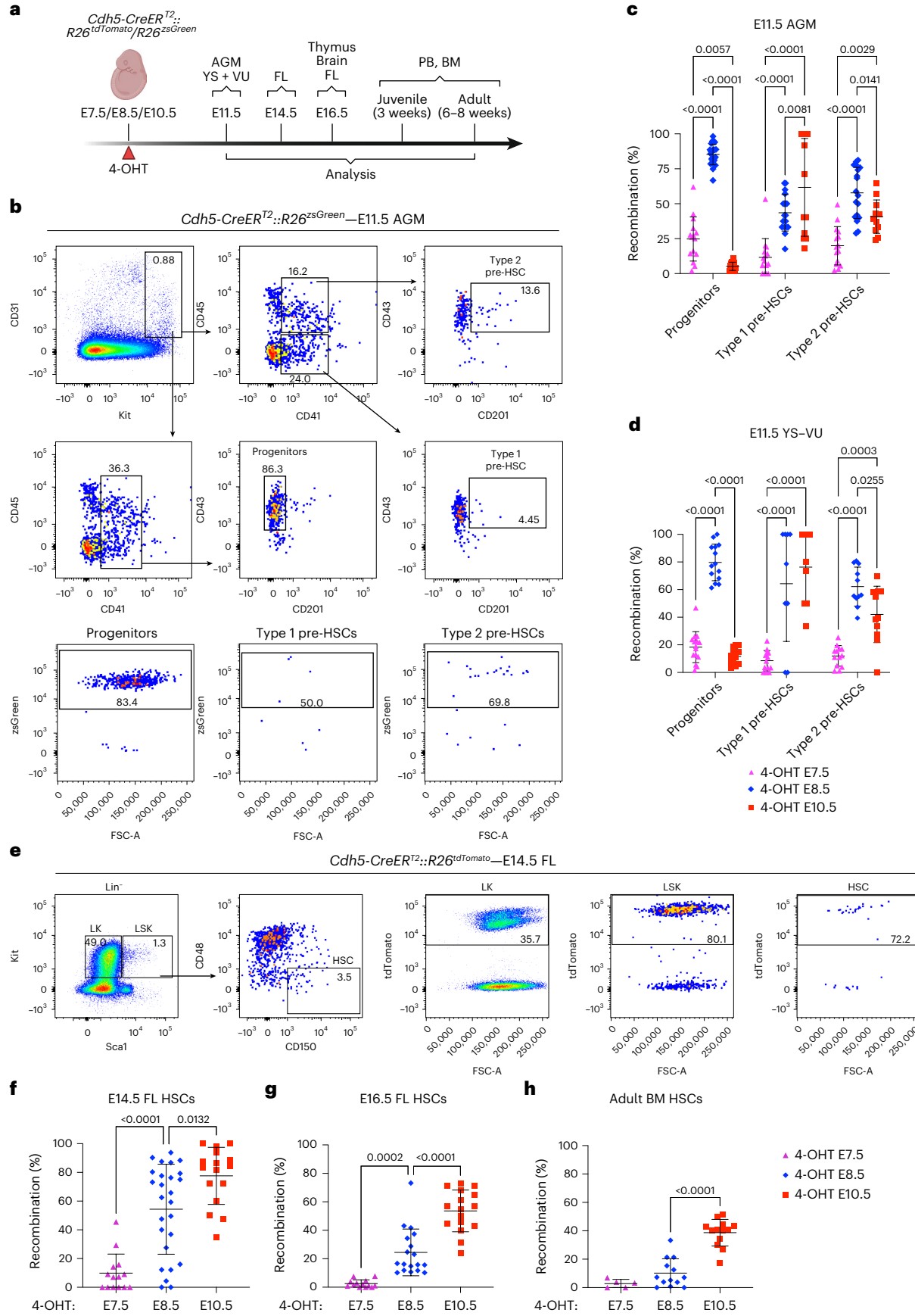

both E8.5 and E10.5 activations yielded substantial pre-HSC labeling at E11.5. While type 1 pre-HSCs were captured at a higher frequency with 4-OHT at E10.5 in the AGM, labeling of more mature type 2 pre-HSCs was significantly higher (~60%) in both AGM and YS + VU with 4-OHT at E8.5, as opposed to an apparently complementary subset (~40%) labeled with 4-OHT at E10.5 (Fig. 1b–d). Strikingly, hematopoietic progenitors other than pre-HSCs (CD31$^+$Kit$^+$CD41$^{low}$CD45$^{+/-}$CD43$^+$CD201$^-$) showed a much more selective labeling pattern and were almost exclusively labeled by the E8.5 activation in AGM and YS + VU (Fig. 1b–d). When YS and VU were dissected and analyzed separately from each other, progenitor subsets in each tissue showed no difference in labeling frequencies (Extended Data Fig. 2f); therefore, we kept these tissues together for the remainder of this study.

We then tested the labeling of fetal and adult phenotypic HSCs (Lin$^-$Kit$^+$Sca1$^+$CD150$^+$CD48$^-$) (Fig. 1e). E14.5, E16.5 FL and adult BM HSCs were extensively labeled with 4-OHT at E10.5, but not at E7.5 (Fig. 1f–h), confirming a previous report[40]. Interestingly, 4-OHT at E8.5 labeled E14.5 FL HSCs with variable efficiency (58% on average; Fig. 1f) not dependent on the specific reporter line (Extended Data Fig. 2g), but decreasing to an average of 25% at E16.5 (Fig. 1g) and 10% in the adult BM (Fig. 1h), raising the possibility of the existence of a wave of 'fetal-restricted' HSCs, as previously suggested[32]. A similar labeling pattern, however, was also observed in Lin$^-$Kit$^+$Sca1$^+$ (LSK) progenitors (Extended Data Fig. 2h–j). Labeling of non-HSC hematopoietic progenitors (LK, Lin$^-$Kit$^+$Sca1$^-$), comprising granulocyte–macrophage progenitors, common myeloid progenitors (CMP) and megakaryocyte–erythroid progenitors (MEP), was highest with 4-OHT at E8.5 in E14.5 FL (Extended Data Fig. 2k), whereas at E16.5 and in adult BM they were mostly labeled with 4-OHT at E10.5 (Extended Data Fig. 2l,m).

Taken together, these data suggest that HE lineage tracing in the *Cdh5-CreER$^{T2}$* model (4-OHT at E8.5) may capture a putative population of HSPCs largely restricted to fetal life. By contrast, 4-OHT pulses at E7.5 or E10.5, respectively, label either EMPs or adult-type HSCs. Therefore, the same genetic model can be used to study the relative fetal and adult contributions of three sequential waves of HE in an unbiased way.

## Fetal lympho-myelopoiesis is largely contributed from hematopoietic progenitors originating from E8.5–E9.5 HE

To examine the fetal lympho-myeloid contribution of the three HE waves, we analyzed those lineages in *Cdh5-CreER$^{T2}$::R26$^{tdTomato}$* E16.5 FL and thymus. 4-OHT activation at E7.5 yielded an average labeling of only 35% of F4/80$^+$CD11b$^{low}$ macrophages, less than 10% of B cells, 10–15% of T cells and 5% of F4/80$^{low/-}$CD11b$^+$ myeloid cells in the E16.5 FL and thymus (Fig. 2a,b). E10.5 activation resulted in labeling of 30% of B and myeloid cells (Fig. 2a), a similar contribution to T cells, with higher labeling (30–40%) in less differentiated thymocytes (DN1 and DN2) (Fig. 2b) and negligible labeling in macrophages. Strikingly, the highest labeling frequencies in all observed lineages were detected with 4-OHT activation at E8.5, yielding on average 70% of labeled B and myeloid cells in the E16.5 FL (Fig. 2a and Extended Data Fig. 3a) and 50–60% of labeled

E16.5 fetal thymocytes (Fig. 2b and Extended Data Fig. 3b). In contrast to E10.5 activation, double-positive (CD4$^+$CD8$^+$DP) T cells showed the highest labeling frequency, and embryonic γδ T cells were preferentially labeled by 4-OHT administration at E8.5, showing an approximately fourfold higher labeling efficiency compared with E10.5 (Fig. 2b). Thus, our data are consistent with the previously reported model suggesting the existence of two separate waves of thymus-settling progenitors[41].

Postnatal analysis confirmed the absence of peripheral blood (PB) labeling (<5%) in all lineages with activation at E7.5, both at 21 days (Extended Data Fig. 3c) and 2 months (Fig. 2c,d), while E10.5 activation resulted in high levels of recombination in all lineages (Fig. 2c,d and Extended Data Fig. 3c). Conversely, activation at E8.5 showed highly variable labeling at 21 days (average 30–35%) (Extended Data Fig. 3c), which showed a decreasing trend at 2 months (20–25%; Fig. 2c,d and Extended Data Fig. 3d). Notably, comparison of fetal and postnatal stages revealed divergent dynamics for the two labeling windows. Labeling obtained with 4-OHT activation at E8.5, which was highest at E16.5, declined markedly after birth, indicating that progeny of this HE wave progressively lose representation in postnatal hematopoiesis (Fig. 2d). By contrast, labeling following 4-OHT activation at E10.5 increased from E16.5 to postnatal stages (Fig. 2d), consistent with the gradual takeover of hematopoiesis by adult-type definitive HSCs. Together, these reciprocal trends indicate a developmental handover between a predominantly fetal hematopoietic program emerging from E8.5 to E9.5 HE and a later-arising program specified from E10.5 HE that sustains postnatal blood production.

These data show that the hematopoietic wave that emerges from HE between E8.5 and E9.5 is a major contributor to fetal, but not adult, lympho-myelopoiesis. As HE labeling at E8.5 marks phenotypic progenitors and pre-HSCs (Fig. 1) and similar labeling dynamics were seen for LK, LSK and HSC (Fig. 2a–d and Extended Data Fig. 2h–m), this wave of HE probably contains the precursors that generate a pre-constituted hierarchy of fetal-restricted hematopoietic progenitors, including fetal HSCs, HSC-independent eMPPs and other progenitors[28–30,34,42]. Therefore, we will collectively refer to these as 'fetal-restricted HSPCs'.

## Fetal-restricted HSPCs first emerge from HE of the vitelline and umbilical arteries

To obtain insight into the dynamics of fetal-restricted HSPC generation, we investigated sites of hematopoietic emergence using whole-mount confocal imaging. As mentioned, Kit$^+$CD31$^+$ hematopoietic clusters in the E9.5 YS, corresponding to EMPs, were equally labeled by 4-OHT activation at E7.5 or E8.5 (Extended Data Fig. 1d,e). CD31$^+$Kit$^+$ hematopoietic clusters in the dorsal aorta (DA), thought to contain pre-HSCs[43], peak at E10.5 (ref. 44). However, the first intra-embryonic Kit$^+$ hematopoietic clusters appear within the portion of the vitelline artery (VA) most proximal to the DA[9,45,46]. Although the majority of these clusters are thought to contain progenitor cells other than pro-HSC[9], vitelline and umbilical arteries are known to represent sites of pre-HSC emergence[7,47]. Importantly, because of the lack of specific ways to trace

---

**Fig. 2 | Extensive but transient lympho-myeloid contribution of fetal-restricted HSPCs at the end of gestation. a**, Quantification of labeled (tdTomato$^+$) B cells (CD45$^+$B220$^+$), myeloid cells (CD45$^+$CD11b$^+$), macrophages (CD45$^+$F4/80$^{hi}$) and total leukocytes (CD45$^+$) in E16.5 FL from *Cdh5-CreER$^{T2}$::R26$^{tdTomato}$* embryos activated at E7.5 (left, $n = 21$), E8.5 (middle, $n = 23$) or E10.5 (right, $n = 13$), analyzed across 5 independent experiments (gating strategy in Extended Data Fig. 3a). Error bars represent the mean ± s.d. **b**, Quantification of labeled thymocyte subsets in *Cdh5-CreER$^{T2}$::R26$^{tdTomato}$* E16.5 fetal thymus (gating strategy in Extended Data Fig. 3b), activated with 4-OHT at E7.5 (left, $n = 13$), E8.5 (middle, $n = 18$) or E10.5 (right, $n = 16$). Thymuses were analyzed individually across four independent experiments. Error bars represent the mean ± s.d. **c**, Quantification of flow cytometric analysis of labeled B cells (CD45$^+$B220$^+$), myeloid cells (CD45$^+$CD11b$^+$), T cells (CD45$^+$CD3e$^+$) and total leukocytes (CD45$^+$) in adult (2 months old) PB from *Cdh5-CreER$^{T2}$::R26$^{tdTomato}$* mice (gating strategy in

Extended Data Fig. 3d), activated with 4-OHT at E7.5 (left, $n = 12$), E8.5 (middle, $n = 13$) or E10.5 (right, $n = 13$). Mice were analyzed individually across seven independent experiments. **d**, Longitudinal labeling of B and T lymphocytes and myeloid cells at the three analysis time points (E16.5, 21 days, and 2 months or adult). E16.5 FL (4-OHT at E7.5, $n = 21$; 4-OHT at E8.5, $n = 23$; 4-OHT at E10.5, $n = 8$), 21-day-old mice PB (4-OHT at E7.5, $n = 12$; 4-OHT at E8.5, $n = 15$; 4-OHT at E10.5, $n = 13$) and adult 2-month-old mice PB (4-OHT at E7.5, $n = 12$; 4-OHT at E8.5, $n = 13$; 4-OHT at E10.5, $n = 13$) were analyzed individually across 12 independent experiments. $P$ values indicated in the figure show significant differences between labeling at E16.5 and 21 days for the E8.5 and E10.5 activation time points. The T cell graph was made based on recombination frequencies within the CD4$^+$CD8$^+$ subset. Error bars are colored for each activation time point and represent the mean ± s.d. $P$ values are indicated in the figure (two-way ANOVA followed by Tukey's multiple comparisons test).

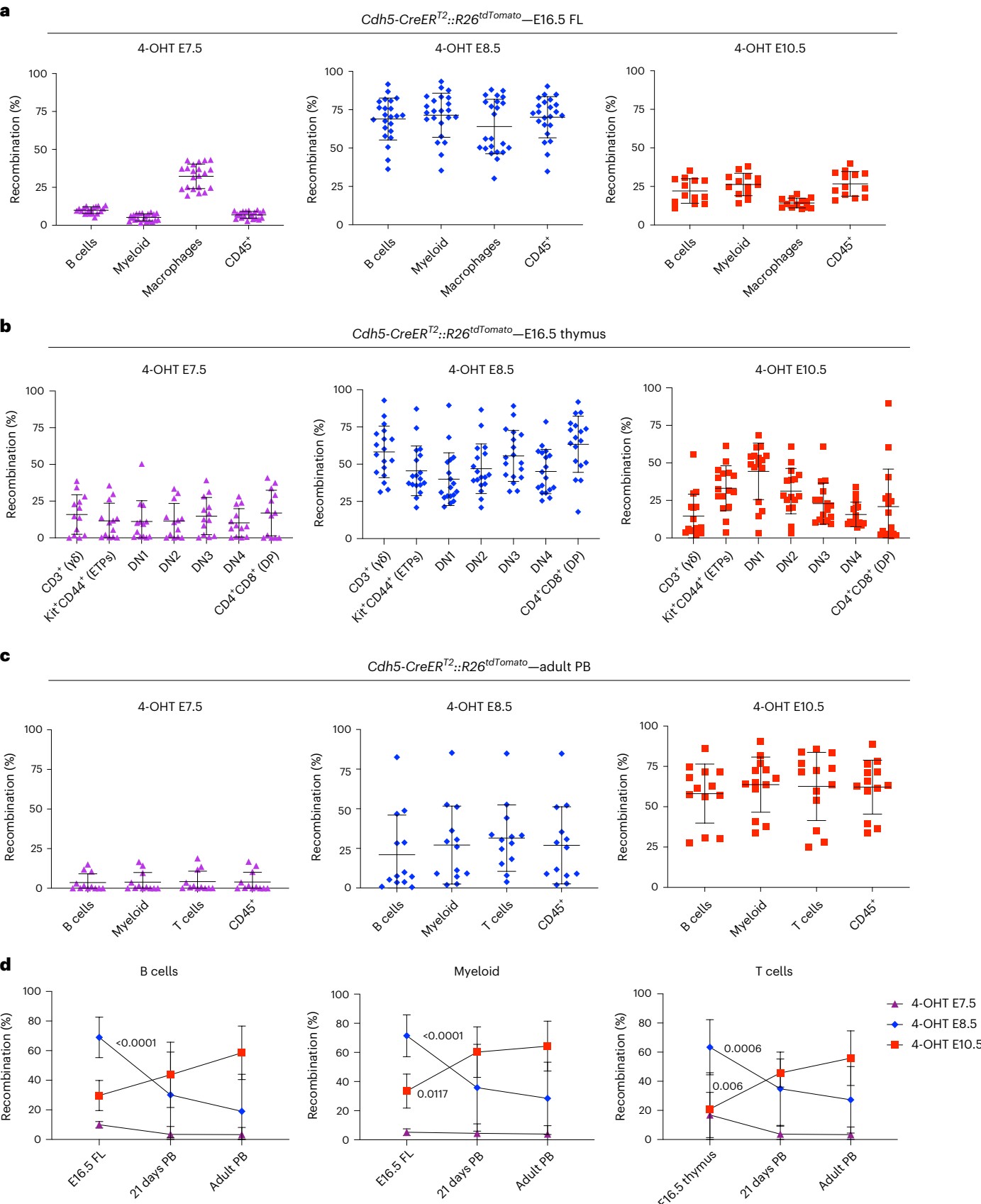

these early progenitors, their contribution has never been analyzed in an unperturbed system. Remarkably, few Kit[+] hematopoietic clusters in the *Cdh5-CreER^T2::R26^EYFP* E9.5 VA were labeled by 4-OHT at E7.5, but, in contrast, their majority was labeled by the E8.5 activation (Fig. 3a,b), correlating with the emergence of fetal-restricted HSPCs. These observations indicate that such progenitors first emerge within the VA. Flow cytometry analysis of E9.5 hematopoietic progenitors other than EMP (non-EMPs, Kit[+]CD41[low]CD16/32[−]) showed absence of differential labeling in the YS, but significantly higher labeling within the caudal part of the embryo (CP) when traced at E8.5 (Extended Data Fig. 4a), consistent with imaging data and an identity of VA clusters independent from EMPs. Next, we evaluated labeling of hematopoietic clusters in the VA and umbilical artery (UA) of E10.5 embryos (CD31[+]Kit[+] or CD31[+]Runx1[+] round-shaped cells) and compared it with that of the DA. 4-OHT activation at E7.5 yielded low labeling frequency of both VU and aortic clusters (Fig. 3c–f). Conversely, 4-OHT at E8.5 resulted in high levels of recombination in VU clusters, but significantly lower in DA clusters (Fig. 3c–f). In line with these results, differential labeling of non-EMPs by flow cytometry was also observed at E10.5 in the AGM (Extended Data Fig. 4b). Interestingly, HE labeling was low in the E10.5–E11.5 AGM at all activation time points, as assessed by whole-mount imaging (CD31[+]Runx1[+] flat-shaped cells; Fig. 3e,f) and flow cytometry (CD31[+]Kit[+]CD41[−]CD45[−]CD43[−]) (Fig. 3g and Extended Data Fig. 4c). Notably, neither E7.5 nor E8.5 activation resulted in appreciable recombination within the ventral aortic endothelium (Fig. 3e), consistent with the lack of tracing of adult BM HSCs (Fig. 2d) arising in this location[48]. AGM non-HE (CD31[+]Kit[−]CD41[−]CD45[−]CD43[−]) was not labeled (4-OHT at E7.5) or labeled at low frequency (<10%; 4-OHT at E7.5 or E8.5) (Fig. 3g).

Previous ex vivo culture assays revealed the presence of pre-HSCs in the YS at E10.5, but not at E9.5 (refs. 9,45,49); from E11.5, YS pre-HSCs seem to decline[10,11,49]. Whole-mount analysis of E10.5 YS identified the presence of large Kit[+] hematopoietic clusters, which were absent at E9.5. Within the YS, these clusters exclusively localized in the VA and its ramifications (which form a continuous connection with the intra-embryonic portion of the VA), and were previously documented to express *Ly6a* (ref. 46) and *Hlf* (ref. 50), markers associated with HSC activity. In *Cdh5-CreER^T2::R26^zsGreen* YS, the majority of cells in these clusters were not labeled with activation at E7.5 but, instead, were consistently labeled with 4-OHT at E8.5 (Extended Data Fig. 4d,e). In line with this and with EMP labeling in our system, while E7.5 activation identified hematopoietic clusters only in the YS vascular plexus, E8.5 activation labeled clusters in both plexus and YS arteries (Extended Data Fig. 4f,g). Similar to what was already observed in the AGM, the levels of labeling of E10.5 and E11.5 YS HE were undetectable to low at all activation time points (Extended Data Fig. 4g,h).

Overall, these results suggest that fetal-restricted HSPCs first emerge from the HE of the VA and the UA. Whereas at E9.5 the location of the hematopoietic clusters likely to contain these progenitors is solely intra-embryonic, from E10.5, these clusters are also found within the major arteries of the YS. The lack of labeled HE already 24 h after activation suggests that endothelial-to-hematopoietic transition (EHT) events take place in vivo within a short time window of <12 h, similar to what was observed ex vivo[6], and that the entirety of labeled HE undergoes EHT, consistently with it being hematopoietic committed and devoid of contribution to the structural endothelium[51].

## EMPs do not significantly contribute to hematopoietic clusters in intra- and extra-embryonic arteries

To gain insight on the origin of hematopoietic clusters in different sites of emergence, we performed lineage tracing using the *Csf1r-iCre* transgenic mouse line, which targets EMPs and their progeny[15,16].

Kit[+] clusters in the E9.5 VA of *Csf1r-iCre::R26^tdTomato* embryos showed near complete absence of labeling (Fig. 4a,d), despite the expected highly efficient labeling of E9.5 EMPs (Fig. 4b). Non-EMPs were not labeled in the E9.5 CP, whereas ~50% recombined in the E9.5 YS, possibly representing immature EMPs and/or *Csf1r*[+] progenitors other than EMPs (Fig. 4b). Interestingly, LMPs were previously shown to emerge in the YS vascular plexus starting from E9.5 and to express *Csf1r* (ref. 20,25). Hematopoietic clusters in the E10.5 DA, VA and UA and large arterial clusters in the E10.5 YS showed very low labeling, confirming that they do not originate from EMPs (Fig. 4c,d and Extended Data Fig. 5a). By contrast, extensive labeling was detected in the E10.5 FL, in agreement with its early colonization by EMPs (Fig. 4c,d). Accordingly, at E10.5, EMP labeling remained high (>90%), while less than 25% non-EMPs were labeled in both YS + VU and embryo proper (Fig. 4e).

These data show that hematopoietic clusters emerging in the major intra- and extra-embryonic arteries, including the E10.5 YS, largely contain cells that develop independently of EMPs.

## A late wave of *Csf1r*[+] progenitors exerts a limited contribution to fetal lympho-myelopoiesis

As 4-OHT activations at E7.5–E8.5 in the *Cdh5-CreER^T2* system both target EMPs and non-EMPs, we wanted to get a better assessment of the contribution of EMPs to fetal lympho-myelopoiesis. To this aim, we took advantage of the tamoxifen-inducible *Csf1r*^MerCreMer line, previously used to trace EMPs in the absence of HSC labeling[15–17,52]. As expected, in *Csf1r*^MerCreMer::R26^tdTomato E9.5–E10.5 embryos, both 4-OHT at E8.5 and E9.5 label EMPs but few non-EMPs (Extended Data Fig. 5b–d), as also confirmed by microglia labeling in the E16.5 brain (Fig. 4f).

**Fig. 3 | Whole-mount confocal imaging localizes the initial emergence of fetal-restricted HSPCs to the vitelline and umbilical arteries. a**, WM-IF confocal analysis of E9.5 (21–27 sp) *Cdh5-CreER^T2::R26^EYFP* embryos in the CP. Maximum-intensity 3D projections are shown; the middle and right panels are magnifications. Arrowheads indicate Kit[+] hematopoietic clusters in the VA, labeled following 4-OHT administration at E8.5 (filled arrowheads) but not at E7.5 (open arrowheads). **b**, Quantification of EYFP labeling within Kit[+] cluster cells in the VA of E9.5 embryos shown in **a**. Measurements were obtained from embryos activated at E7.5 (*n* = 3) or E8.5 (*n* = 5), using 1–4 images per embryo (11 images for E7.5; 6 images for E8.5). Data are presented as mean ± s.d. The *P* value is indicated in the figure (two-tailed unpaired Student's *t*-test). **c**, WM-IF confocal analysis of the AGM region of E10.5 (32–36 sp) *Cdh5-CreER^T2::R26^EYFP* embryos. The left and middle panels show maximum-intensity 3D projections; the boxed region is shown as a single 2.5-µm optical slice (right). Arrowheads indicate Kit[+] hematopoietic clusters in the UA, unlabeled after E7.5 activation (open arrowheads; top), and labeled after E8.5 activation (filled arrowheads; bottom). Embryos activated at E7.5 (*n* = 9) or E8.5 (*n* = 12) were analyzed in 7 independent experiments. **d**, Quantification of labeling in Kit[+] hematopoietic clusters within the AGM at E10.5, shown in **c**. Clusters in the DA and VU were quantified separately. Data were derived from 9 embryos (E7.5 activation) and 12 embryos (E8.5 activation), with 3–6 images per embryo (39 images for E7.5; 56 images for E8.5). Data are presented as mean ± s.d. *P* values are indicated in the figure (two-way ANOVA with Tukey's multiple comparisons test). **e**, WM-IF confocal analysis of the E10.5 AGM region of E10.5 (32–36 sp) *Cdh5-CreER^T2::R26^EYFP* embryos. Boxed areas in maximum-intensity projections are magnified in single 2.5-µm optical slices. Runx1[+]CD31[+] round-shaped hematopoietic cluster cells (arrowheads) and flat-shaped HE (asterisks) are indicated. Embryos activated at E7.5 (*n* = 4) or E8.5 (*n* = 4) were analyzed. **f**, Quantification of labeling in Runx1[+]CD31[+] cluster cells and HE from **e**, analyzed separately in DA and VU regions. Data were derived from 4 (E7.5 activation) and 4 embryos (E8.5 activation), with 5–12 images per embryo (33 images for E7.5; 40 images for E8.5). Data are presented as mean ± s.d. *P* values are indicated in the figure (two-way ANOVA with Tukey's test). **g**, Flow cytometric quantification of labeled HE (Ter119[−]CD31[+]Kit[+]CD45[−]CD41[−]CD43[−]) and non-HE (Ter119[−]CD31[+]Kit[−]CD45[−]CD41[−]CD43[−]) in AGM of E11.5 *Cdh5-CreER^T2::R26^zsGreen* embryos (gates in Extended Data Fig. 4c). Embryos activated at E7.5 (*n* = 15), E8.5 (*n* = 11) or E10.5 (*n* = 12) were analyzed across 5 independent experiments. Data are presented as mean ± s.d. *P* values are indicated in the figure (two-way ANOVA with Tukey's test).

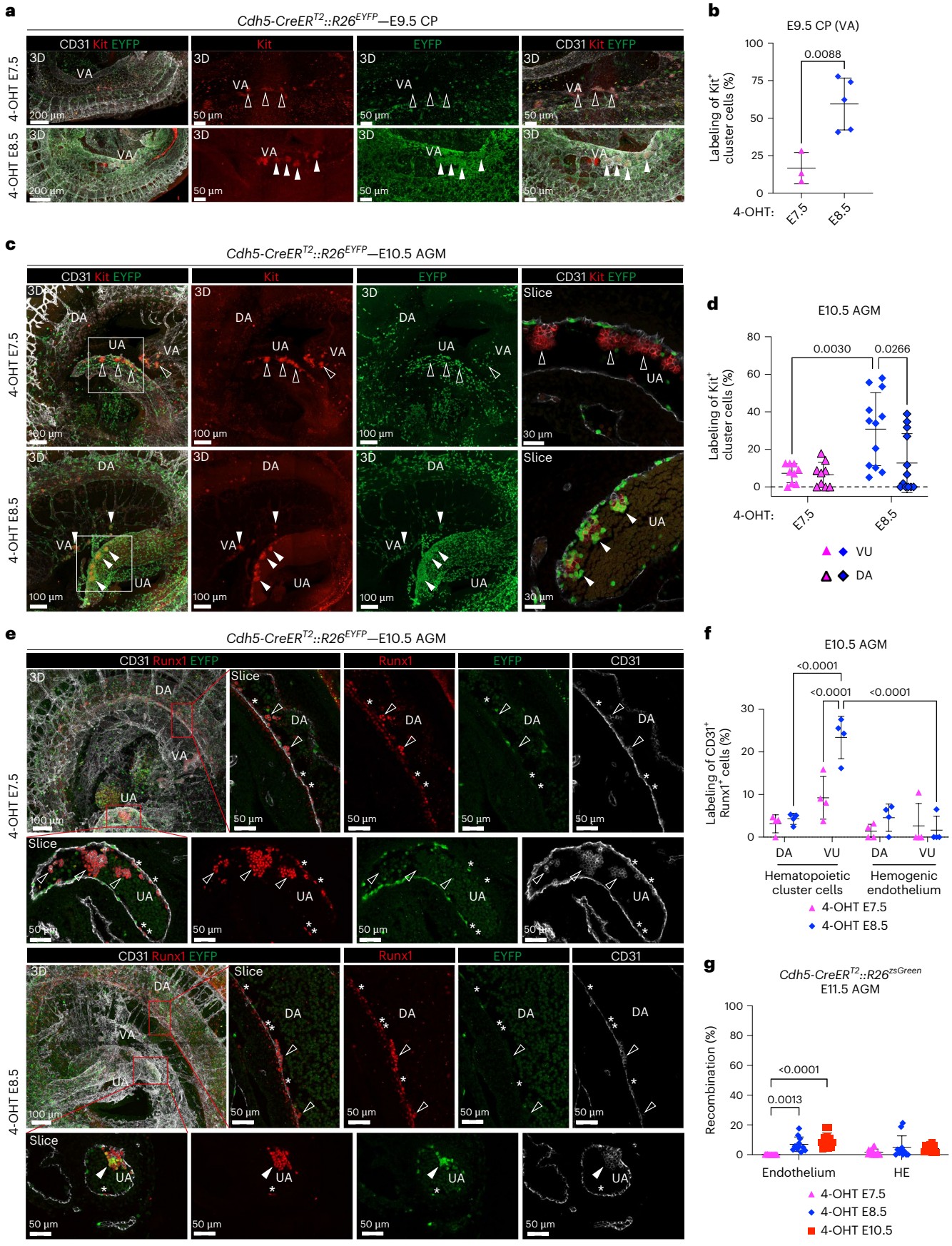

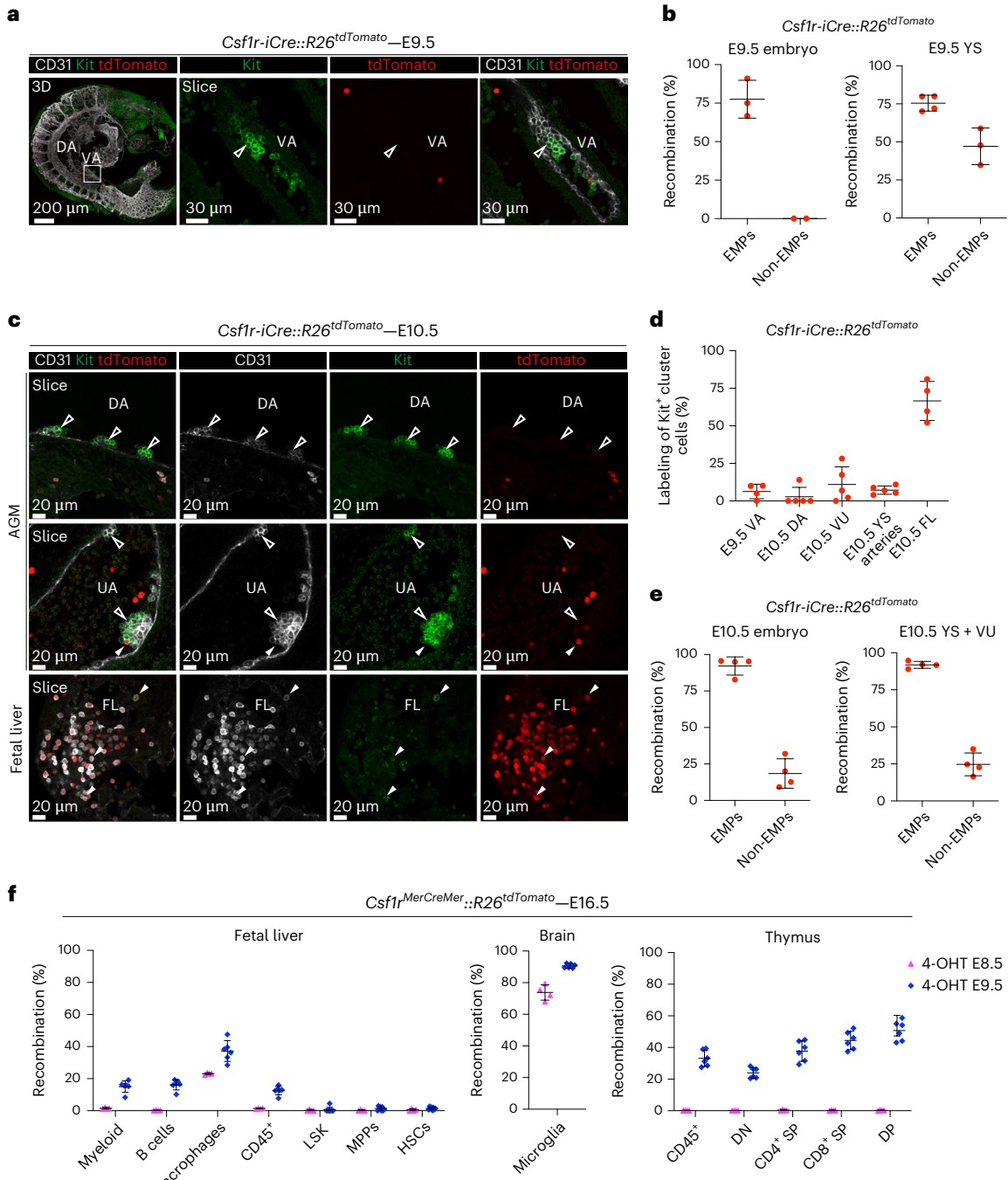

**Fig. 4 | *Csf1r* lineage tracing reveals a non-EMP identity of arterial hematopoietic clusters and the contribution of *Csf1r*⁺ progenitors to fetal lympho-myelopoiesis. a**, WM-IF confocal analysis of E9.5 *Csf1r-iCre::R26^tdTomato* embryos. A 3D magnification (left) or single 2.5-µm-thick optical slices (middle, right) are shown. Open arrowheads indicate unlabeled (tdTomato⁻) Kit⁺ clusters in the VA. Four embryos were analyzed in two independent experiments.
**b**, Flow cytometric quantification of labeled (tdTomato⁺) EMPs (Ter119⁻Kit⁺CD41^lo CD16/32⁺) and non-EMPs (Ter119⁻Kit⁺CD41^loCD16/32⁻) in E9.5 *Csf1r-iCre::R26^tdTomato* YS and embryos (gates in Extended Data Fig. 1b). Four YS and four embryos were analyzed. Data are presented as mean ± s.d. **c**, WM-IF confocal analysis of the E10.5 *Csf1r-iCre::R26^tdTomato* AGM region (top rows) and FL (bottom). Single 2.5-µm optical sections are shown. Arrowheads indicate Kit⁺ hematopoietic clusters in the DA, UA and hematopoietic cells in the FL. Open arrowheads mark unlabeled cells; filled arrowheads indicate tdTomato⁺ cells. Five embryos were analyzed across three independent experiments. **d**, Quantification of labeling from WM-IF

images shown in **a** and **c**. Measurements were obtained from E9.5 (*n* = 4) and E10.5 (*n* = 5) embryos, using 2–8 images per embryo (21 E9.5 images; 40 E10.5 images total). Data are presented as mean ± s.d. **e**, Flow cytometric quantification of labeled (tdTomato⁺) EMPs and non-EMPs in E10.5 *Csf1r-iCre::R26^tdTomato* YS + VU and embryos. Four YS and four embryos were analyzed. Data are presented as mean ± s.d. **f**, Flow cytometric quantification of E16.5 *Csf1r^MerCreMer::R26^tdTomato* FL, brain and thymus following 4-OHT administration at E8.5 (*n* = 4) or E9.5 (*n* = 6). Shown is the percentage of tdTomato⁺ cells within FL myeloid cells (Ter119⁻CD45⁺CD11b⁺F4/80⁻), B cells (Ter119⁻CD45⁺B220⁺), macrophages (Ter119⁻CD45⁺CD11b⁻F4/80⁺), CD45⁺ cells (Ter119⁻CD45⁺), LSK (Lin⁻Kit⁺Sca1⁺), MPPs (Lin⁻Kit⁺Sca1⁺CD48⁺CD150⁻) and HSCs (Lin⁻Kit⁺Sca1⁺CD48⁻CD150⁺); brain microglia (Kit⁺CD45⁺CD11b⁺F4/80⁺); and thymus CD45⁺, DN (CD45⁺CD4⁻CD8⁻), CD4⁺SP (CD45⁺CD4⁺CD8⁻), CD8⁺SP (CD45⁺CD4⁻CD8⁺) and DP (CD45⁺CD4⁺CD8⁺). An experimental schematic is shown in Extended Data Fig. 5b. Data are presented as mean ± s.d. from two independent experiments.

In the E16.5 FL and thymus, E8.5 activation yielded no detectable labeling, except FL macrophages (23%) (Fig. 4f). By contrast, E9.5 activation labeled a subset of myeloid and B cells (<20%) and 30–40% T cells in the E16.5 FL and thymus, with the highest labeling detected in more differentiated thymocytes (CD4$^+$CD8$^+$DP, 50% labeling) (Fig. 4f). LSK, including HSCs and multipotent hematopoietic progenitors (MPPs; CD48$^+$CD150$^-$) were not labeled with either activation (Fig. 4f).

These data show that $Csf1r^+$ progenitors contribute in vivo to fetal lymphoid cells, confirming a previous report[52]. Lympho-myeloid contribution of $Csf1r^+$ progenitors is limited in the FL but, surprisingly, more pronounced in the thymus. Notably, in $Csf1r^+$ progenitors, lymphoid potential appears only at E9.5. Our data provide support for the existence of at least two waves of $Csf1r^+$ progenitors emerging outside of the main arteries, with only the second being endowed with lympho-myeloid potential.

## B and T lymphoid potential appears in intra- and extra-embryonic HE between E8.5 and E9.5

To determine whether the ex vivo potential of fetal-restricted HSPCs mirrored their in vivo fate, we isolated traced and untraced cells from $Cdh5\text{-}CreER^{T2}::R26^{tdTomato/zsGreen}/R26^{zsGreen}$ E9.5 YS or CP and E10.5 YS + VU or AGM (Fig. 5a,b). We performed colony-forming unit-culture (CFU-C) assays able to detect single or combined erythroid, myeloid and megakaryocyte potential of mature progenitors. CFU-Cs largely segregated with labeled cells at both activations in either YS or CP–AGM (Fig. 5c), in agreement with the absence of differential labeling of EMPs at these two time points (Extended Data Fig. 1b–e).

We next tested B and T lymphoid potential of HE using bulk OP9 and OP9-DL1 co-culture assays, respectively. Here we plated E10.5 AGM or YS + VU cells without previous isolation of the traced or untraced fractions and normalized the percentage of labeled B and T lymphocytes after culture to initial Kit$^+$ cell labeling. We found that cells traced at E8.5 generated mature B and T lymphocytes with significantly higher frequency than those labeled at E7.5 in both AGM and YS + VU (Extended Data Fig. 6a–d). To quantify the frequency of B and T cell progenitors within labeled and unlabeled hemato-endothelial cells (CD31$^+$ and/or Kit$^+$; 4-OHT at E8.5) (Extended Data Fig. 6e,f), we performed a limiting dilution assay (LDA) in OP9/OP9-DL1 co-cultures. While B and T cell progenitors were almost exclusively detected in the labeled fractions (Fig. 5d,e and Extended Data Fig. 6g,h), T progenitors were significantly more frequent than B progenitors in both AGM (1:81 versus 1:204, $P = 0.016$) and YS + VU (1:123 versus 1:314, $P = 0.012$) (Fig. 5d,e).

These results show that while EMP emergence takes place from E7.5 onwards, progenitors showing B and T lymphoid potential first emerge from intra- and extra-embryonic HE between E8.5 and E9.5. The different frequencies of B and T lymphocyte progenitors observed in AGM and YS + VU are consistent with a model in which developmental uncoupling may already take place in the tissues of emergence of these precursors.

## Single-cell transcriptomics coupled with HE lineage tracing identify distinct subsets of pre-HSPCs in AGM and YS + VU

To determine the transcriptional identity of fetal-restricted HSPCs and evaluate the relationship between their origin and molecular signature, we performed single-cell RNA sequencing (scRNA-seq). We pooled hemato-endothelial CD31$^+$ and/or Kit$^+$ cells from $Cdh5\text{-}CreER^{T2}::R26^{tdTomato}$ AGM or YS + VU (4-OHT at E8.5) and separated them into labeled (tdT$^+$) and unlabeled (tdT$^-$) cells (Extended Data Fig. 6a,b). A total of 44 clusters were identified (Extended Data Fig. 7a). Pre-HSPCs distributed in two distinct clusters and were recognized by their transcriptional signature, characterized by the combined expression of the hematopoietic genes $Myb$, $Kit$, $Runx1$, $Adgrg1$ and $Flt3$ together with genes reported to identify HSCs or their immediate precursors, including $Hlf$ (refs. 28,50,53), $Cd27$ (refs. 54,55), $Mecom$ (ref. 28), $Cd93$ (ref. 56), $Hoxa7$ and $Hoxa9$ (ref. 53) among others (Fig. 6a,b). Pre-HSPC 1 were largely contributed by labeled AGM and unlabeled YS + VU cells; by contrast, tdT$^+$ YS + VU cells made up >75% of pre-HSPC 2 (Fig. 6c,d). Both clusters expressed $Cd27$ (Fig. 6b and Extended Data Fig. 7b). Differential gene expression analysis between the two clusters highlighted genes involved in hematopoietic differentiation as well as metabolic and inflammatory genes (Extended Data Fig. 7c,d and Supplementary Table 2).

Mapping to existing datasets[29,55] confirmed a stem and progenitor cell identity of the pre-HSPC 1 cluster, while a lymphoid progenitor signature was more enriched in pre-HSPC 2 (Fig. 6e,f). Separate evaluation of the signature expression levels in each subset revealed that uncommitted progenitor signatures were particularly expressed in AGM cells of both clusters, while the lymphoid progenitor signature was expressed at the highest level in tdT$^+$ cells of the YS + VU, found almost exclusively in the pre-HSPC 2 cluster (Extended Data Fig. 7e,f).

To further dissect pre-HSPC heterogeneity at the transcriptional level, we restricted our analysis to pre-HSPC clusters (Fig. 6g). Relative gene set comparison evidenced that the pre-HSC transcriptional signature was expressed at the highest level by labeled cells of the AGM (Fig. 6h) and that our whole-mount imaging preferentially localizes within the intra-embryonic portion of the VU (Fig. 3). By contrast, expression of genes characteristic of hematopoietic differentiation was higher in tdT$^+$ cells of the YS + VU (Fig. 6i). AGM and YS + VU pre-HSPCs showed a distinct metabolic signature, with glycolysis prominent in AGM and OXPHOS genes expressed at a higher level in YS + VU (Extended Data Fig. 8a). Pro-inflammatory genes, previously suggested to have a role in AGM HSC development[57–60], showed heterogeneous expression. Interestingly, interferon response genes were highest in labeled cells of AGM, while $Tnf$, $Nfkb$, $Tlr4$ and innate immune response genes were more expressed in labeled cells of the YS + VU (Extended Data Fig. 8b). Heterogeneity between tdT$^+$ AGM and YS + VU pre-HSPC was confirmed by differential gene expression analysis, which highlighted higher expression of ribosomal genes in the latter, indicative of higher metabolic activity (Extended Data Fig. 8c,d and Supplementary Table 3)[61]. Genes involved in signaling pathways implied in HSC specification such as $Notch$, $Shh$ and $TGF\text{-}\beta$ (ref. 62) were

**Fig. 5 | Ex vivo B and T lymphoid potential emerges in intra- and extra-embryonic HE at E8.5, while erythro-myeloid potential is already present at E7.5. a**, Schematic overview of CFU-C, OP9/OP9-Dl1 co-cultures and limiting dilution assays used to assess ex vivo hematopoietic potential of traced and untraced hemato-endothelial cells. **b**, Representative flow cytometric gating strategy for isolation of Ter119$^-$-labeled (tdTomato$^+$ or zsGreen$^+$) and unlabeled (tdTomato$^-$ or zsGreen$^-$) cell fractions from E10.5 $Cdh5\text{-}CreER^{T2}$ YS + VU, following 4-OHT administration at E7.5 or E8.5 (shown). **c**, CFU-C output from labeled and unlabeled Ter119$^-$ cell fractions isolated from $Cdh5\text{-}CreER^{T2}::R26^{tdTomato}/R26^{zsGreen}$ E9.5 CP and YS (left), or E10.5 AGM and YS + VU (right), following 4-OHT activation at E7.5 or E8.5. Colony numbers are normalized to 1,000 Kit$^+$ cells plated. E9.5: 4-OHT E7.5 ($n = 12$), 4-OHT E8.5 ($n = 6$); E10.5: 4-OHT E7.5 ($n = 4$), 4-OHT E8.5 ($n = 4$) samples analyzed across six independent experiments. Starting cell numbers ranged from 100 to 4,300 (YS) and 40 to 150,000 (CP/AGM).

GEMM, granulocyte–erythroid–monocyte/macrophage–megakaryocyte; G/M/ GM, granulocyte–monocyte/macrophage; Ery, erythroid. Data are presented as mean ± s.d. E9.5 CP 4-OHT E7.5: $P = 0.1204$; E9.5 CP 4-OHT E8.5: $P = 0.0988$; significant $P$ values are indicated in the figure (two-tailed unpaired Student's $t$-test). **d**,**e**, B cell (**d**) or T cell (**e**) LDAs performed with $Cdh5\text{-}CreER^{T2}::R26^{zsGreen}$ E10.5 AGM and YS + VU cells isolated as shown in Extended Data Fig. 6e,f. Representative flow cytometric analyses are shown in Extended Data Fig. 6g,h. Top tables show, in the left columns, the number of cells seeded per well. Progenitor frequencies were calculated based on the presence of B or T lymphocytes in OP9 or OP9-DL1 co-cultures, respectively. ELDA software was used to perform this calculation and to generate the graphs. Samples were analyzed in three (B cell LDA) and two (T cell LDA) independent experiments. zsG, $zsGreen$; tdT, $tdTomato$; n.d., not determined. Illustration in **a** created in BioRender; Brunelli, S. https://biorender.com/1jl9wh1 (2026).

highly expressed in tdT⁻ AGM pre-HSPCs, probably the most immature subset (Extended Data Fig. 8e). Taken together, our data show evidence of molecular heterogeneity within pre-HSPCs at the time of HSC emergence and confirm at the transcriptomic level that 4-OHT at E8.5 in *Cdh5-CreER^T2* mice labels the majority of E10.5 pre-HSPCs, probably representing the precursors of fetal-restricted HSPCs. Our data also suggest that pre-HSPCs in AGM and YS + VU include cells at distinct stages of maturation, with those located in the AGM representing the most uncommitted subset, and the ones in the YS + VU being primed for differentiation.

## Differentially labeled CD27⁺Kit⁺ hematopoietic clusters selectively localize to vitelline and umbilical arteries

*Cd27* was one of the highly enriched genes in pre-HSPCs (Fig. 6b and Extended Data Fig. 7b). It was recently shown to be expressed in HSCs, type 2 pre-HSCs and lymphoid progenitors in E10.5–E11.5 AGM[54,55]. To assess whether CD27 would allow a more specific localization of labeled pre-HSPCs, we performed whole-mount confocal imaging of E10.5 *Cdh5-CreER^T2* embryos (Fig. 7a). At this stage, we noticed that a higher percentage of cells within VU Kit⁺ cluster cells expressed CD27 as compared with DA or YS arteries (Fig. 7b), while clusters in the

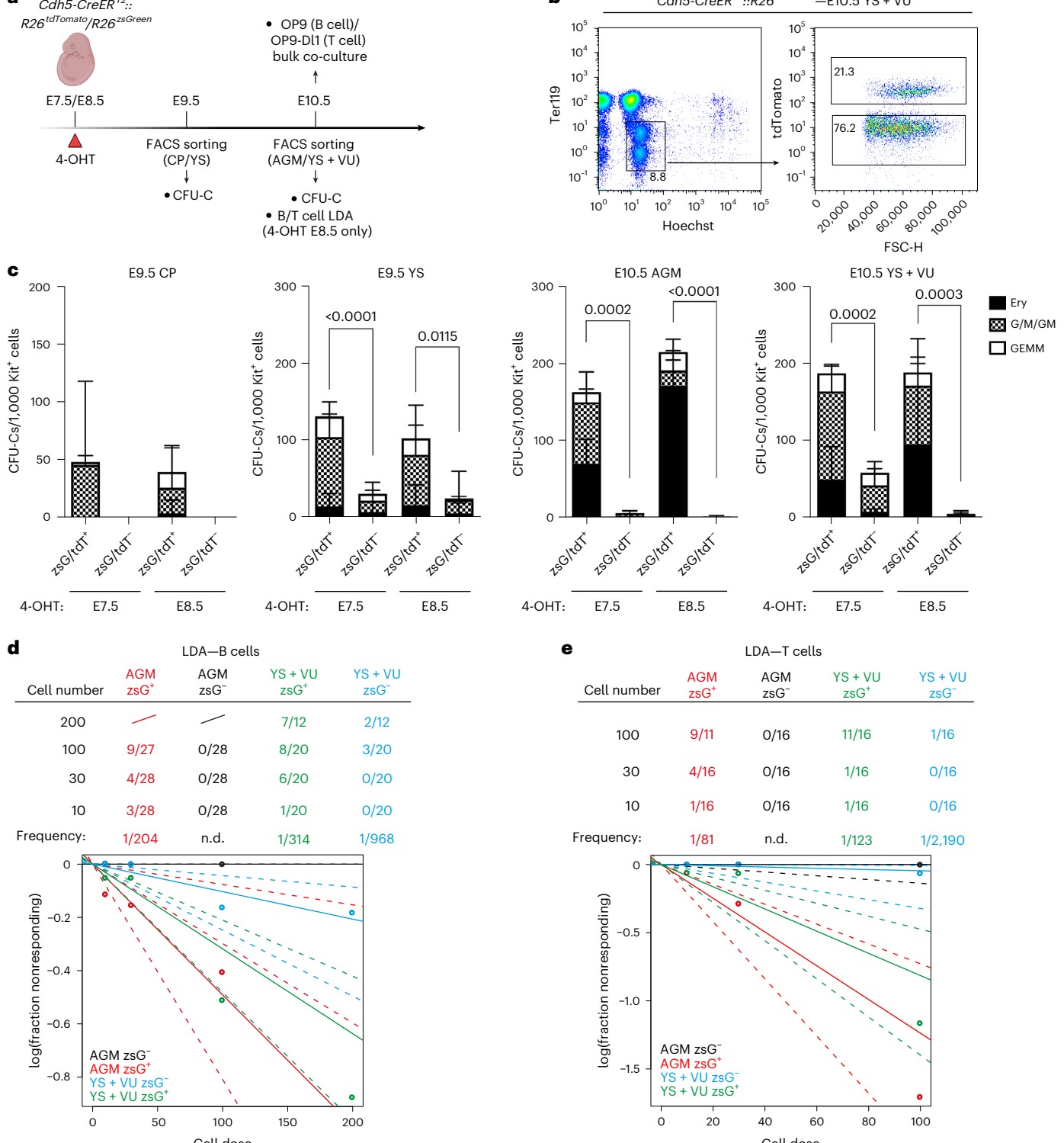

E9.5 YS did not express CD27 (Extended Data Fig. 9a), in agreement with the reported lack of CD27 expression in EMPs[54]. Evaluation of labeling of E10.5 Kit⁺CD27⁺ cluster cells showed that E7.5 activation labeled a minority of those clusters in both AGM and YS (Fig. 7a,c,d and Extended Data Fig. 9b). By contrast, 4-OHT at E8.5 yielded significantly higher labeling in the AGM (Fig. 7a,c) and in YS arteries (Fig. 7d and Extended Data Fig. 9b). Strikingly, within the AGM, labeled Kit⁺CD27⁺ clusters were detected in the VU but not in the DA (Fig. 7a,c). These results strongly suggest that CD27⁺ pre-HSPCs with lymphoid potential other than EMPs, identified with 4-OHT at E8.5 in *Cdh5-CreER*[T2] embryos, specifically localize to VU in both intra- and extra-embryonic regions.

### Fetal-restricted HSPCs and adult-type HSCs show distinct dynamics of engraftment

To establish whether fetal-restricted HSPCs could yield multi-lineage engraftment, we directly transplanted E11.5 AGM and YS + VU from *Cdh5-CreER*[T2]*::R26*[tdTomato]*/R26*[zsGreen] pulsed with 4-OHT at E8.5 (fetal-restricted HSPCs) or E10.5 (adult-type HSCs) into lethally irradiated syngeneic CD45.1 recipients (Fig. 8a,b). We did not detect any engraftment (defined as >5% donor chimerism) from YS + VU. Remarkably, all recipient mice repopulated with cells traced at E8.5, but none of those with cells traced at E10.5, showed high levels of donor-derived labeling in myeloid and lymphoid cells of the PB (Fig. 8c and Extended Data Fig. 10a). Analysis of BM at 16 weeks after transplantation confirmed PB data, with differentiated cells labeled only with E8.5 activation (Fig. 8d). Within the BM progenitor cell compartment, LSK, MPP and LK were labeled only in mice transplanted with AGMs traced at E8.5 (Fig. 8d and Extended Data Fig. 10b). Notably, immunophenotypic HSCs were not found labeled with either activation time point (Fig. 8d and Extended Data Fig. 10b). Within transplanted E11.5 AGM pools, a similar percentage of type 1 and type 2 pre-HSCs were labeled with both activations, while CD31⁺Kit⁺CD41^low^CD45⁺/⁻CD43⁺CD201⁻ progenitors were highly labeled with the E8.5 activation and not labeled with activation at E10.5 (Extended Data Fig. 10c), as previously shown (Fig. 1c). These data suggest that, in the E11.5 AGM, adult repopulation potential may be confined to fetal-restricted HSPCs and is likely to reside in the CD201⁻ subfraction, while adult-type HSCs labeled at E10.5 are not yet competent for engraftment and most probably require further maturation.

Next, we performed competitive transplants in which unfractionated cells from E14.5 *Cdh5-CreER*[T2]*::R26*[tdTomato]*/R26*[EYFP] FL pulsed with 4-OHT at E8.5 or E10.5 were transplanted into lethally irradiated recipient mice (Fig. 8a). Chimerism levels and labeling frequencies within donor-derived fractions were followed over time in PB. To account for variability in labeling efficiency, donor-derived labeling was normalized to the percentage of labeled LSK in each donor sample before transplantation (Extended Data Fig. 10d). Both E14.5 FL cells labeled at E8.5 and E10.5 showed multi-lineage engraftment potential in primary recipients (Fig. 8e and Extended Data Fig. 10e). However, analysis of

the progenitor compartment in BM of primary recipients at 16 weeks showed that only labeled cells pulsed at E10.5 could expand in the host BM niche (Fig. 8f and Extended Data Fig. 10f,g). We then performed secondary transplantations (Fig. 8a). In secondary recipients, the PB contribution of fetal-restricted HSPCs decreased, whereas that of adult-type HSCs significantly increased with time (Fig. 8e). After terminal analysis, no labeled phenotypic HSCs from donor cells pulsed at E8.5 were detected in the BM of secondary recipients, while labeled progenitors originally pulsed at E10.5, including phenotypic HSCs, were still detected (Fig. 8f).

Taken together, these results suggest that, during development, adult multi-lineage hematopoietic engraftment potential first appears in fetal-restricted HSPCs; however, these cells are biased for differentiation and devoid of long-term self-renewal potential. Our data are consistent with adult-repopulating HSCs being still largely immature in the E11.5 AGM; therefore, as recently proposed[33,63], they may require the FL microenvironment to complete their maturation.

## Discussion

The earliest intra-embryonic hematopoietic clusters are generated in the portion of the VA most proximal to the DA at E9.5 (refs. 9,45,46). Although the VU are known to harbor HSC precursors, this knowledge relied on primary transplantation experiments[7,47] in which post-transplant contribution to HSC was not assessed. Due to the lack of specific ways to identify and label VU-derived hematopoietic progenitors, their physiological role during development was unclear. Here we show in an unperturbed in vivo context that this vascular site is a source of multipotent hematopoietic progenitors that substantially contribute to fetal lympho-myelopoiesis. Using temporally resolved HE lineage tracing, whole-mount imaging and functional transplantation assays, we reveal that the HE of the VU originates HSPCs acting as major contributors to fetal lympho-myelopoiesis (Fig. 8g). Thus, the HE wave we identified contains the precursors of the recently described pre-constituted FL hematopoietic hierarchy[28,42], including fetal HSCs[32,33] and eMPPs[29,30], whereas adult-type HSCs are mainly generated by the HE of the DA. Given that the EHT output is heterogeneous, this raises the question of whether this diversification takes place after EHT, or it is already imprinted at the HE stage. Recent evidence supports the latter[64,65]. The hematopoietic contribution of fetal-restricted HSPCs rapidly declines after birth, marking them as a primarily transient population.

Our lineage tracing strategy cannot discriminate between the separate contributions of individual progenitors (that is, fetal HSCs or eMPPs) but rather identifies a temporally and spatially distinct wave of HE as the source of fetal-restricted HSPCs. Although the existence of fetal-restricted HSCs and eMPPs had been shown previously, the precise time and anatomical location of their emergence has not been investigated before. The first report to identify developmentally restricted HSCs was based on *Flt3* lineage tracing[32]; however, this study primarily relied on FL transplantation assays to probe their contribution and

**Fig. 6 | Combined HE lineage tracing and scRNA-seq identify distinct subsets of E10.5 pre-HSPCs. a**, UMAP of 26,180 cells (Ter119⁻Kit⁺/CD31⁺) isolated from AGM and YS + VU of E10.5 *Cdh5-CreER*[T2]*::R26*[tdTomato] embryos activated with 4-OHT at E8.5. AGM tdTomato⁺ (5,465), AGM tdTomato⁻ (8,135), YS + VU tdTomato⁺ (4,445) and YS + VU tdTomato⁻ (8,135) cells were separately sequenced (strategy for cell isolation shown in Extended Data Fig. 6e,f). Cells were taken from 9 AGM and 14 YS + VU (1 litter of 9 embryos for AGM; 2 litters of 11 and 3 embryos for YS + VU). Cells are colored according to individual cluster identities, and pre-HSPC clusters 1 and 2 are circled. A complete cell-type annotation is shown in Extended Data Fig. 7a. **b**, Bubble plot showing the expression level of HSC and hematopoietic progenitor genes for each cell cluster in the whole dataset. The dot size indicates the percentage of cells expressing each gene, and the dot color represents the gene expression level. Pre-HSPC 1 and 2 are highlighted with a black box and a gray box, respectively. **c**, UMAP plots showing cell cluster distribution of AGM tdTomato⁻, AGM tdTomato⁺, YS + VU tdTomato⁻ and YS + VU tdTomato⁺ samples.

Pre-HSPC clusters 1 and 2 are circled. **d**, Bar plot showing the relative contribution of AGM tdTomato⁻, AGM tdTomato⁺, YS + VU tdTomato⁻ and YS + VU tdTomato⁺ to pre-HSPC clusters 1 and 2. **e,f**, Gene expression of published signatures for HSC[55] (**e**) and AGM-derived progenitors ('Prog') and lymphoid progenitors ('Ly')[29] (**f**) in pre-HSPC 1 and pre-HSPC 2 clusters. The average expression of the top 50 genes for each signature was calculated on single cells using the AddModuleScore function in Seurat. **g**, UMAP plot showing the subclustering of pre-HSPC 1 and 2. Dots represent cells colored according to their origin. **h**, Heatmap showing the relative expression levels of pre-HSC signature genes among AGM tdTomato⁻, AGM tdTomato⁺, YS + VU tdTomato⁻ and YS + VU tdTomato⁺ cells within pre-HSPC clusters 1 and 2. **i**, Heatmap showing the relative expression levels of hematopoietic differentiation genes among AGM tdTomato⁻, AGM tdTomato⁺, YS + VU tdTomato⁻ and YS + VU tdTomato⁺ cells within pre-HSPC clusters 1 and 2.

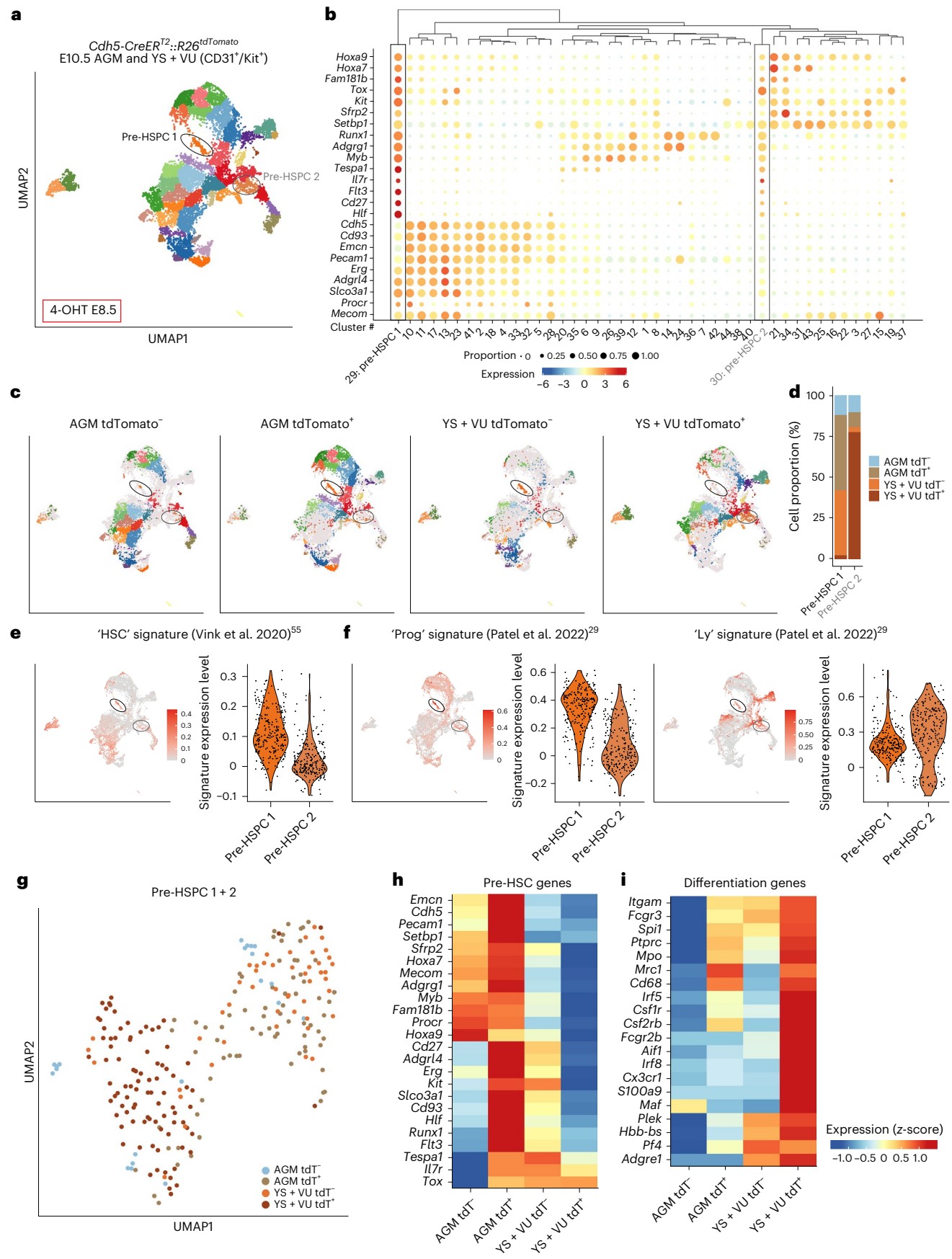

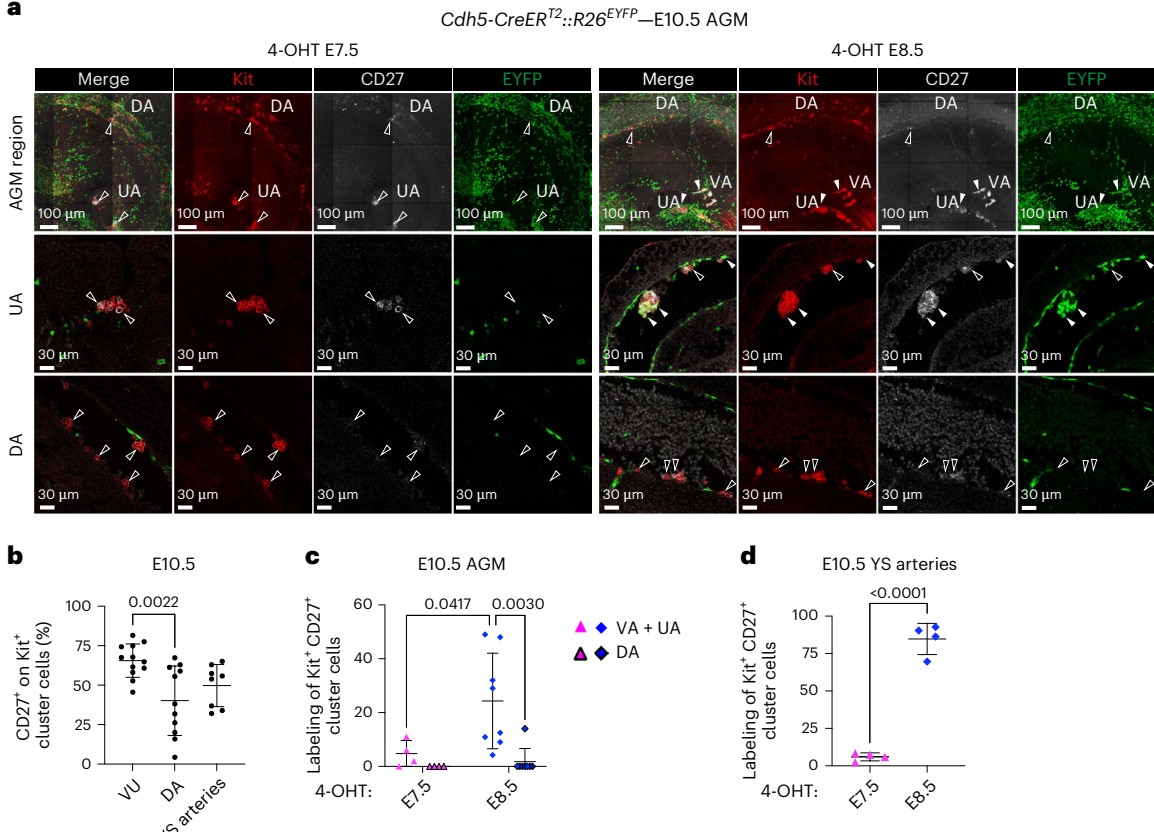

**Fig. 7 | CD27+ hematopoietic clusters emerging from E8.5–E9.5 HE specifically localize to the VU. a**, WM-IF confocal analysis of E10.5 *Cdh5-CreER^T2^::R26^EYFP^* AGM (4-OHT at E7.5 or E8.5). Upper panels show a 3D maximum-intensity projection. Middle (UA) and lower (DA) panels show single 2.5-μm-thick optical slices. Arrowheads indicate examples of labeled (EYFP+; filled arrowheads) or unlabeled (EYFP−; open arrowheads) Kit+CD27+ cells. 4-OHT E7.5 (*n* = 4) and 4-OHT E8.5 (*n* = 8) different YS were analyzed in 3 independent experiments. **b**, Quantification of WM-IF confocal analysis shown in **a**, representing the percentage of CD27+ cells on the total of Kit+ cells. DA, VU (*n* = 12) and YS arteries (*n* = 8) were analyzed in 7 independent experiments. Around 3–9 images per sample were quantified: 105 (DA, VU) and 52 (YS arteries) total images. Error bars represent the mean ± s.d. The *P* value is indicated in the figure (one-way ANOVA followed by Tukey's multiple comparisons test). **c**, Quantification of WM-IF confocal analysis in **a** representing the percentage of labeled EYFP+ cells on the total of Kit+CD27+ AGM cluster cells. The 4-OHT E7.5 (*n* = 4), and 4-OHT E8.5 (*n* = 8) embryos were analyzed in 4 independent experiments. Around 5–15 images per sample were used: 34 (4-OHT E7.5) and 71 (4-OHT E8.5) total images. Error bars represent mean ± s.d. *P* values are indicated in the figure (two-way ANOVA followed by Tukey's multiple comparisons test). **d**, Quantification of WM-IF confocal analysis shown in Extended Data Fig. 9b representing the percentage of labeled EYFP+ cells on the total of Kit+CD27+ cluster cells in YS arteries. Around 3–9 images per sample were quantified: 25 (4-OHT E7.5) and 27 (4-OHT E8.5) total images. Error bars represent the mean ± s.d. The *P* value is indicated in the figure (two-tailed unpaired Student's *t*-test).

thus offered little information on their origin and physiological role. A previous study also used *Flt3* lineage tracing, but adopted an inducible strategy coupled with clonal barcoding to identify a population of eMPPs that arise early in embryogenesis and contribute to postnatal multi-lineage output[29]. This and another recent study[30] mainly focused on postnatal contribution and did not investigate in depth the emergence of eMPPs. Another report showed that hematopoietic progenitors in the FL are generated independently of adult-type HSCs and that *Mecom* levels, normally higher in the intra-embryonic region, can play a functional role in HSPC specification[28,42]. We found *Mecom* to be more expressed in AGM than in YS + VU pre-HSPCs (Fig. 6h), in line with an identity of the latter distinct from adult-type HSCs. The high expression of *Mecom* in labeled AGM pre-HSPCs (Fig. 6h) confirms that *Mecom*-expressing cells are predominantly located in the intra-embryonic portion of the main arteries, including VU[28]. Thus, *Mecom* expression levels may not only be important to specify adult HSCs, consistent with *EVI1^creERT2^* labeling phenotypic HSCs in the E14.5 FL[28], which our data suggest to be in part fetal-restricted (Fig. 1f).

Our pulse-chase approach labels *Cdh5*-expressing endothelial and HE cells within a 12-h in vivo window[36,37]. Surprisingly, already 24 h after labeling, this resulted in near-exclusive recombination of hematopoietic clusters, while much lower tracing frequencies were detected in the HE or the adjacent vascular endothelium. HE is a transient hematopoietic-committed cell population, and it is not bipotent[51]. In addition, HE was shown to represent a lineage distinct from arterial endothelium[66]. Thus, our data suggest that labeled HE undergoes EHT in a rapid time window of a few hours in vivo, consistent with what was observed ex vivo in time-lapse imaging studies[6], and does not contribute to structural endothelium. The specificity of the labeling of distinct hematopoietic waves arises because of the sequential emergence of each HE, captured with our alternative activation modes. Neither 4-OHT activation at E7.5 nor at E8.5 labels (or labels very few) endothelial cells in the ventral side of the DA, providing a strong rationale for the lack of labeling of adult-type HSC (Fig. 3e and Fig. 7a). In addition, 4-OHT at E7.5 labels a minority of cells in the VU, supporting that this activation mode does not mark fetal-restricted HSPCs arising in this location. Moreover, the fact that we detected very low labeling of non-HE in hematopoietic sites even with the activation models that label hematopoietic cells in the same locations additionally raises the intriguing possibility that a substantial portion of the early embryonic endothelium of these regions may in fact be hemogenic, as recently proposed[67].

*Csf1r*-based lineage tracing clarified the relationship between EMPs and fetal-restricted HSPCs. Both intra- and extra-embryonic hematopoietic clusters in the VU arise independently of *Csf1r*+ progenitors, confirming that VU clusters do not contain EMPs or their progeny and suggesting that most of the *Csf1r*-expressing progenitors probably arise in the YS vascular plexus. We made the interesting observation that late-emerging *Csf1r*+ progenitors contributed modestly (<20%) to fetal myeloid and B lymphoid compartments and more markedly to thymus-settling progenitors in the fetal thymus. These data support the existence of at least two *Csf1r*+ progenitor waves: an early EMP wave restricted to erythro-myeloid output and a later wave with broader lympho-myeloid potential, probably overlapping with LMPs[20,25]. The limited contribution to B and myeloid cells in the E16.5 FL implies that, at this stage, the majority of these cells originate from *Csf1r*-independent progenitors, that is, fetal-restricted HSPCs. Nevertheless, the higher labeling frequencies observed in the thymus emphasize the functional uncoupling between T and B myeloid-producing progenitors during development, further supported by our LDA assay identifying different frequencies of B and T cell progenitors in both AGM and YS + VU[41,68,69].

scRNA-seq revealed two pre-HSPC subsets in the E10.5 embryo, prominently originated by HE specified between E8.5 and E9.5, with shared progenitor signatures (*Kit*, *Runx1*, *Hlf*, *Cd27*, *Mecom*, *Hoxa7*, *Hoxa9*) but overall distinct molecular programs. AGM-derived pre-HSPCs showed an uncommitted profile with a strong glycolytic signature, whereas YS + VU-derived cells showed increased oxidative metabolism, ribosomal activity and differentiation-priming transcripts. These transcriptional differences suggest that fetal-restricted HSPCs represent a heterogeneous mixture of progenitors, with intra-embryonic cells poised to mature into fetal HSCs and extra-embryonic cells already primed for differentiation. Functional heterogeneity between intra- and extra-embryonic HE has been shown[26] and recent work characterized their transcriptomic heterogeneity[31,65]. Interestingly, intra-embryonic HE shows elevated chromatin and RNA splicing gene expression and greater isoform complexity compared with YS HE[65].

Our transplantation experiments show that fetal-restricted HSPCs possess multi-lineage potential yet lack durable self-renewal. Given the labeling patterns observed in source tissues and in repopulated recipients, the progenitor subsets, which could yield engraftment in our E11.5 AGM transplants, were CD43+CD201− progenitors (probably the most differentiated fraction) and/or phenotypic CD201+ pre-HSCs and HSCs labeled at E8.5, but not those labeled at E10.5. Further experiments will be needed to validate this finding. Competitive FL transplants further confirmed that HSPCs labeled at E8.5 can yield multi-lineage engraftment in primary recipients but fail to persist after secondary transplantation, in contrast to the durable reconstitution by E10.5-labeled adult-type HSCs. These results underscore the intrinsic differentiation bias of fetal-restricted HSPCs and their functional distinction from definitive HSCs that may require the FL microenvironment for full maturation, as recently suggested[33,63]. Although we cannot formally exclude that part of the observed dynamics reflects differences in the timing of pre-HSC emergence and maturation, the combined evidence from lineage output, transplantation, postnatal contribution and spatial origin argues that our labeling strategy captures functionally heterogeneous HSPCs rather than a single homogeneous HSC pool.

Our data reinforce and enrich the concept that embryonic hematopoiesis is layered into multiple spatially and temporally distinct programs. While HSPC emergence is initially segregated in space, multi-lineage progenitors eventually converge in the FL[1]. For a long time, the FL was thought to be an intermediate reservoir for HSC expansion after their initial generation in the AGM, in preparation to their lifelong residency in the BM. Recent work revolutionized this concept and showed that FL is a complex hub in which distinct waves of progenitors differentiate, expand and mature with mechanisms still largely unknown[28,29,33,56]. Indeed, our and others' data imply that fetal-restricted HSPCs and adult-type HSCs coexist in the FL, where they undergo different processes leading, respectively, to differentiation and self-renewal. Testing to what extent HSPC fate is intrinsically pre-determined, and what exactly is the role of the microenvironment, will offer crucial insight. On the basis of our findings, it is tempting to speculate that before FL colonization, the emergence and transiting of fetal-restricted HSPCs in the VU niche may influence their identity and fate. Metabolic[45,70] and inflammatory factors[57–60] were shown to promote HSC development. Indeed, here we observed that these genes

**Fig. 8 | Fetal-restricted HSPCs yield multi-lineage engraftment, but they do not contain long-term HSCs and their contribution declines in serial transplantations. a**, Experimental schematic of E11.5 AGM and YS + VU and E14.5 FL transplantation assays. **b**, Percentage (%) of chimerism (percentage of donor CD45.2+) in adult lethally irradiated mice transplanted with E11.5 *Cdh5-CreER^T2^::R26^tdTomato^*/*R26^zsGreen^* AGM (2 e.e. per recipient), in PB (left) and BM (right) 16 weeks after transplantation. Each bar indicates a single recipient mouse. The dashed line indicates 5%, the threshold we considered for repopulation. **c**, Longitudinal analysis showing the frequency of labeled (zsGreen+ or tdTomato+) donor myeloid (CD45.2+CD11b+), B cells (CD45.2+B220+) and T cells (CD45.2+CD3e+) in PB of mice transplanted with E11.5 AGM. The gating strategy is shown in Extended Data Fig. 10a. A total of 4 (4-OHT E8.5) and 2 (4-OHT E10.5) mice were analyzed in 4 independent transplant experiments. Data are presented as mean ± s.d. A summary of E11.5 transplantation data is provided in Supplementary Table 4. **d**, Labeling frequency of donor myeloid, B and T cells, MPPs (Lin−CD45.2+Kit+Sca1+CD48−CD150−), LSK (Lin−CD45.2+Kit+Sca1+) LK (Lin−CD45.2+Kit+Sca1−) and HSCs (Lin−CD45.2+Kit+Sca1+CD48−CD150+) in BM of AGM transplanted mice. A total of 4 (4-OHT E8.5) and 2 (4-OHT E10.5) mice were analyzed in 4 independent transplant experiments. The gating strategy is shown in Extended Data Fig. 10b. Data are presented as mean ± s.d. **e**, Longitudinal analysis showing the percentage of PB chimerism (top) and normalized labeling frequency (bottom) in adult lethally irradiated mice transplanted with E14.5 *Cdh5-CreER^T2^::R26^EYFP^* or *Cdh5-CreER^T2^::R26^tdTomato^* FL cells activated with 4-OHT at E8.5 or E10.5 (1° Tx) or adult BM cells from 1° TX (2° TX). The frequency of labeled (EYFP+/tdTomato+) donor myeloid, B and T cells was normalized on the percentage of labeled LSK before transplant and shown as log₁₀. The gating strategy is shown in Extended Data Fig. 10e. *n* = 11 (4-OHT E8.5 1° Tx), *n* = 13 (4-OHT E10.5 1° Tx), *n* = 10 (4-OHT E8.5 2° Tx) and *n* = 11 (4-OHT E10.5 2° Tx)

recipient mice were analyzed in 2 independent experiments. Data are presented as mean ± s.d. A summary of PB data is provided in Supplementary Table 5 (1°Tx) and Supplementary Table 6 (2°Tx). Asterisks indicate statistical significance in comparisons between 4-OHT at E8.5 and 4-OHT at E10.5. *P* values: myeloid in 2° Tx at 12 weeks, *P* = 0.0332; and at 16 weeks, *P* = 0.0368; B cells in 2° Tx at 16 weeks, *P* = 0.0225; and T cells in 2° Tx at 16 weeks, *P* = 0.0288 (two-way ANOVA followed by Tukey's multiple comparisons test). **f**, Percentage of BM chimerism (top) and normalized labeling frequency (bottom) of HSCs, MPPs, LSK and LK subsets in primary and secondary transplanted mice from **e**, 16 weeks after transplant. The frequency of labeled donor HSCs, MPPs, LSKs and LKs was normalized on the initial labeling of each cell subset in the donor tissue (FL for 1° Tx or BM for 2° TX) and shown as log₁₀. The gating strategy is shown in Extended Data Fig. 10g. No labeled HSCs were found in 2° Tx recipients with 4-OHT at E8.5; log₁₀(0) is indicated with #. Data are presented as mean ± s.d. A summary of BM data is provided in Supplementary Table 5 (1° Tx) and Supplementary Table 6 (2° Tx). The *P* value as indicated in the figure (two-way ANOVA followed by Tukey's multiple comparisons test). **g**, Conceptual schematic summarizing timeline and anatomical sites of the subsets of HE and hematopoietic progenitors identified within this and other studies, along with labeling modes. Indicated is the contribution to fetal macrophages, myeloid cells, and B and T lymphocytes, with E16.5 as the reference time point. Asterisks indicate that a contribution was evaluated using multiple Cre lines; that is, macrophages and T cells are contributed by Csf1r+ progenitors (*Csf1r^MerCreMer^*, 4-OHT E9.5). However, because the combined contribution of Csf1r+ progenitors plus fetal-restricted HSPCs (*Cdh5-CreER^T2^*, 4-OHT E8.5) is higher than the Csf1r+ contribution alone, an exclusive contribution of fetal-restricted HSPCs can be inferred. Illustrations created in BioRender: **a**, Brunelli, S. https://biorender.com/ntjunf5 (2026); **g**, Brunelli, S. https://biorender.com/bgf6y8y (2026).

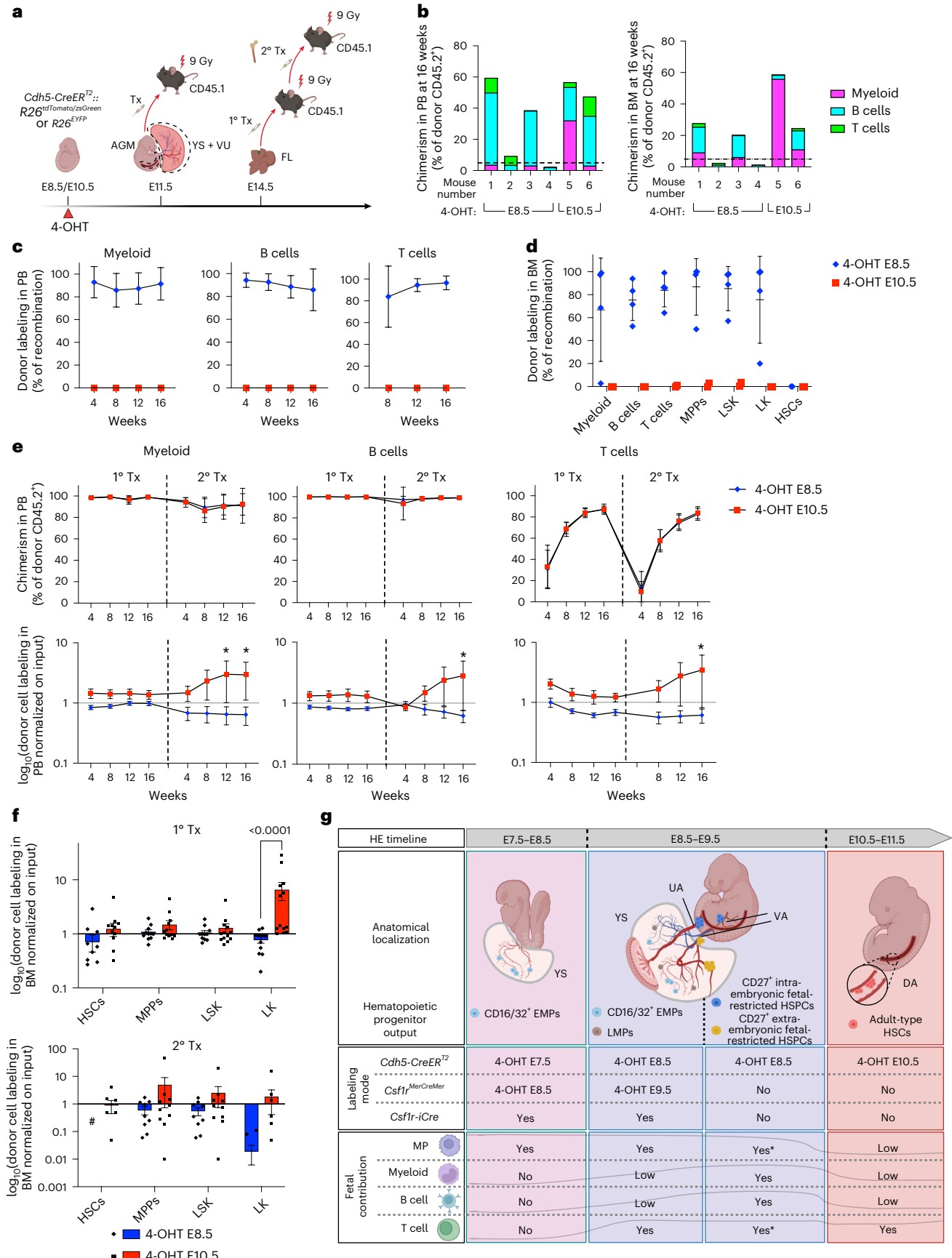

show variable levels of expression in subsets of pre-HSCs of different origins (Extended Data Figs. 7 and 8). Interferon signaling, more active in AGM-derived pre-HSPCs (Extended Data Fig. 8b), is required for the switch between fetal and adult HSCs, beginning before birth[71]. Therefore, it is possible that inflammatory cues are one of the main extrinsic factors required to specify adult-type HSC identity. Thus, it will be interesting to investigate what the specific signals are that regulate fetal-restricted HSPCs.

Collectively, our findings provide mechanistic and spatial resolution to the long-standing question of how fetal-restricted and adult-type hematopoietic programs are organized in the embryo. The discovery that a distinct subset of HE in the vitelline and umbilical arteries gives rise to a transient wave of multipotent progenitors that dominate fetal lympho-myelopoiesis highlights a previously overlooked site and mechanism of hematopoietic diversification. These insights refine our understanding of layered hematopoiesis and can inform studies of pediatric malignancies with prenatal origins, in which leukemic transformation may occur within these developmentally restricted progenitor pools.

## Methods

### Mice and embryos

$Cdh5\text{-}CreER^{T2}$ (ref. 35), $Csf1r\text{-}iCre$ (ref. 72), $Csf1r^{MerCreMer}$ (ref. 73), $R26^{zsGreen}$, $R26^{tdTomato}$ (ref. 74) and $R26^{EYFP}$ (ref. 75) transgenic mice were previously described and were genotyped according to reported protocols (further details and primers are listed in Supplementary Table 1). $R26^{zsGreen}$, $R26^{tdTomato}$ or $R26^{EYFP}$ females aged 6–16 weeks were subjected to overnight timed matings with $Cdh5\text{-}CreER^{T2}$, $Csf1r\text{-}iCre$ or $Csf1r^{MerCreMer}$ males. Successful mating was judged by the presence of vaginal plugs the next morning, which was considered 0.5 days after conception (E0.5). Embryos were collected and dissected in phosphate-buffered saline (PBS) supplemented with 10% fetal bovine serum (FBS), EDTA (2 mM), 50 U ml$^{-1}$ penicillin and 50 mg ml$^{-1}$ streptomycin[45,51,76]. E9.5–E11.5 embryos were carefully staged by counting somite pairs; older embryos were staged by morphological criteria. For $Cdh5\text{-}CreER^{T2}$ fate mapping, a single dose of 37.5 mg kg$^{-1}$ of 4-OHT dissolved in corn oil was delivered by intraperitoneal injections to pregnant females at E7.5, E8.5 or E10.5. To counteract adverse effects of 4-OHT on pregnancies, 4-OHT solutions were supplemented with progesterone (18.75 mg kg$^{-1}$). For $Csf1r^{MerCreMer}$ fate mapping, a single dose of 75 mg kg$^{-1}$ 4-OHT without progesterone was used. All transgenic mouse lines were maintained on a CD45.2 C57BL/6 genetic background, except for females used for $Csf1r^{MerCreMer}$ timed matings and other matings for generation of adult mice with 4-OHT activation during embryogenesis, which were, instead, of C57BL/6/FVB mixed background (F1).

Mice were housed in individually ventilated cages or filter top cages with a 12 h:12 h light–dark cycle (350/450 lux) and unrestricted access to food and water in the animal facilities at the San Raffaele Scientific Institute, University of Oxford, or at the University of Milan. Standardized housing conditions included 22 °C (±2 °C) temperature and relative humidity of 55% (±5%). All mouse experiments were performed in accordance with experimental protocols approved by the San Raffaele Scientific Institute and the University of Milan Institutional Animal Care and Use Committees and authorized by the Italian Ministry of Health (authorization numbers 503/2019-PR, 753/2023-PR and 351/2022-PR). All procedures carried out at the University of Oxford were in compliance with United Kingdom Home Office regulations and the Oxford University Clinical Medicine Animal Welfare and Ethical Review Committee (Project Licence number PP9552402).

### Flow cytometry analysis and cell sorting

Single-cell suspensions were obtained from embryonic tissues (yolk sac, embryo caudal part, FL) by incubating for 15 min at 37 °C in calcium–magnesium-free PBS supplemented with FBS 10%, 50 U ml$^{-1}$ penicillin, 50 mg ml$^{-1}$ streptomycin, EDTA 2 mM and collagenase type I (Sigma) 0.12% (w/v), followed by mechanical dissociation by pipetting. Single-cell suspension from the fetal thymus was obtained by mechanical dissociation and passed through a 26-gauge needle. PB samples were collected by tail vein bleeding using a scalpel; BM was obtained by flushing long bones using a syringe and filtered in 40-μm strainers. PB, BM and FL samples were treated with the appropriate amount of RBC lysis buffer. For flow cytometry analysis and cell sorting, single-cell suspensions were incubated with conjugated antibodies at 4 °C in the dark for 15 min (refs. 45,51,76). A list of antibodies used for flow cytometry can be found in Supplementary Table 1. Voltages, compensation and gates were set using unstained, single-stained and fluorescence-minus-one controls. Dead cells were excluded based on Hoechst 33258 (Hellobio) or 7-AAD (Sigma) incorporation. Flow cytometry data acquisition was carried out using a LSR Fortessa X-20 (BD) analyzer and BD FACSDiva software (version 8.0.2). Cell sorting was performed using a MoFLO Astrios cell sorter equipped with Summit software version 6.3 (both from Beckman Coulter), a BD FACSDiscover S8 with FACS Chorus software, or FACSAria II with BD FACSDiva software. An average sorting rate of 500–1,000 events per second at a sorting pressure of 25 psi with a 100-μm nozzle was maintained. Flow cytometric data were analyzed using FlowJo software version 10 (BD).

### CFU-C assays

YS + VU and caudal parts were dissected from E9.5 (21–27 somite pairs (sp)) or E10.5 (32–36 sp) $Cdh5\text{-}CreER^{T2}::R26^{tdTomato}$, $Cdh5\text{-}CreER^{T2}::R26^{zsGreen}$ or $Cdh5\text{-}CreER^{T2}::R26^{EYFP}$ concepti. Three YS + VU and caudal parts were each pooled and processed into single-cell suspensions. Labeled and unlabeled cells were isolated by flow cytometry. CFU-C assays were performed using Methocult M3434 (Stem Cell Technologies), according to the manufacturer's instructions. Cells were plated in duplicate dishes and cultured at 37 °C and 5% $CO_2$ in a humidified chamber. Colonies were scored after 7 days. Numbers of colonies are normalized to 1,000 Kit$^+$ cells seeded, considering the average of the percentage of Kit$^+$ cells in labeled and unlabeled fractions of Ter119$^-$ cells of YS + VU and caudal parts from E9.5 or E10.5 $Cdh5\text{-}CreER^{T2}::R26^{tdTomato}$, $Cdh5\text{-}CreER^{T2}::R26^{zsGreen}$ or $Cdh5\text{-}CreER^{T2}::R26^{EYFP}$ concepti, activated with 4-OHT at E7.5 or E8.5.

### Bulk and limiting dilution OP9 and OP9-Dl1 co-cultures

For bulk cultures, yolk sacs with YS + VU and caudal parts were dissected from E10.5 (32–36 sp) $Cdh5\text{-}CreER^{T2}::R26^{tdTomato}$ or $Cdh5\text{-}CreER^{T2}::R26^{zsGreen}$ concepti activated with 4-OHT at E7.5 or E8.5. Three YS + VU and caudal parts were each pooled and processed into single-cell suspensions. From each sample, 1 embryo equivalent (e.e.) was analyzed by flow cytometry to determine the percentage of labeling (tdTomato$^+$ or zsGreen$^+$) in Kit$^+$ cells. Then, 1 e.e. was plated on confluent OP9 (mouse BM stromal cell line; ATCC CRL-2749) or Delta-like 1-expressing OP9 (OP9-Dl1; a gift from J.C. Zuniga-Pflucker)[77] stromal cells in six-well plates in induction medium (αMEM, 10% FBS, supplemented with 10 ng ml$^{-1}$ IL-7 and 10 ng ml$^{-1}$ Flt3 ligand). At 4 days, the exhausted medium was replaced with fresh medium. At 7 days, hematopoietic cells that grew in suspension were collected from each well, split 1:2 and plated on new OP9 and OP9-Dl1 confluent cells. The two resulting samples were considered a technical duplicate. After 10 days from the start of the culture, non-adherent cells were collected and analyzed by flow cytometry.

For LDAs, YS + VU and AGM were dissected from E10.5 (32–36 sp) $Cdh5\text{-}CreER^{T2}::R26^{zsGreen}$ embryos activated with 4-OHT at E8.5. YS + VU and AGM from an entire litter (7–12 e.e.) were each pooled and processed into single-cell suspensions. Sorted hemato-endothelial cells (sorting strategy shown in Extended Data Fig. 6e,f) were plated on confluent OP9 or OP9-Dl1 stromal cells in 96-well plates at doses of 10, 30, 100 or 200 cells per well. For B cell differentiation, cultures were maintained in αMEM with 10% FBS supplemented with 10 ng ml$^{-1}$ IL-7 and 10 ng ml$^{-1}$ Flt3 ligand. OP9 feeders and medium were refreshed every

4–5 days according to the status of the stromal layer, and non-adherent cells were collected and analyzed by flow cytometry on day 10. For T cell differentiation, cultures were maintained in αMEM with 20% FBS supplemented with 10 ng ml$^{-1}$ IL-7 and 10 ng ml$^{-1}$ Flt3 ligand. The OP9-Dl1 feeders and medium were refreshed every 4–5 days according to the status of the stromal layer, and non-adherent cells were collected and analyzed by flow cytometry on day 14. ELDA software was used to calculate the frequency of B and T cell progenitors[78].

### Whole-mount immunofluorescence analysis and imaging

Whole-mount immunofluorescence (WM-IF) of embryonic tissues was performed according to previously described protocols[45,79]. Briefly, embryos and yolk sacs were dissected and fixed in a 4% paraformaldehyde solution in PBS for 30 min to 2 h at 4 °C. For E10.5 embryos, the limb buds and body wall were removed before fixation to expose the aorta and the main intra-embryonic arteries. Next, samples were treated with a permeabilizing-blocking solution (0.4% Triton X-100, 2% donkey serum, 2% FBS, 0.2% bovine serum albumin) and incubated overnight at 4 °C in the dark with primary antibodies. A second step of incubation with appropriate secondary antibodies was then carried out in the same conditions. Antibodies used for WM-IF are listed in Supplementary Table 1. After staining, embryos were cleared in a benzyl alcohol–benzyl benzoate solution and mounted on Superfrost glass slides with FastWell flexible silicone gaskets (Grace Bio-Labs). Yolk sacs were cleared overnight in a 50% solution of glycerol in PBS at 4 °C and then flat-mounted on Superfrost glass slides in the same solution. Samples were imaged using a Zeiss 710 confocal microscope equipped with an LD LCI Plan-Apochromat 25×/0.8 Imm Corr DIC M27 objective or an EC Plan-Neofluar 40×/1.30 Oil DIC M27 objective. Confocal image acquisition was carried out using Zeiss Zen software version 2.3 SP1; image processing and analysis were carried out using Imaris and Imaris-Viewer software version 10.2 and earlier versions (Bitplane), ImageJ/Fiji (versions 2.3.5-2.9.0) and Adobe Photoshop 2024 and earlier versions.

### scRNA-seq of E10.5 AGM and YS + VU

AGM and YS + VU dissected from E10.5 *Cdh5-CreER$^{T2}$::R26$^{tdTomato}$* embryos (31–37 sp; 4-OHT at E8.5) were processed into single-cell suspensions as described here. Live Ter119$^-$CD31$^+$/Kit$^+$ cells were isolated by FACS as shown in Extended Data Fig. 6e,f and split into tdTomato$^+$ and tdTomato$^-$. Labeled and unlabeled cells were analyzed separately.

scRNA-seq libraries were generated using a Chromium instrument (10x Genomics) with a Next GEM Single Cell 3′ kit. Libraries were quantified using a Qubit fluorometer (Thermo Fisher) and their profile was analyzed using a TapeStation instrument (Agilent). NGS sequences were generated using a Novaseq 6000 instrument (Illumina) with a target of 25,000 reads per cell. Following multiplexing, raw fastq reads were processed using cellranger v6.1 and aligned against the mm10 mouse genome (GENCODE vM23/Ensembl 98) modified with the mkref tool to add an artificial 'tdTomato' chromosome. The associated genome annotation GTF file was modified accordingly. Filtered count matrices generated with cellranger were processed with Seurat v4.0 (ref. 80) package implemented in R (versions 3.2.3–4.2.1). Cells with gene counts >300 and <8,000 and fractions of mitochondrial reads <0.20 were kept for downstream processing.

Following the filtering process, the dataset included 26,180 cells (5,465 AGM tdTomato$^+$, 8,135 AGM tdTomato$^-$, 4,445 YS + VU tdTomato$^+$ and 8,135 YS + VU tdTomato$^-$). After individual matrices were converted in Seurat objects via the Read10X function, data were normalized and transformed using the SCTransform Variance Stabilizing Transformation using the glmGamPoi method, while also regressing out for feature counts, percentages of mitochondrial counts and cell phases. Data generated from both samples were subsequently integrated with a canonical correlation analysis using the PrepSCTIntegration, FindIntegrationAnchors and IntegrateData commands, by using SCT as the normalization method. Dimensionality reduction of the integrated

data was initially carried out using principal component analysis and subsequently with uniform manifold approximation and projection (UMAP) algorithms, by retaining the first 30 principal components of the principal component analysis. Clusters were identified with the Louvain algorithm; their number was selected using the Clustree tool[81] (version 0.4.3) by maximizing cluster stability. Individual cell types were identified by using SingleR (ref. 82; version 1.0.1) as well as with manual data curation. A list of software programs and packages used for scRNA-seq analysis can be found in Supplementary Table 1.

Expression of publicly available gene signatures (top 50 differentially expressed genes (DEGs)) for nascent embryonic HSCs and progenitors[29,55] was assessed in our dataset using the AddModuleScore function in Seurat (ctrl = 100).

### GO biological process gene set enrichment analysis

Gene set enrichment analysis was conducted using a ranked list of DEGs ($P < 0.05$) between pre-HSPC 2 and pre-HSPC 1 (Extended Data Fig. 7d) and a ranked list of DEGs between AGM tdT$^+$ and YS + VU tdT$^+$ pre-HSPCs (Extended Data Fig. 8d). Analyses were carried out using R package ClusterProfiler (version 4.8.3)[83] querying the Gene Ontology (GO) database. The analysis was performed by setting the number of permutations to 10,000 and the minimum gene set to 3 and the maximum to 800. GO terms were considered significant with a selected cutoff $P$ value of 0.05.

### In vivo transplantation

For AGM and YS + VU transplantation experiments, tissues were dissected from E11.5 *Cdh5-CreER$^{T2}$::R26$^{zsGreen}$* embryos (4-OHT at E8.5 or E10.5). Single-cell suspensions were prepared as described here. Syngeneic C57BL/6 (CD45.1) recipient mice were lethally irradiated (9 Gy, split dose) before intravenous transplantation of single-cell suspensions obtained from 2 embryo equivalents of each tissue. For FL transplantation experiments, syngeneic C57BL/6 (CD45.1) recipient mice were lethally irradiated (9 Gy, split dose) before intravenous transplantation of $1 \times 10^6$ unfractionated FL cells (primary transplantation) or $2 \times 10^6$ adult BM cells (secondary transplantation) from E14.5 *Cdh5-CreER$^{T2}$::R26$^{TdTomato}$* or *Cdh5-CreER$^{T2}$:R26$^{EYFP}$* mice (4-OHT at E8.5 or E10.5). For both AGM and YS + VU transplantation experiments, and FL transplantation experiments, donor-derived chimerism and percentage of labeling (tdTomato$^+$, EYFP$^+$ or zsGreen$^+$) within donor cells was determined by flow cytometry in PB at 4 weeks, 8 weeks and 12 weeks after transplantation, and in PB and BM at 16 weeks after transplantation. BM from primary and secondary transplanted mice was analyzed by flow cytometry to determine the percentage of labeling of donor hematopoietic stem and progenitor cells. A comprehensive summary of transplantation analysis data is provided in Supplementary Table 4 (E11.5 AGM transplants), Supplementary Table 5 (FL primary transplants) and Supplementary Table 6 (FL secondary transplants).

### Quantification and statistical analysis

Statistical analyses were performed using GraphPad Prism v10.2.1 and later versions. No specific randomization method was used. Animals were allocated into experimental groups according to their genotype. The investigators were not blinded to group allocations during data collection and analysis. In each of the graphs shown in the figures, individual data points represent biological replicates. To determine the level of significance, unpaired two-tailed Student *t*-test and one-way and two-way ANOVA followed by Tukey's multiple comparisons test were used as indicated in figure legends. $P < 0.05$ was considered statistically significant, and the level of significance is indicated where relevant.

### Reporting summary

Further information on research design is available in the Nature Portfolio Reporting Summary linked to this article.

## Data availability

All data supporting the findings of this study are included in the Article and Supplementary Information. Raw scRNA-seq data of YS + VU and AGM from E10.5 *Cdh5-CreER*[T2]*::R26*[tdTomato] embryos (4-OHT at E8.5) are available at the National Center for Biotechnology Information Sequence Read Archive data repository under accession number Bio-Project ID PRJNA898269 (sample accession numbers: SRR28006358, SRR28006359, SRR28006360 and SRR28006361). Source data are provided with this paper.

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

## Acknowledgements

We thank R. Gamberale, A. Giovenzana, G. Di Meo and the personnel of the animal facilities at San Raffaele Scientific Institute and the University of Milan for their support and technical help. We thank M. Iannacone (IRCCS San Raffaele Scientific Institute, Milan, Italy) for

providing *R26^zsGreen* mice. E.A. was supported by Fondazione Cariplo (grant number 2018-0102), Leukemia Research Foundation (award ID: 831382), Cariplo Telethon Alliance (grant number GJC22013) and Worldwide Cancer Research (grant reference 24-0083). E.A. and A.F. were supported by the Italian Ministry of University and Research (grant numbers P20223HEZC and 2022TT8Z3M). C. Barone was supported by Fondazione Umberto Veronesi. E.A. and C. Barone acknowledge support from the COST (European Cooperation in Science and Technology) Action IMMUNO-model, CA21135. G.A. and M.D.B. were supported by program MC_UU_00029/5 in the Medical Research Council (MRC) Molecular Haematology Unit (MHU) Core award, as were experiments performed by Y.P. and N.T.M. The flow cytometry facility at the MRC Weatherall Institute of Molecular Medicine (WIMM) is supported by the MRC Translational Immune Discovery Unit (TIDU), MRC MHU (MC_UU_12009), National Institute for Health and Care Research (NIHR) Oxford Biomedical Research Centre (BRC), Kay Kendall Leukaemia Fund (KKL1057), John Fell Fund (131/030 and 101/517), the Edward Penley Abraham (EPA) fund (CF182 and CF170) and the MRC WIMM Strategic Alliance awards G0902418 and MC_UU_12025. Work in the ADi laboratory is supported by the Italian Telethon Foundation (SR-Tiget grant award TGT21007). F.S.-d.-S. was funded by a postdoctoral grant from REVIVE (ANR-10-LABX-73). A.F. was supported by the Fondazione Cariplo (2018-0298) and the Associazione Italiana per la Ricerca sul Cancro (22905). S. Brunelli was supported by the European Union's Horizon 2020 research and innovation program under Marie Skłodowska-Curie grant agreement number 860034. F.T.-F. was supported by the European Union's Horizon 2020 research and innovation program under the Marie Skłodowska-Curie grant agreement number 860034 and Cariplo Telethon Alliance grant number GJC22013. R.P. was supported by Associazione Italiana Ricerca sul Cancro IG-29341, Italian MUR Dipartimenti di Eccellenza 2023–2027 (l. 232/2016, art. 1, commi 314–337) and European Union—NextGenerationEU through the Italian Ministry of University and Research under PNRR—M4C2-I1.3 Project PE_00000019 'HEAL ITALIA'.

## Author contributions

C. Barone conceived, designed and performed experiments; analyzed and interpreted data; made figures; and wrote the paper. G.Q., R.O., F.T.-F and A.M. performed experiments, analyzed and interpreted the data, and edited the paper. G.A. analyzed and interpreted data and performed bioinformatic analysis. Y.P., N.T.M., E.M., A. Cazzola, A.P. and M.N. performed experiments and analyzed data. T.M., B.N., F.S.-d.-S., G.Z., V.B., A. Domingues, E.D.E., G.S. and A. Colonna performed experiments. M.M., D.D.A. and S.S. made scRNA-seq libraries. S. Bombelli and S.D.M. performed cell sorting. C.D.O. and M.N.A. provided technical support. C. Bianchi, R.A.P., R.M. and M.F.T.R.D.B. provided reagents and materials. A. Cumano and S. Brunelli provided reagents and materials and interpreted data. A. Ditadi and A.F. performed experiments, interpreted data and provided reagents and materials. R.A.P. provided reagents and materials, analyzed and interpreted data, and performed bioinformatic analysis. E.A. conceptualized, supervised and led the study; performed experiments; analyzed and interpreted data; made figures; and wrote the paper. All authors read the paper and approved its final version.

## Competing interests

The authors declare no competing interests.

## Additional information

**Extended data** is available for this paper at https://doi.org/10.1038/s44161-026-00793-8.

**Correspondence and requests for materials** should be addressed to Emanuele Azzoni.

¹School of Medicine and Surgery, University of Milano-Bicocca, Monza, Italy. ²MRC Molecular Haematology Unit, MRC Weatherall Institute of Molecular Medicine, Radcliffe Department of Medicine, University of Oxford, Oxford, UK. ³San Raffaele Telethon Institute for Gene Therapy, IRCCS San Raffaele Scientific Institute, Milan, Italy. ⁴Vita-Salute San Raffaele University, Milan, Italy. ⁵Unit of Lymphocytes and Immunity, Immunology Department, Institut Pasteur, Paris, France. ⁶INSERM U1223, Paris, France. ⁷Université Paris Cité, Cellule Pasteur, Paris, France. ⁸National Facility for Light Imaging, Flow Cytometry Unit, Human Technopole, Milan, Italy. ⁹Department of Biosciences, University of Milan, Milan, Italy. ¹⁰Fondazione IRCCS San Gerardo dei Tintori, Monza, Italy. ¹¹Present address: Biomedical Research in Melanoma-Animal Models and Cancer Laboratory, Vall d'Hebron Research Institute (VHIR), Vall d'Hebron Hospital Barcelona-UAB, Barcelona, Spain. ¹²Present address: Department of Biology, University of Copenhagen, Copenhagen, Denmark. ¹³Present address: Roche Pharma Research and Early Development, Pharmaceutical Sciences, Roche Innovation Center Basel, F. Hoffmann-La Roche Ltd., Basel, Switzerland. ¹⁴Present address: Université Paris Cité INSERM 1144 OPTEN, Paris, France. ¹⁵These authors contributed equally: Cristiana Barone, Giulia Quattrini. ✉e-mail: emanuele.azzoni@unimib.it

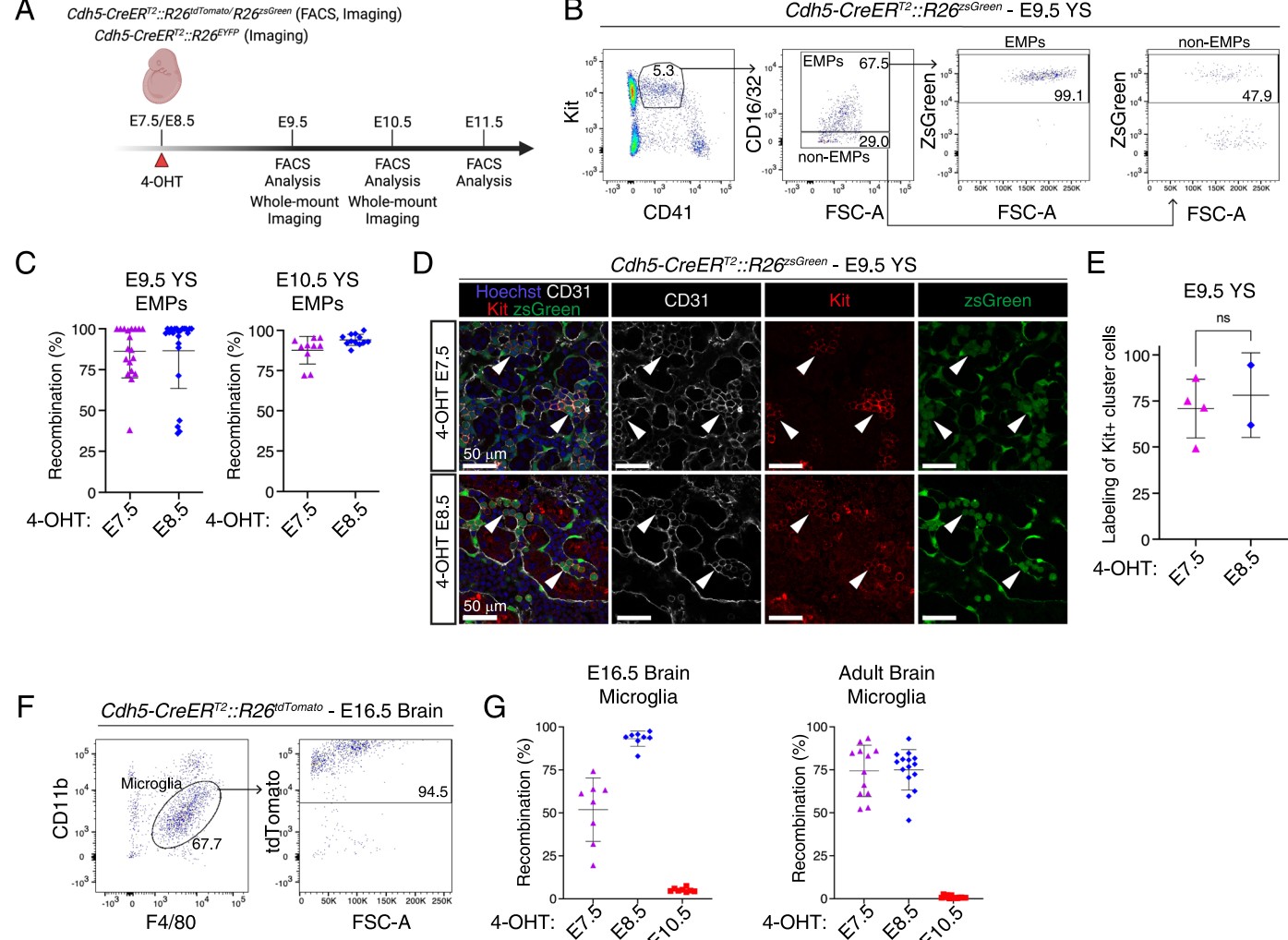

**Extended Data Fig. 1 | Lineage tracing analysis of EMPs and microglia in** *Cdh5-CreER^T2* **mice.** Related to Fig. 1. (**A**) Visual schematic of lineage tracing experiments in E9.5, E10.5 and E11.5 embryos. (**B-C**) Representative flow cytometric analysis (**B**) and labelling quantification (**C**) of EMPs (Ter119⁻ CD41^lo Kit⁺ CD16-32⁺) and non-EMPs (Ter119⁻ CD41^low Kit⁺ CD16-32⁻) in E9.5 *Cdh5-CreER^T2::R26^zsGreen* yolk sac (YS), activated with 4-OHT E7.5. E9.5 4-OHT E7.5 (n = 19), E9.5 4-OHT E8.5 (n = 23), E10.5 4-OHT E7.5 (n = 10), E10.5 4-OHT E8.5 (n = 12) YS were analyzed individually across 5 independent experiments. Error bars represent mean ± s.d. (**D**) Confocal whole mount immunofluorescence (WM-IF) analysis of E9.5 *Cdh5-CreER^T2::R26^zsGreen* YS. Arrowheads indicate CD31⁺ Kit⁺ hematopoietic cell clusters, labeled with 4-OHT at E7.5 (top) and E8.5 (bottom).

4-OHT E7.5 (n = 4), and 4-OHT E8.5 (n = 2) YS were analyzed. Scale bar: 50 µm. (**E**) Quantification of WM-IF analysis in (**D**). 3-9 images/sample were quantified; 12 (4-OHT E7.5) and 6 (4-OHT E8.5) total images. 4-OHT E7.5 (n = 4), and 4-OHT E8.5 (n = 2) YS were analyzed. Error bars represent mean ± s.d. ns = non-significant (two-tailed unpaired Student's *t*-test). (**F-G**) Representative flow cytometric analysis (**F**) and labeling quantification (**G**) of brain microglia (CD45⁺ CD11b^low F4/80⁺) in *Cdh5-CreER^T2::R26^tdTomato* E16.5 embryos, activated with 4-OHT at E7.5, E8.5 (shown here) or E10.5. 4-OHT E7.5 (n = 8), 4-OHT E8.5 (n = 8), 4-OHT E10.5 (n = 8) were analyzed individually across 3 independent experiments. Error bars represent mean ± s.d. Illustration in **a** created in BioRender; Brunelli, S. https://biorender.com/drgis12 (2026).

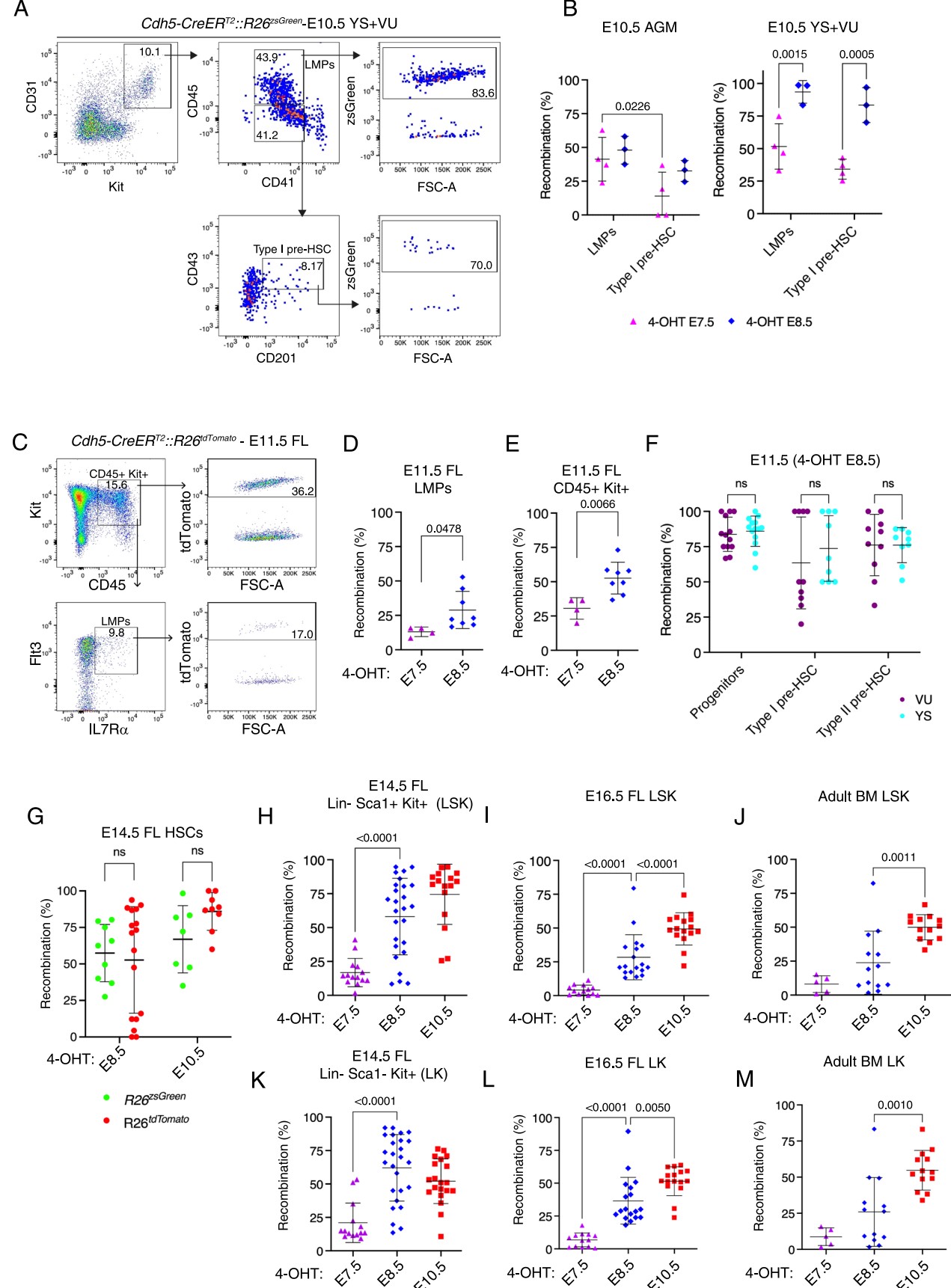

**Extended Data Fig. 2 | See next page for caption.**

**Extended Data Fig. 2 | Flow cytometric analysis of LMPs, LSK and LK labeling in *Cdh5-CreER^{T2}* embryos and adult mice.** Related to Fig. 1. (**A-B**) Representative flow cytometric analysis (**A**) and labeling quantification (**B**) of LMPs (Ter119⁻ CD31⁺ Kit⁺ CD45⁺ CD41⁺) and type 1 pre-HSC (Ter119⁻ CD31⁺ Kit⁺ CD45⁻ CD41^{low} CD43⁺ CD201⁺) in E10.5 *Cdh5-CreER^{T2}::R26^{zsGreen}* AGM and YS + VU, labeled with 4-OHT at E7.5, E8.5 (shown here). 4-OHT E7.5 (n = 4), 4-OHT E8.5 (n = 3) were analyzed individually across 2 independent experiments. Error bars represent mean ± s.d. P-values as reported in figure (two-way ANOVA followed by Tukey's multiple comparisons test). (**C-D**) Flow cytometric analysis (**C**) and labeling quantification (**D**) of LMPs (Lin⁻ CD45⁺ Kit⁺ Flt3⁺ IL7Ra⁺) in E11.5 *Cdh5-CreER^{T2}::R26^{tdTomato}* fetal liver (FL). 4-OHT E7.5 (n = 4), and 4-OHT E8.5 (n = 8) FL were analyzed individually in 2 independent experiments. Error bars represent mean ± s.d. P-value as reported in figure (two-tailed unpaired Student's *t*-test). (**E**) Quantification of flow cytometric analysis of labeled hematopoietic progenitor cells (CD45⁺ Kit⁺) in E11.5 *Cdh5-CreER^{T2}::R26^{tdTomato}* fetal liver (FL) identified as in (**C**). Replicates as in (**C-D**). Error bars represent mean ± s.d. P-value as reported in figure, (two-tailed unpaired Student's *t*-test). (**F**) Quantification

of flow cytometric analysis of labeled (zsGreen⁺) progenitors, type 1 and type 2 pre-HSCs in E11.5 *Cdh5-CreER^{T2}::R26^{zsGreen}* VU and YS when dissected separately (gating strategy shown in Fig. 1b). Error bars represent mean ± s.d. ns = not significant (two-way ANOVA followed by Tukey's multiple comparisons test). (**G**) Comparison of flow cytometric analysis of E14.5 FL HSCs as shown in Fig. 1e, shown separately for *Cdh5-CreER^{T2}::R26^{zsGreen}* and *Cdh5-CreER^{T2}::R26^{tdTomato}* embryos. Error bars represent mean ± s.d. (**H-M**) Quantification of flow cytometric analysis of labeled LSK (Lineage⁻Kit⁺Sca1⁺) and LK hematopoietic progenitor cells (Lineage⁻Kit⁺Sca1⁻) in *Cdh5-CreER^{T2}::R26^{tdTomato}* or *Cdh5-CreER^{T2}::R26^{zsGreen}* E14.5 FL (**H** for LSK, **K** for LK), E16.5 FL (**I** for LSK, **L** for LK) and adult BM (2-months-old) (**J** for LSK, **M** for LK) activated with 4-OHT at E7.5, E8.5 or E10.5, related to Fig. 1e. (E14.5 4-OHT at E7.5 (n = 14), E14.5 4-OHT at E8.5 (n = 26), E14.5 4-OHT at E10.5 (n = 16), E16.5 4-OHT E7.5 (n = 13), E16.5 4-OHT E8.5 (n = 18), E16.5 4-OHT E10.5 (n = 17), 2 months 4-OHT E7.5 (n = 5), 2 months 4-OHT E8.5 (n = 13), 2 months 4-OHT E10.5 (n = 13) were analyzed individually in 17 independent experiments. Error bars represent mean ± s.d. P-values as reported in figure, (one-way ANOVA followed by Tukey's multiple comparisons test).

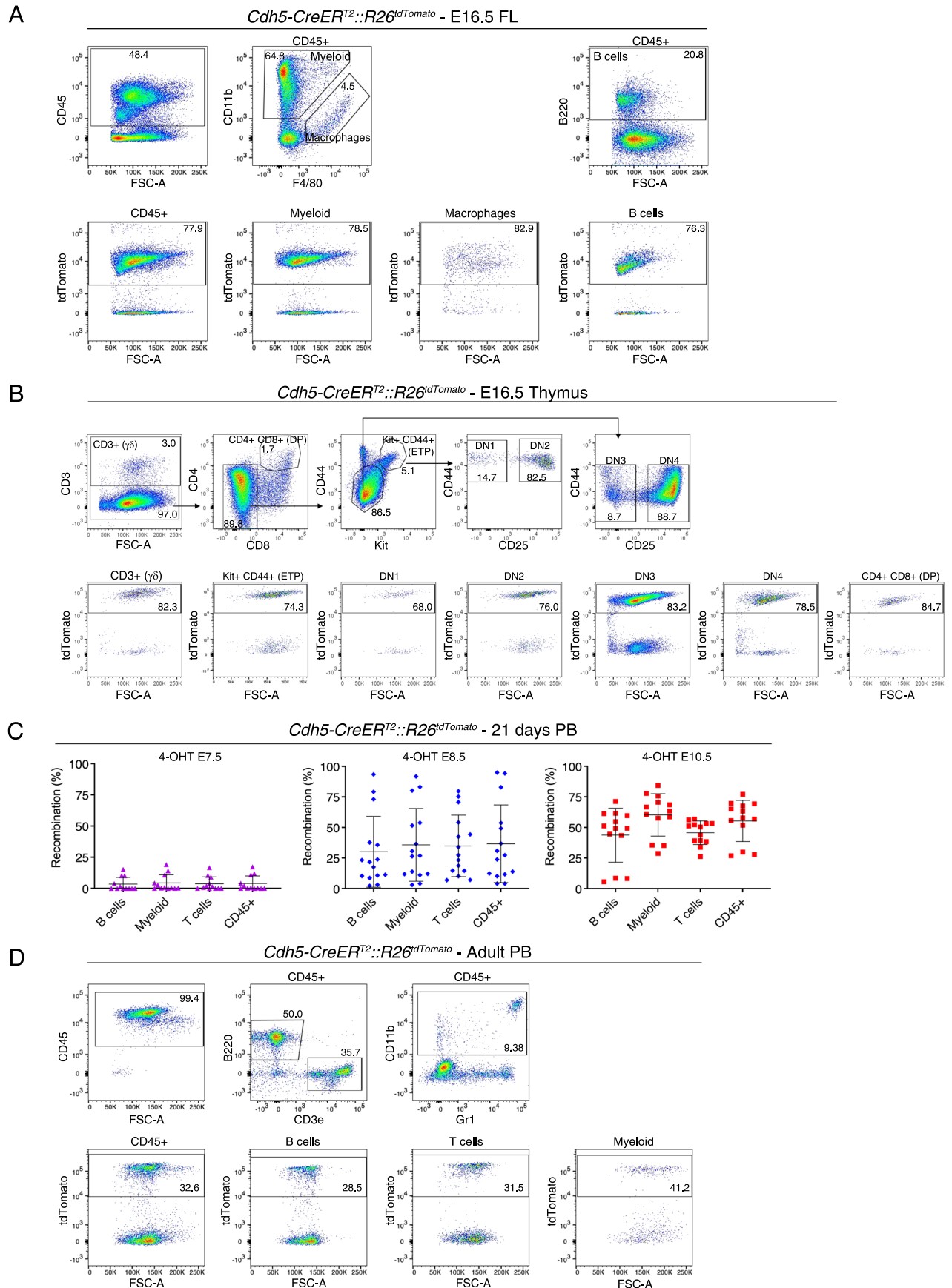

**Extended Data Fig. 3 | See next page for caption.**

**Extended Data Fig. 3 | Flow cytometric analysis of fetal and postnatal lympho-myeloid contribution in *Cdh5-CreER^T2* mice.** Related to Fig. 2. (**A**) Representative flow cytometric analysis of CD45⁺ (leukocytes), myeloid cells (CD45⁺ CD11b⁺), macrophages (CD45⁺ F4/80^hi), B cells (CD45⁺ B220⁺) in E16.5 *Cdh5-CreER^T2::R26^tdTomato* FL, labeled with 4-OHT at E7.5, E8.5 (shown here) or E10.5. 4-OHT E7.5 (n = 21), 4-OHT E8.5 (n = 23), 4-OHT E10.5 (n = 8) FL were analyzed individually in 5 independent experiments. The corresponding quantification is shown in Fig. 2a. (**B**) Representative flow cytometric analysis of thymocytes in E16.5 *Cdh5-CreER^T2::R26^tdTomato* or *Cdh5-CreER^T2::R26^zsGreen* fetal thymus, labeled with 4-OHT at E7.5, E8.5 (shown here) or E10.5. 4-OHT E7.5 (n = 13), 4-OHT E8.5 (n = 18), 4-OHT E10.5 (n = 16) thymuses were individually analyzed in 4

independent experiments. Quantification shown in Fig. 2b. (**C**) Quantification of flow cytometric analysis of B cells (CD45⁺ B220⁺), myeloid cells (CD45⁺ CD11b⁺), T cells (CD45⁺ CD3e⁺), CD45⁺ (leukocytes) in juvenile 21 days old *Cdh5-CreER^T2::R26^tdTomato* PB. 4-OHT E7.5 (n = 12), 4-OHT E8.5 (n = 15), and 4-OHT E10.5 (n = 13) mice were analyzed individually in 7 independent experiments. Error bars represent mean ± s.d. (**D**) Representative flow cytometric analysis of postnatal CD45⁺ (leukocytes), B cells (CD45⁺ B220⁺), T cells (CD45⁺ CD3e⁺), myeloid cells (CD45⁺ CD11b⁺) in *Cdh5-CreER^T2::R26^tdTomato* PB labeled with 4-OHT at E7.5, E8.5 (2-months-old adult mice; shown here) or E10.5. 4-OHT E7.5 (n = 12), 4-OHT E8.5 (n = 13), and 4-OHT E10.5 (n = 13) mice were analyzed individually in 7 independent experiments. The corresponding quantification is shown in Fig. 2c

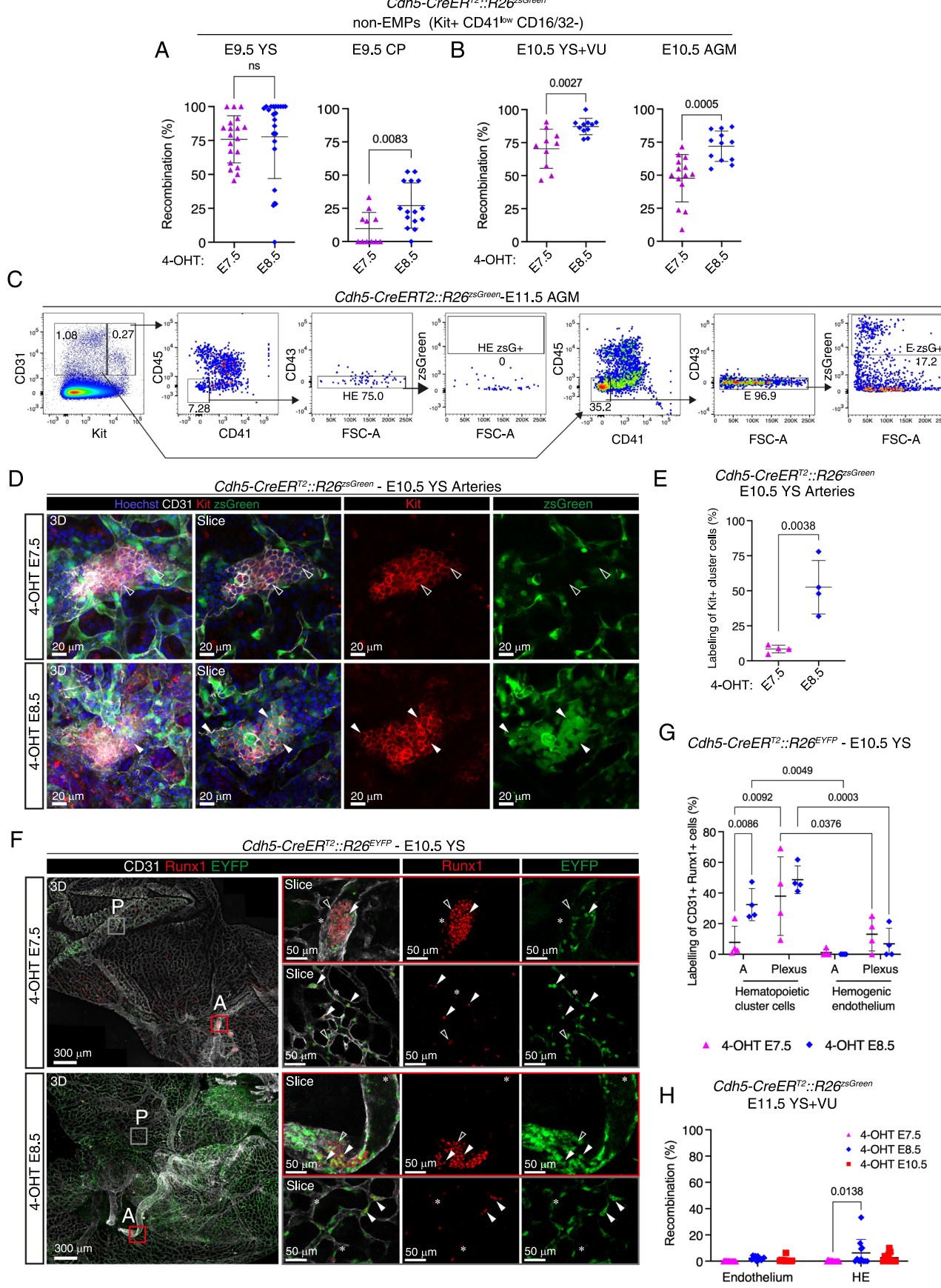

**Extended Data Fig. 4 | See next page for caption.**

**Extended Data Fig. 4 | Flow cytometric analysis and whole-mount confocal imaging of non-EMPs, endothelial cells and hematopoietic clusters in *Cdh5-CreER^{T2}* embryos and YS.** Related to Fig. 3. (**A**) Quantification of flow cytometric analysis of non-EMP (Ter119⁻Kit⁺ CD41^{low} CD16/32⁻) cells in *Cdh5-CreER^{T2}::R26^{zsGreen}* E9.5 YS (left) and caudal part (CP, right), labeled with 4-OHT at E7.5 or E8.5. Gates as shown in Extended Data Fig. 1B. 4-OHT E7.5 (n = 19), and 4-OHT E8.5 (n = 23) YS analyzed in 4 independent experiments. 4-OHT E7.5 (n = 11), and 4-OHT E8.5 (n = 15) CP analyzed in 3 independent experiments. Error bars represent mean ± s.d. ns = non-significant; P-values as reported in figure, (two-tailed unpaired Student's *t*-test). (**B**) Quantification of flow cytometric analysis of non-EMP (Ter119⁻ Kit⁺ CD41^{low} CD16/32⁻) cells in *Cdh5-CreER^{T2}::R26^{zsGreen}* E10.5 YS including vitelline and umbilical arteries (VU), left and AGM, right, labeled with 4-OHT E7.5 or E8.5. Gates as shown in Extended Data Fig. 1B. 4-OHT E7.5 (n = 10), and 4-OHT E8.5 (n = 12) YS analyzed in 4 independent experiments. 4-OHT E7.5 (n = 14), and 4-OHT E8.5 (n = 12) AGM analyzed in 4 independent experiments. Error bars represent mean ± s.d. P-values as reported in figure (two-tailed unpaired Student's *t*-test). (**C**) Representative flow cytometric analysis of AGM region hemogenic endothelium (HE, Ter119⁻ Kit⁺ CD31⁺ CD41⁻CD43⁻) and endothelium (Ter119⁻ Kit⁻ CD31⁺CD45⁻ CD41⁻ CD43⁻) in E11.5 *Cdh5-CreER^{T2}::R26^{zsGreen}* embryos, activated with 4-OHT at E10.5, related to Fig. 3g. (**D**) Confocal WM-IF analysis of E10.5 *Cdh5-CreER^{T2}::R26^{zsGreen}* YS large arteries. Left panels show 3D maximum intensity projections. Middle and right panels show single 2.5 μm-thick optical slices. Arrowheads indicate large Kit⁺ hematopoietic clusters, unlabeled with 4-OHT at E7.5 (empty arrowheads; top), or labeled at E8.5 (white arrowheads; bottom). 4-OHT E7.5 (n = 4), and 4-OHT E8.5 (n = 4) different YS were analyzed in 2 independent experiments. Scale bars: 20 μm. (**E**) Labeling quantification of Kit⁺ cluster cells in the YS large arteries as displayed

in (**D**). Measurements were performed on images from 4-OHT E7.5 (n = 4), and 4-OHT E8.5 (n = 4) different YSs (2-4 images / YS); 12 (4-OHT E7.5), 12 (4-OHT E8.5) different images used. Error bars represent mean ± s.d. P-value as reported in figure (two-tailed unpaired Student's *t*-test). (**F**) Confocal WM-IF analysis of E10.5 (32-36sp) *Cdh5-CreER^{T2}::R26^{EYFP}* YS. The left panels show maximum intensity 3D projections. Boxed area in the merged image is magnified in the right panels and shows a single 2.5 μm-thick optical slice. Color-code indicates magnified arterial (red box) and vascular plexus (grey box) regions. Arrowheads indicate Runx1⁺ CD31⁺ round-shaped hematopoietic cluster cells, labeled (white arrowheads) and unlabeled (empty arrowheads). Asterisks indicate Runx1⁺ CD31⁺ flat-shaped hemogenic endothelium cells. 4-OHT E7.5 (n = 4), and 4-OHT E8.5 (n = 4) different embryos were analyzed Scale bars: 300 μm (3D), 50 μm (slice). A: artery; P: plexus. (**G**) Labeling quantification of Runx1⁺ CD31⁺ round-shaped hematopoietic cluster cells and Runx1⁺ CD31⁺ flat-shaped hemogenic endothelium cells in E10.5 *Cdh5-CreER^{T2}::R26^{EYFP}* YS as shown in (**F**). Cells located in the arteries (A) cells were quantified separately from cells in the YS vascular plexus. Measurements were performed on images from 4-OHT E7.5 (n = 4), and 4-OHT E8.5 (n = 4) different embryos (3-6 images / YS); Error bars represent mean ± s.d. P-values as reported in figure (two-way ANOVA followed by Tukey's multiple comparisons test). (**H**) Quantification of flow cytometric analysis of labeled hemogenic endothelium (HE) (Ter119⁻CD31⁺ Kit⁺ CD45⁻ CD41⁻ CD43⁻) and endothelium (Ter119⁻ CD31⁺ Kit⁻ CD45⁻CD41⁻ CD43⁻) in YS + VU of E11.5 *Cdh5-CreER^{T2}::R26^{zsGreen}* embryos (gates as in (**C**)). 4-OHT at E7.5 (n = 15), 4-OHT at E8.5 (n = 11), 4-OHT at E10.5 (n = 12) AGM were analyzed individually in 5 independent experiments. Error bars represent mean ± s.d. P-value as reported in figure (two-way ANOVA followed by Tukey's multiple comparisons test).

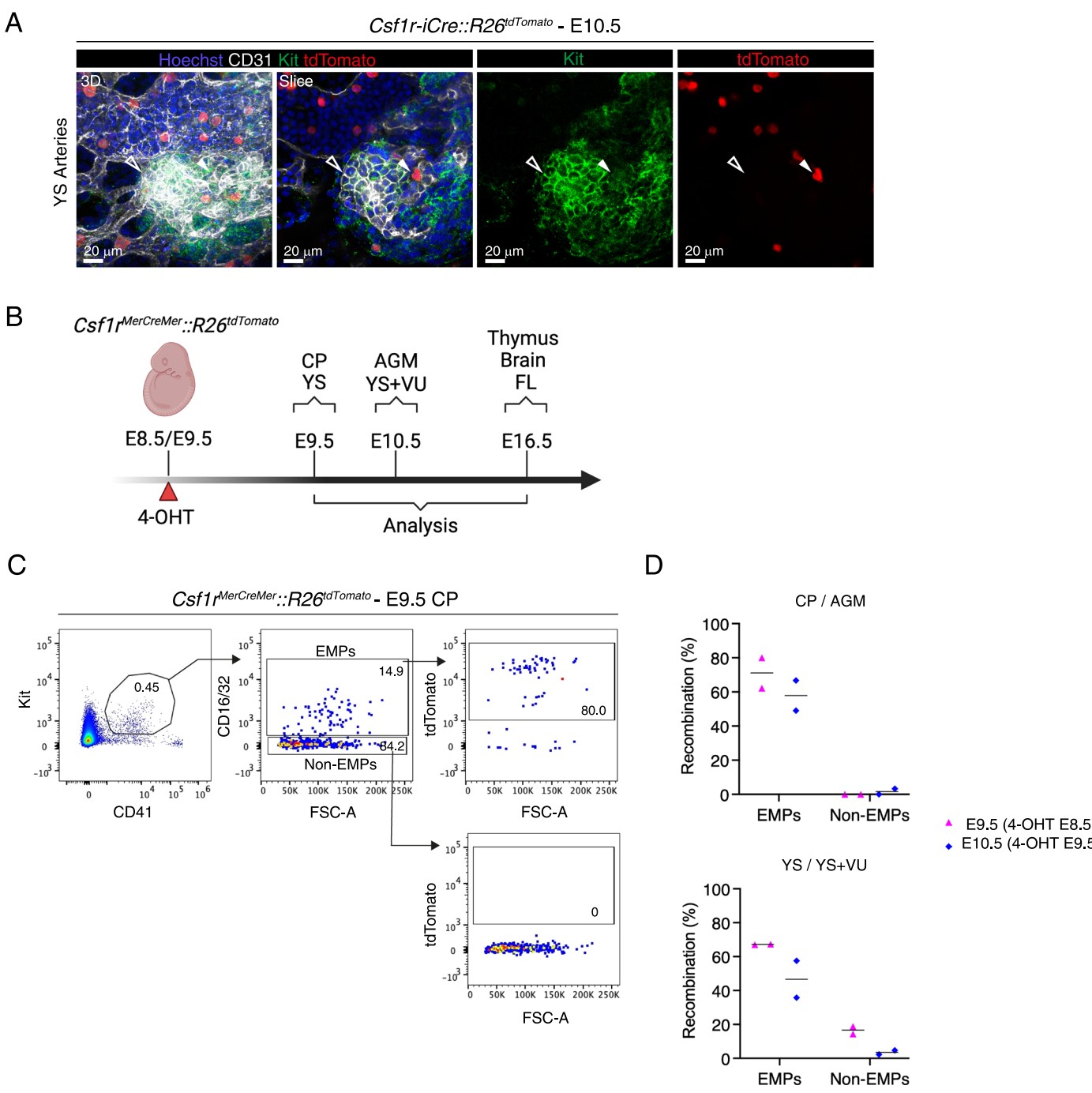

**Extended Data Fig. 5 | Csf1r lineage tracing yields highly specific labeling of immunophenotypically defined EMPs.** Related to Fig. 4. (**A**) Confocal WM-IF analysis of E10.5 *Csf1r-iCre::R26^tdTomato* YS large arteries. Left panel shows a 3D maximum intensity projection; other images are single 2.5 μm-thick optical slices. Arrowheads indicate Kit+ hematopoietic clusters. Empty arrowheads indicate unlabeled (tdTomato−) cells, white arrowheads indicate labeled (tdTomato+) ones. Quantification is shown in Fig. 4d. A total number of 5 different embryos were analyzed in 3 independent experiments.

(**B**) Visual schematic of lineage tracing experiments in *Csf1r^MerCreMer^::R26^tdTomato* E9.5-E10.5-E16.5 embryos, related to Fig. 4f and Extended Data Fig. 5c, d. (**C-D**) Representative flow cytometric analysis (**C**) and labelling quantification (**D**) of EMPs (Ter119− Kit+ CD41^low CD16/32+ ) and non-EMPs (Ter119−, Kit+ CD41^low CD16/32−) in *Csf1r^MerCreMer^::R26^tdTomato* E9.5 embryos, activated with 4-OHT at E8.5. E9.5 embryos 4-OHT E8.5 (n = 2), and E10.5 embryos 4-OHT E8.5 (n = 2) were analyzed in 2 independent experiments. Line indicates mean. Illustration in **b** created in BioRender; Brunelli, S. https://biorender.com/qj9wrbv (2026).

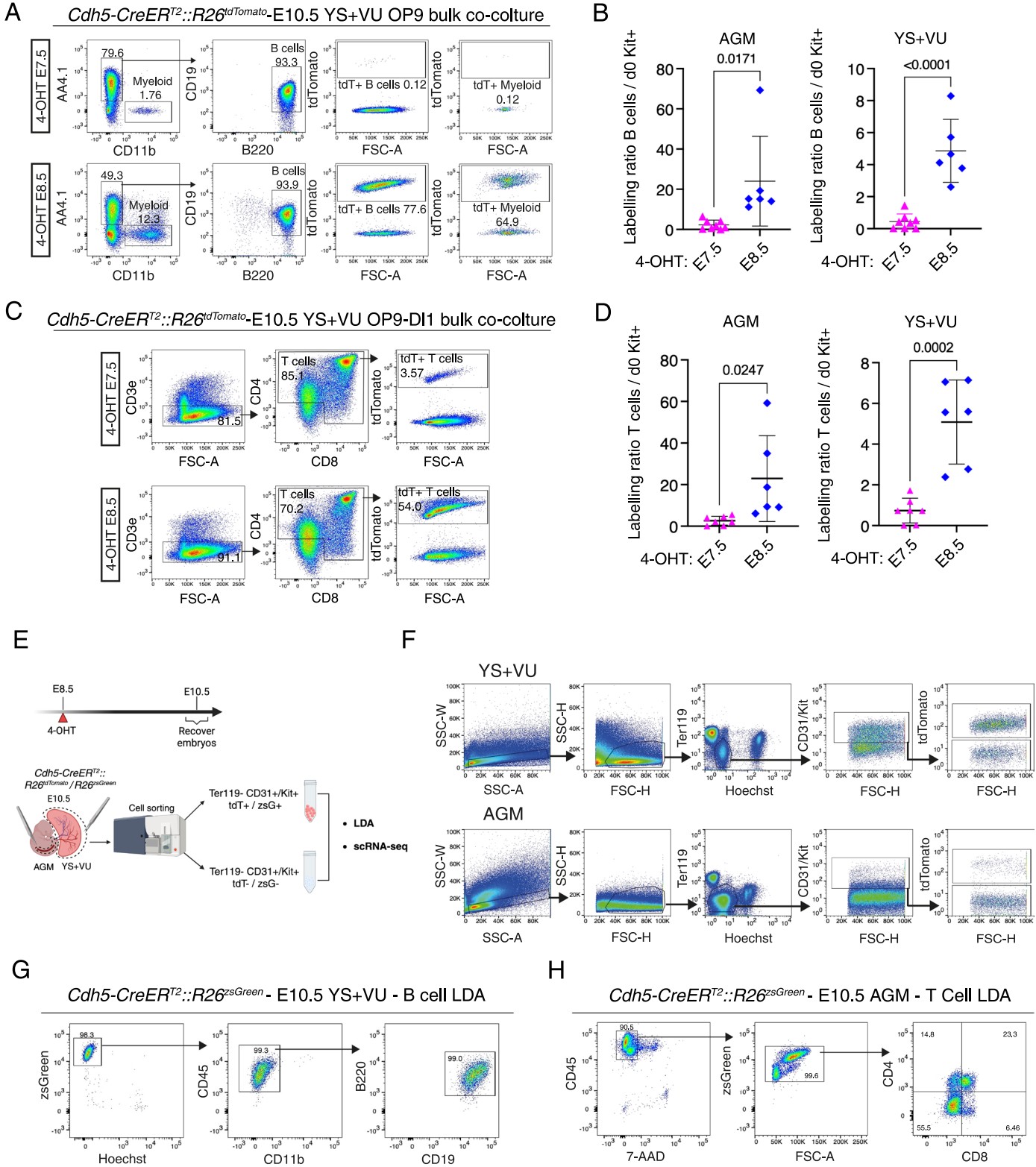

**Extended Data Fig. 6 | See next page for caption.**

**Extended Data Fig. 6 | Hemato-endothelial cell sorting strategy and ex vivo assessment of the lympho-myeloid potential of fetal-restricted HSPCs through bulk and limiting dilution co-culture assays.** Related to Fig. 5. (**A-B**) OP9 (B cell) bulk co-culture assays. A representative flow cytometric analysis is shown in (A). Values in (B) represent the frequency of labeled B (CD45$^+$ AA4.1$^+$ B220$^+$ CD19$^+$) cells at day (d)10 of co-culture, normalized to the percentage of labeled Kit$^+$ cells at d0, from E10.5 AGM and YS + VU activated with 4-OHT at E7.5 or E8.5. N = 8 (4-OHT E7.5) and n = 6 (4-OHT E8.5) different samples were analyzed across 4 independent experiments. Error bars represent mean ± s.d. P-values as indicated in figure (two-tailed unpaired Student's $t$-test). (**C-D**) OP9-Dl1 (T cell) bulk co-culture assays. A representative flow cytometric analysis is shown in (C). Values in (D) represent the frequency of labeled T cells at day (d)10 of co-culture normalized on the percentage of labeled Kit$^+$ cells at d0, from E10.5 AGM and YS + VU activated with 4-OHT at E7.5 or E8.5. N = 7 (4-OHT E7.5) and n = 6 (4-OHT E8.5) different samples were analyzed across 4 independent experiments. Each data point represents a biological replicate. Error bars represent mean ± s.d. P-values as indicated in figure (two-way ANOVA

followed by Tukey's multiple comparisons test). (**E**) Experimental schematic showing the cell sorting strategy used for scRNA-Seq and for Limiting Dilution Assays (LDA) of AGM and YS + VU from E10.5 *Cdh5-CreERT2::R26$^{tdTomato}$/R26$^{zsGreen}$* embryos, with 4-OHT activation at E8.5. Hemato-endothelial (Ter119$^-$ CD31$^+$ and/or Kit$^+$) cells were selected and tdTomato/zsGreen$^+$ or tdTomato/zsGreen$^-$ fractions were separately sorted from either AGM or YS + VU. (**F**) Representative flow cytometric gating strategy for the isolation of live Ter119$^-$ CD31$^+$ and/or Kit$^+$ tdTomato$^+$ and tdTomato$^-$ for scRNA-Seq or LDA as represented in (**E**). (**G**) Representative flow cytometric analysis of B cell LDA OP9 co-cultures from E10.5 *Cdh5-CreER$^{T2}$::R26$^{zsGreen}$* YS + VU with 4-OHT at E8.5. 3 independent experiments were performed. The corresponding quantification is shown in Fig. 5d. (**H**) Representative flow cytometric analysis of T cell LDA OP9-Dl1 co-cultures from E10.5 *Cdh5-CreER$^{T2}$::R26$^{zsGreen}$* AGM with 4-OHT at E8.5. 2 independent experiments were performed. The corresponding quantification is shown in Fig. 5e. Illustration in **e** created in BioRender; Brunelli, S. https://biorender.com/Sigowp4 (2026).

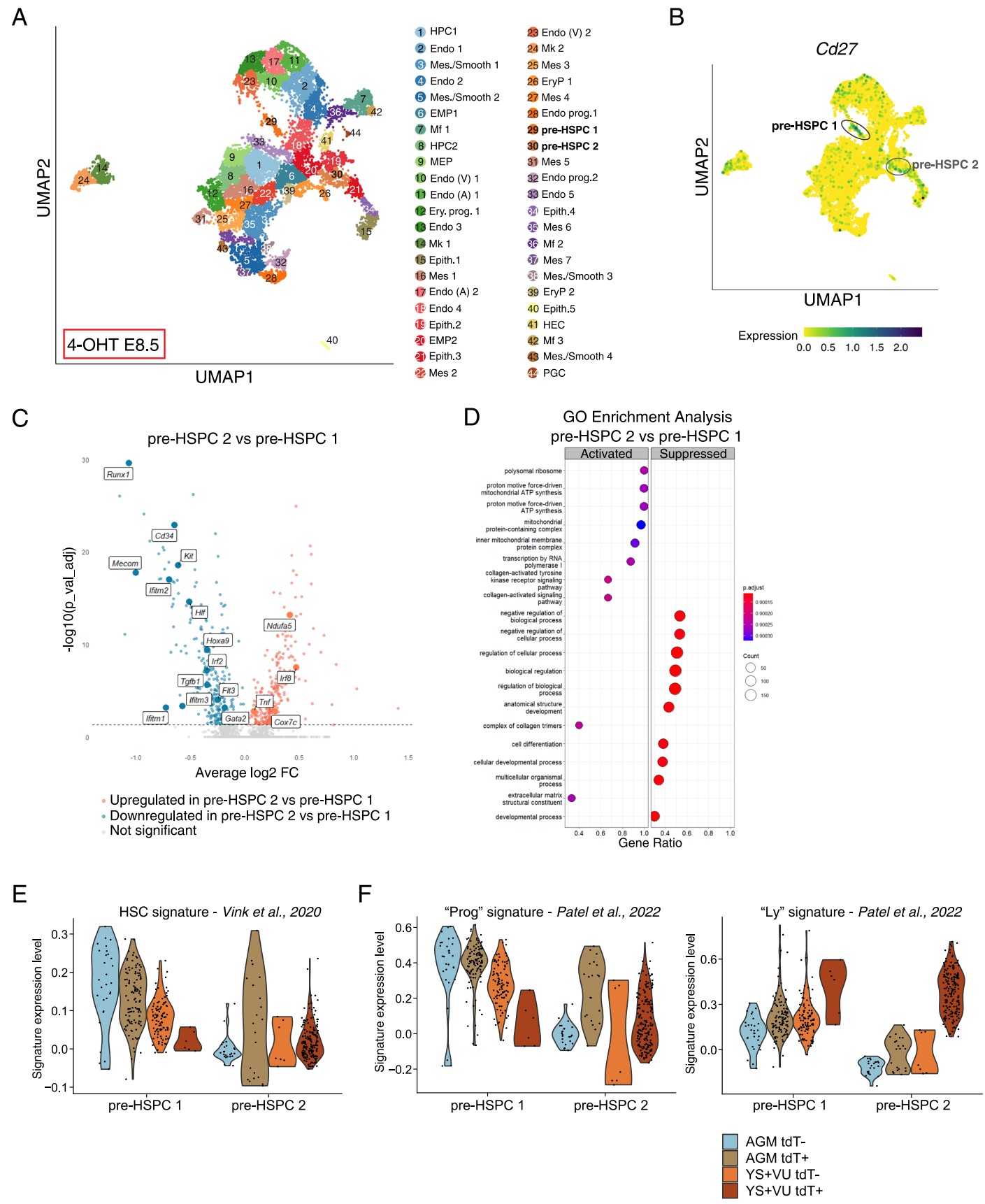

Extended Data Fig. 7 | See next page for caption.

**Extended Data Fig. 7 | scRNA-Seq of E10.5 AGM and YS + VU hemato-endothelial cells isolated from *Cdh5-CreER^T2::R26^tdTomato* embryos (4-OHT at E8.5) allows the identification and comparison of subsets of pre-HSPCs with distinct origins.** Related to Fig. 6. (**A**) UMAP plot showing complete clustering annotation of the scRNA-Seq dataset (cell isolation strategy as shown in Extended Data Fig. 6D, E). (**B**) UMAP plot showing *Cd27* expression. Pre-HSPC clusters 1 and 2 are circled. (**C**) Volcano plot (Average log2 FC versus negative log of adjusted P value) used to visualize statistically significant gene expression changes (adjusted P value < 0.05) between pre-HSPC 2 and pre-HSPC 1. Upregulated genes are labeled in red and downregulated genes labels in blue. Highlighted in labels are known genes related with hematopoiesis, inflammation and metabolic processes. A total number of 587 genes were differentially expressed (complete list in Supplementary Table 2). (**D**) GO biological process gene set enrichment analysis (GSEA) performed on a ranked list of differentially expressed genes between pre-HSPC 2 and pre-HSPC 1. Dot size indicates the gene count and color intensity represents enrichment score (adjusted P value). (**E-F**) Gene expression of published signatures for (**E**) HSC (Vink et al., *Cell Reports* 2020), (**F**) AGM-derived progenitors ("Prog") and lymphoid progenitors ("Ly") (Patel et al., *Nature* 2022) in pre-HSPC 1 and pre-HSPC 2 clusters, considering the tissue origin (AGM tdT⁻, AGM tdT⁺, YS + VU tdT⁻, YS + VU tdT⁺). The average expression of top 50 genes for each signature was calculated on single cells using the AddModuleScore function in Seurat.

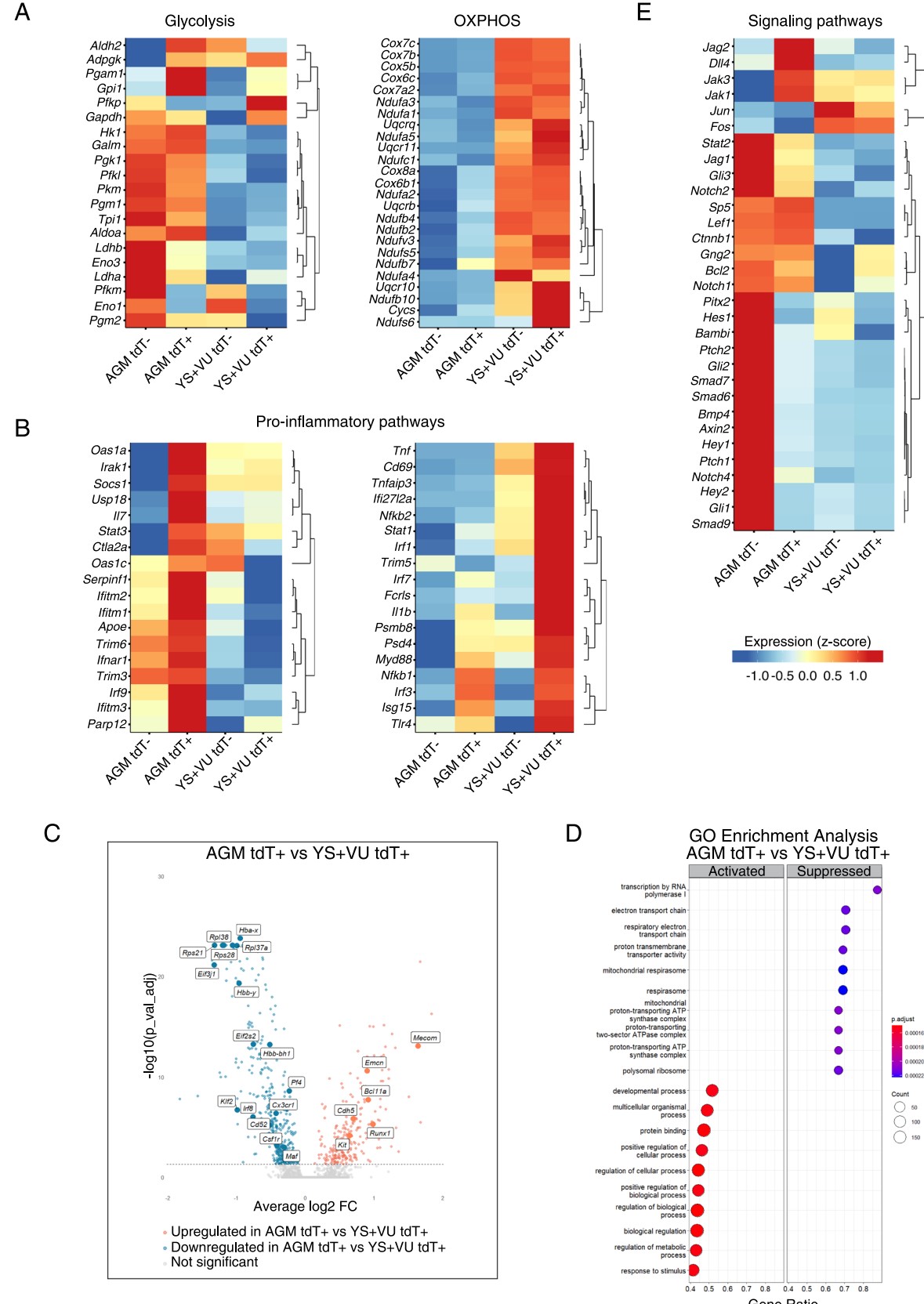

**Extended Data Fig. 8 | See next page for caption.**

**Extended Data Fig. 8 | Pre-HSPCs of distinct origin show different expression levels of genes involved in metabolism, inflammation and hematopoiesis.** Related to Fig. 6. (**A**) Heatmap showing the relative expression levels of selected genes involved in the metabolic pathways of glycolysis (left) and oxidative phosphorylation (OXPHOS, right) among AGM tdTomato⁻, AGM tdTomato⁺, YS + VU tdTomato⁻ and YS + VU tdTomato⁺ pre-HSPCs. (**B**) Heatmap showing the relative expression levels of selected genes involved in pro-inflammatory pathways among AGM tdTomato⁻, AGM tdTomato⁺, YS + VU tdTomato⁻ and YS + VU tdTomato⁺ pre-HSPCs. (**C**) Volcano plot (Average log2 FC versus negative log of adjusted P value) used to visualize statistically significant gene expression changes (adjusted P value < 0.05) between AGM tdTomato⁺ and YS + VU

tdTomato⁺ pre-HSPCs. Upregulated genes are labeled in red and downregulated genes labels in blue. Highlighted in labels are known hematopoietic and ribosomal genes. A total number of 698 genes were differentially expressed (complete list in Supplementary Table 3). (**D**) GO biological process gene set enrichment analysis (GSEA) performed on a ranked list of differentially expressed genes between AGM tdTomato⁺ and YS + VU tdTomato⁺ pre-HSPCs. Dot size indicates the gene count and color intensity represents enrichment score (adjusted P value). (**E**) Heatmap showing the relative expression levels of selected genes involved in developmental and hematopoietic processes (for example Notch, BMP, Shh signaling pathways) among AGM tdTomato⁻, AGM tdTomato⁺, YS + VU tdTomato⁻ and YS + VU tdTomato⁺ pre-HSPCs.

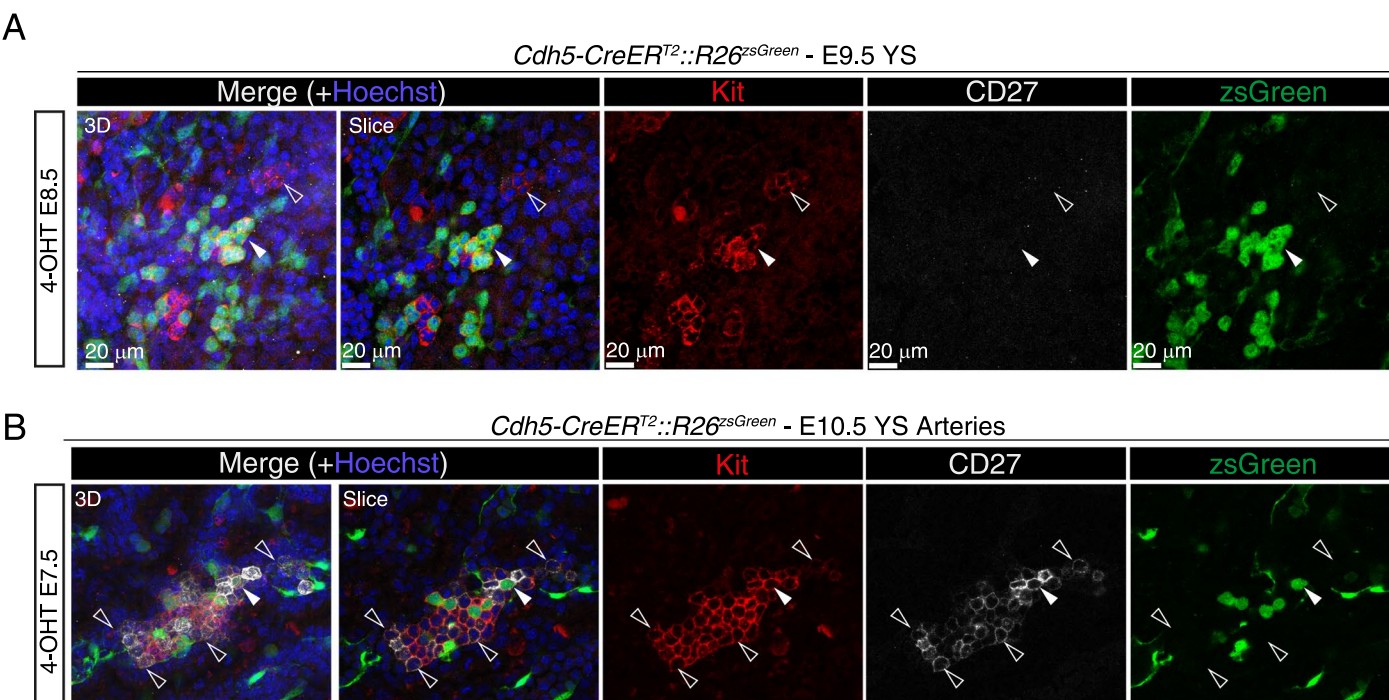

**Extended Data Fig. 9 | Whole-mount confocal imaging analysis of Kit⁺ CD27⁻ and Kit⁺ CD27⁺ clusters in the YS.** Related to Fig. 7. (**A**) Confocal WM-IF analysis of E9.5 *Cdh5-CreER^T2::R26^zsGreen* YS (4-OHT E8.5) showing lack of CD27 expression within Kit⁺ clusters. Left panel shows maximum intensity 3D projection. Middle and right panels show single 2.5 mm-thick slices. Arrowheads indicate Kit⁺ CD27⁻ clusters, labeled (zsGreen⁺; white arrowhead) or unlabeled (zsGreen⁻; empty arrowhead). A total number of 3 different YS were analyzed in 2 independent experiments. Scale bar: 20 μm. (**B**) Confocal WM-IF analysis of E10.5 *Cdh5-CreER^T2::R26^zsGreen* YS large arteries. Images show single 2.5 μm-thick optical slices. Arrowheads indicate examples of labeled (zsGreen⁺; white arrowheads) or unlabeled (zsGreen⁻; empty arrowheads) Kit⁺ CD27⁺ cells. 4-OHT E7.5 (n = 4) and 4-OHT E8.5 (n = 4) different YS were analyzed in 3 independent experiments. Scale bars: 20 μm.

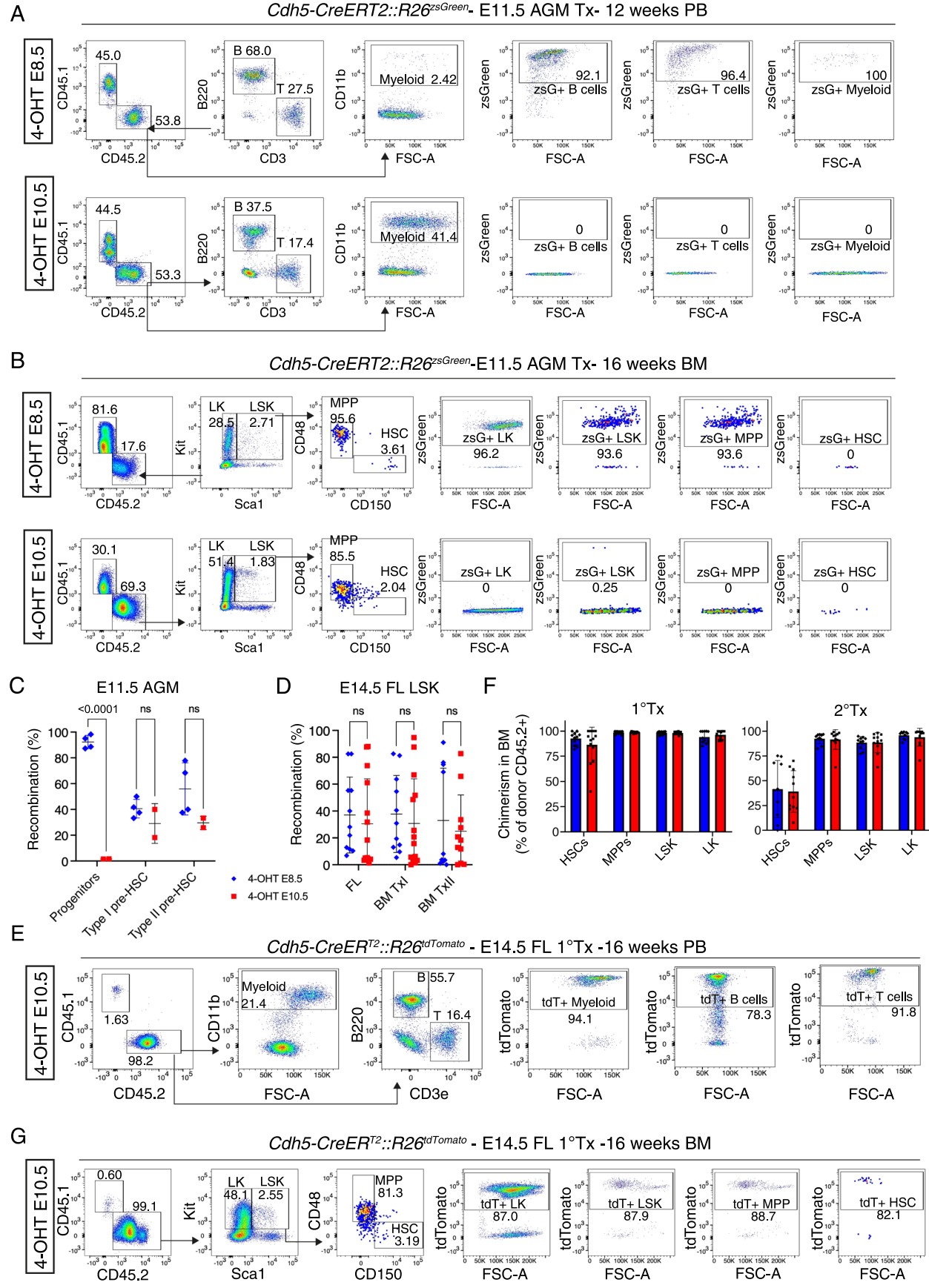

**Extended Data Fig. 10 | See next page for caption.**

**Extended Data Fig. 10 | Flow cytometric analysis of E11.5 AGM and E14.5 FL transplants.** Related to Fig. 8. (**A**) Representative flow cytometric analysis of PB from adult C57 BL/6 CD45.1 mice transplanted with *Cdh5-CreER^T2^::R26^zsGreen^* E11.5 AGM, activated with 4-OHT at E8.5 (top) or 4-OHT at E10.5 (bottom). FACS plots show hematopoietic populations and labeling percentages of donor myeloid cells, B cells and T cells. The corresponding quantification is shown in Fig. 8b, c. (**B**) Representative flow cytometric analysis of BM from adult C57 BL/6 CD45.1 mice transplanted with *Cdh5-CreER^T2^::R26^zsGreen^* E11.5 AGM, activated with 4-OHT at E8.5 (top) or 4-OHT at E10.5 (bottom). FACS plots show hematopoietic progenitor populations and labeling percentages of donor LKs, LSK, MPPs and HSCs. The corresponding quantification is shown in Fig. 8d. (**C**) Quantification of flow cytometric analysis of immunophenotypic pre-HSC labeling in *Cdh5-CreER^T2^::R26^zsGreen^ and Cdh5-CreER^T2^::R26^tdTomato^* E11.5 transplanted AGM (related to Fig. 8a–d), labeled with 4-OHT at E8.5 or E10.5. Four (4-OHT E8.5) and two (4-OHT E10.5) mice were analyzed in four independent transplant experiments. Error bars represent mean ± s.d. P-value as reported in figure, ns = non-significant (two-way ANOVA followed by Tukey's multiple comparisons test). (**D**) Quantification of flow cytometric analysis of LSK cells labeling in *Cdh5-CreER^T2^::R26^EYFP^ and Cdh5-CreER^T2^::R26^tdTomato^* E14.5 transplanted FL and bone marrow (BM) of primary and secondary transplanted mice labeled with 4-OHT at E8.5 or E10.5. N = 11

(4-OHT E8.5 1° Tx), n = 13 (4-OHT E10.5 1° Tx), n = 10 (4-OHT E8.5 2° Tx) and n = 11 (4-OHT E10.5 2° Tx) recipient mice were analyzed in 2 independent experiments. Error bars represent mean ± s.d. ns = non-significant (two-way ANOVA followed by Tukey's multiple comparisons test). (**E**) Representative flow cytometric analysis of PB from adult C57 BL/6 CD45.1 mice transplanted with *Cdh5-CreER^T2^::R26^tdTomato^* E14.5 FL, activated with 4-OHT at E8.5 or 4-OHT at E10.5 (shown here). FACS plots show hematopoietic populations and labeling percentages of donor myeloid cells, B cells and T cells. The corresponding quantification is shown in Fig. 8e. (**F**) Percentage (%) of chimerism (% of donor CD45.2⁺) in the BM HSCs, MPPs, LSK and LK of adult lethally irradiated of mice transplanted with E14.5 *Cdh5-CreER^T2^::R26^EYFP^* or *Cdh5-CreER^T2^::R26^tdTomato^* FL cells activated with 4-OHT at E8.5 or E10.5 (1° Tx) or adult BM cells from 1° TX (2° TX). N=11 (4-OHT E8.5 1° Tx), n=13 (4-OHT E10.5 1° Tx), n=10 (4-OHT E8.5 2° Tx) and n=11 (4-OHT E10.5 2° Tx) recipient mice were analyzed in 2 independent experiments. Error bars represent mean ± s.d. (**G**) Representative flow cytometric analysis of BM from adult C57 BL/6 CD45.1 mice transplanted with *Cdh5-CreER^T2^::R26^tdTomato^* E14.5 FL, activated with 4-OHT at E8.5 or 4-OHT at E10.5 (shown here). FACS plots show hematopoietic progenitor populations and labeling percentages of donor LKs, LSK, MPPs and HSCs. The corresponding quantification is shown in Fig. 8f.

# Reporting Summary

## Statistics

For all statistical analyses, confirm that the following items are present in the figure legend, table legend, main text, or Methods section.

| n/a | Confirmed | |
|---|---|---|
| ☐ | ☒ | The exact sample size (*n*) for each experimental group/condition, given as a discrete number and unit of measurement |
| ☐ | ☒ | A statement on whether measurements were taken from distinct samples or whether the same sample was measured repeatedly |
| ☐ | ☒ | The statistical test(s) used AND whether they are one- or two-sided<br>*Only common tests should be described solely by name; describe more complex techniques in the Methods section.* |
| ☒ | ☐ | A description of all covariates tested |
| ☐ | ☒ | A description of any assumptions or corrections, such as tests of normality and adjustment for multiple comparisons |
| ☐ | ☒ | A full description of the statistical parameters including central tendency (e.g. means) or other basic estimates (e.g. regression coefficient) AND variation (e.g. standard deviation) or associated estimates of uncertainty (e.g. confidence intervals) |
| ☐ | ☒ | For null hypothesis testing, the test statistic (e.g. *F*, *t*, *r*) with confidence intervals, effect sizes, degrees of freedom and *P* value noted<br>*Give P values as exact values whenever suitable.* |
| ☒ | ☐ | For Bayesian analysis, information on the choice of priors and Markov chain Monte Carlo settings |
| ☒ | ☐ | For hierarchical and complex designs, identification of the appropriate level for tests and full reporting of outcomes |
| ☒ | ☐ | Estimates of effect sizes (e.g. Cohen's *d*, Pearson's *r*), indicating how they were calculated |

*Our web collection on statistics for biologists contains articles on many of the points above.*

## Software and code

Policy information about availability of computer code

| Data collection | Imaging Data:<br>Zeiss Zen software (Zen Black) version 2.3 SP1 FP3<br>Fiji/ImageJ Cell Counter tool (v.2.3.5-2.9.0)<br>Imaris (v 9.7.2)<br><br>Flow cytometry Data:<br>BD FACSDiva software (version 8.0.2)<br>Summit software version 6.3<br>BD FACS Chorus software 6.1.0. |
|---|---|
| Data analysis | Imaging Data analysis:<br>Imaris Viewer (v.10.2)<br>Imaris (v. 9.7.2)<br>Fiji/ImageJ (v.2.3.5-2.9.0)<br>Adobe Photoshop 2024<br><br>Flow Cytometry Data Analysis:<br>FlowJo (v. 10)<br><br>scRNA-Seq Data analysis: |

Seurat (v. 4.0)
SCTransform
R (R-3.2.3 – R-4.2.1)
Louvain
CellRanger (v 6.1)
Clustree (v. 0.4.3)
SingleR (v1.0.1)
ShinyCell (https://github.com/SGDDNB/ShinyCell)
ClusterProfiler (v. 4.8.3)
glmGamPoi  (https://github.com/const-ae/glmGamPoi)

Statistics:
GraphPad Prism (v. 10.2.1-10.6.1)
Microsoft Excel (v. 16.96 and earlier versions)

For manuscripts utilizing custom algorithms or software that are central to the research but not yet described in published literature, software must be made available to editors and reviewers. We strongly encourage code deposition in a community repository (e.g. GitHub). See the Nature Portfolio guidelines for submitting code & software for further information.

## Data

Policy information about availability of data

All manuscripts must include a data availability statement. This statement should provide the following information, where applicable:
- Accession codes, unique identifiers, or web links for publicly available datasets
- A description of any restrictions on data availability
- For clinical datasets or third party data, please ensure that the statement adheres to our policy

Raw scRNA-seq data of YS+VU and AGM from E10.5 Cdh5-CreERT2::R26tdTomato embryos (4-OHT at E8.5) are available at the NCBI Sequence Read Archive (SRA) data repository with the accession number BioProject ID: PRJNA898269 (sample accession numbers: SRR28006358, SRR28006359, SRR28006360, SRR28006361).

The following published datasets were used in this study:
GSE180357 (Patel et al., 2022)
GSE143637 (Vink et al., 2020)

## Research involving human participants, their data, or biological material

Policy information about studies with human participants or human data. See also policy information about sex, gender (identity/presentation), and sexual orientation and race, ethnicity and racism.

| Reporting on sex and gender | N/A |
| Reporting on race, ethnicity, or other socially relevant groupings | N/A |
| Population characteristics | N/A |
| Recruitment | N/A |
| Ethics oversight | N/A |

Note that full information on the approval of the study protocol must also be provided in the manuscript.

# Field-specific reporting

Please select the one below that is the best fit for your research. If you are not sure, read the appropriate sections before making your selection.

☒ Life sciences        ☐ Behavioural & social sciences        ☐ Ecological, evolutionary & environmental sciences

For a reference copy of the document with all sections, see nature.com/documents/nr-reporting-summary-flat.pdf

# Life sciences study design

All studies must disclose on these points even when the disclosure is negative.

| Sample size | For in vivo experiments, whenever possible the number of samples required to obtain statistically reliable data, while avoiding both an excess of animals used for each experiment and unnecessary repetitions, was calculated using the G*Power package (version 3.1.9.3) based on data in the literature, conducted using similar models. The number of samples for each group was set to highlight an effect size of 1.2 (with a power of at least 80%, error α = 0.05). In the remaining cases no statistical method was used to pre-determine sample size, but sample sizes are consistent with our previous studies and other studies in the field. |

| Data exclusions | No data were excluded from flow cytometry, imaging and CFU-C experiments.<br>For OP9 co-cultures, one experiment was excluded from final analysis due to a very low (near zero) initial percentage of Kit+ progenitors labelling, that was interpreted as inefficient 4OHT-dependent recombination, likely due to technical problems with I/P injection in one single mouse.<br>In E11.5 AGM/YS-VU transplantation experiments 4 mice from 4-OHT E8.5 AGM and 7 mice from 4-OHT E8.5 YS-VU were excluded from analysis, because the chimerism was under the predetermined treshold of 5% from the first PB analysis.<br>In E14.5 FL transplantation experiments (secondary transplant), one mouse from the 4OHT E8.5-eYFP group, and two mice from the 4OHT E10.5-tdTomato group were excluded from the analysis as they died between 3 weeks and 1 month post-transplant. The most likely cause cause of death in these mice was fight wounds.<br>For scRNA-Seq, cells with genes count < 300 and > 8000 and fraction of mitochondrial reads > 0.20 were excluded from downstream processing. |
|---|---|
| Replication | Numbers of biological replicates and independent experiments (different litters) are indicated in figure legends and each experiment was replicated at least once (N=2 independent biological replicates)<br>For flow cytometry and imaging experiments, embryos were analyzed individually and each individual embryo is considered as one biological replicate. For ex vivo CFU-C assays and bulk OP9 co-cultures, each biological replicate was further split into two technical replicates, then averaged in the analysis. For ex vivo LDA experiments several e.e. from a litter were pooled together (7 to 12 e.e.), and sorted cells were seeded at the doses of 10, 30, 100 or 200 cells per well, in a minimum of 8 and a maximum of 12 wells per condition. Each pool is considered a biological replicate. For E11.5 AGM/YS-VU transplantation experiments 2 e.e. were transplanted in each recipient, considered as a biological replicate. For E14.5 FL transplantation experiments, each recipient mouse was transplanted with cells from a different embryo and considered a biological replicate. |
| Randomization | No specific randomization method was used. Animals were allocated into experimental groups according to their genotype. |
| Blinding | No specific methods were used for blinding. The investigators were not blinded to group allocations during data collection and analysis. |

# Reporting for specific materials, systems and methods

We require information from authors about some types of materials, experimental systems and methods used in many studies. Here, indicate whether each material, system or method listed is relevant to your study. If you are not sure if a list item applies to your research, read the appropriate section before selecting a response.

## Materials & experimental systems

| n/a | Involved in the study |
|---|---|
| ☐ | ☒ Antibodies |
| ☐ | ☒ Eukaryotic cell lines |
| ☒ | ☐ Palaeontology and archaeology |
| ☐ | ☒ Animals and other organisms |
| ☒ | ☐ Clinical data |
| ☒ | ☐ Dual use research of concern |
| ☒ | ☐ Plants |

## Methods

| n/a | Involved in the study |
|---|---|
| ☒ | ☐ ChIP-seq |
| ☐ | ☒ Flow cytometry |
| ☒ | ☐ MRI-based neuroimaging |

# Antibodies

| Antibodies used | Antibodies used in this study are listed below (antibody name; clone name where applicable; Supplier name; Cat. number; RRID identifier; working dilution)<br>FLOW CYTOMETRY ANTIBODIES<br>Rat monoclonal anti-Ter119 APC-fire750 (clone TER-119) BioLegend Cat#116250; RRID: AB_2819833 1:200<br>Rat monoclonal anti-CD117 (c-Kit) FITC (clone 2B8) eBioscience Ref: 11-1171-85; RRID: AB_465187 1:100<br>Rat monoclonal anti-CD41 PE-Cy7 (clone eBioMWReg30) eBioscience Ref: 25-0411-82; RRID: AB_1234970 1:200<br>Rat monoclonal anti-CD16/32 APC (clone 93) BioLegend Cat#101326; RRID: AB_1953273 1:200<br>Rat monoclonal anti-CD150 PE-Cy7 (SLAM) (clone TC15-12F12.2) BioLegend Cat#115914; RRID: AB_439797 1:400<br>Armenian hamster monoclonal anti-CD48 APC (clone HM48-1) BioLegend Cat#103412; RRID: AB_571997 1:400<br>Rat monoclonal anti-Ly6A/E (Sca-1) Pacific Blue (clone E13-161.7) BioLegend Cat#122520; RRID: AB_2143237 1:100<br>Rat monoclonal anti-CD117 (c-kit) APC-Cy7 (clone 2B8) BioLegend Cat#105826; RRID: AB_1626278 1:600<br>Rat monoclonal anti-CD45 APC-eFluor 780 (clone 30-F11) eBioscience Cat#47-0451-82; RRID: AB_1548781 1:200<br>Rat monoclonal anti-CD117 (c-kit) PE-Cy7 (clone 2B8) BioLegend Cat#105814; RRID: AB_313223 1:100<br>Rat monoclonal anti-CD135 Brilliant Violet 421 (clone A2F10) BioLegend Cat#135313; RRID: AB_2562338 1:200<br>Rat monoclonal anti-CD127 (IL-7Ra) PE (clone A7R34) BioLegend Cat#135009; RRID: AB_1937252 1:200<br>Rat monoclonal anti-Ter119 PE-Cy5 (clone TER-119) BioLegend Cat#116210; RRID: AB_313711 1:400<br>Armenian hamster monoclonal anti-CD3e PE-Cy5 (clone 145-2C11) BioLegend Cat#100310; RRID: AB_312675 1:200<br>Rat monoclonal anti-F4/80 PE-Cy5 (clone BM8) BioLegend Cat#123112; RRID: AB_893482 1:200<br>Mouse monoclonal anti-NK1.1 PE-Cy5 (clone PK136) BioLegend Cat#108716; RRID: AB_493590 1:400<br>Rat monoclonal anti-Ly6G/Ly6C (Gr1) PE-Cy5 (clone RB6-8C5) BioLegend Cat#108410; RRID: AB_313375 1:400<br>Rat monoclonal anti-CD19 PE-Cy5 (clone 6D5) BioLegend Cat#115509; RRID: AB_313644 1:400<br>Rat monoclonal anti-CD45R/B220 PE-Cy5 (clone RA3-6B2) BioLegend Cat#103210; RRID: AB_312995 1:200<br>Rat monoclonal anti-CD45 PE (clone 30-F11) BioLegend Cat#103106; RRID: AB_312971 1:200<br>Rat monoclonal anti-CD45R/B220 APC (clone RA3-6B2) BioLegend Cat#103212; RRID: AB_312997 1:200 |
|---|---|

Rat monoclonal anti-CD45R/B220 APC-Cy7 (clone RA3-6B2) BioLegend Cat#103224; RRID: AB_313007 1:200
Rat monoclonal anti-CD11b PE-Cy7 (clone M1/70) BioLegend Cat#101216; RRID: AB_312799 1:200
Rat monoclonal anti-Ly6G/Ly6C (Gr1) APC (clone RB6-8C5) BioLegend Cat#108412; RRID: AB_313377 1:200
Mouse monoclonal anti-CD45.2 FITC (clone 104) BioLegend Cat#109806; RRID: AB_313443 1:400
Mouse monoclonal anti-CD45.1 PE (clone A20) BioLegend Cat#110708; RRID: AB_313497 1:200
Mouse monoclonal anti-CD45.1 BV786 (clone A20) BioLegend Cat#110743; RRID: AB_2563379 1:200
Rat monoclonal anti-CD93 (AA4.1) APC (clone AA4.1) BioLegend Cat#136510; RRID: AB_2275868 1:200
Rat monoclonal anti-F4/80 APC (clone BM8) BioLegend Cat#123116; RRID: AB_893481 1:200
Rat monoclonal anti-CD4 PE-Cy5 (clone RM4-5) BioLegend Cat#100514; RRID: AB_312717 1:200
Rat monoclonal anti-CD8a APC-Cy7 (clone 53-6.7) BioLegend Cat#100714; RRID: AB_312753 1:200
Rat monoclonal anti-CD25 PE-Cy7 (clone PC61) BioLegend Cat#102016; RRID: AB_312865 1:200
Rat monoclonal anti-CD44 APC (clone IM7) BioLegend Cat#103012; RRID: AB_312963 1:400
Mouse monoclonal anti-CD45.2 APC-Cy7 (clone 104) BioLegend Cat#109824; RRID: AB_830789 1:100
Rat monoclonal anti-CD31 APC (clone 390) BioLegend Cat#102410; RRID: AB_312905 1:200
Rat monoclonal anti-CD117 (c-kit) BV 786 (clone 2B8) BD Horizon Cat#564012; RRID: AB_2732005 1:200
Rat monoclonal anti-CD117 (c-kit) APC (clone 2B8) BioLegend Cat#105812; RRID: AB_313221 1:200
Rat monoclonal anti-CD117 (c-kit) PE (clone 2B8) BioLegend Cat#105807; RRID: AB_313216 1:600
Rat monoclonal anti-CD201 PE (clone 1560) BD Pharmingen Cat#566337; AB_2739694 1:100
Rat monoclonal anti-mouse CD16/CD32 antibody (Fc Block), clone 2.4G2 BD Biosciences Cat# 553142; RRID:AB_394657 1:500
Rat monoclonal Anti-Mouse CD44 BV510 (clone IM7) BD Biosciences Cat#563114; RRID: AB_2738011 1:400
Armenian Hamster monoclonal Anti-Mouse CD3e BUV395 (clone 145-2C11) BD Biosciences Cat#563565; RRID: AB_2738278 1:100
Rat monoclonal Anti-Mouse CD4 BV786 (clone RM4-5) BD Biosciences Cat#563727; RRID: AB_2728707 1:400
Rat monoclonal Anti-Mouse CD8a Pacific Blue (clone 53-6.7) BioLegend Cat#100725; RRID: AB_493425 1:300
Rat monoclonal Anti-Mouse CD43 Alexa Fluor 700 (clone S11) BioLegend Cat#143213; RRID: AB_2800660 1:200
Human recombinant monoclonal Anti-Mouse CD45 APC Vio770 (clone REA737) Miltenyi Cat#130-110-662; RRID: AB_2658231 1:200
Rat monoclonal Anti-Mouse CD25 APC (clone PC61) BD Biosciences Cat# 557192; RRID:AB_398623 1:200
Rat monoclonal Anti-Mouse CD4 Pacific Blue (clone RM4-5) BD Biosciences Cat# 558107; RRID:AB_397030 1:200
Rat monoclonal Anti-Mouse CD8a PE (clone 53-6.7) BD Biosciences Cat# 553033; RRID:AB_394571 1:200
Rat monoclonal Anti-Mouse CD45 BV605 (clone 30-F11) BioLegend Cat# 103139; RRID: AB_2562341 1:200
Rat monoclonal Anti-Mouse CD11b FITC (clone M1/70) BioLegend Cat#101206; RRID: AB_312789 1:200
Rat monoclonal Anti-Mouse F4/80 AF700 (clone BM8) BioLegend Cat#123130; RRID: AB_2293450 1:200
Rat monoclonal Anti-Mouse CD4 BV786 (clone RM4-5) eBioscience Cat# 417-0042-82; RRID: AB_2921053 1:200
Rat monoclonal Anti-Mouse CD8a PE-Cy7 (clone 53-6.7) eBioscience Cat# 25-0081-82; RRID: AB_469584 1:200
Rat monoclonal Anti-Mouse CD45R/B220 APC (clone RA3-6B2) BD Biosciences Cat#553092; RRID: AB_398531 1:200
Rat monoclonal Anti-Mouse F4/80 FITC (clone BM8) BioLegend Cat#123108; RRID: AB_893502 1:200
Rat monoclonal Anti-Mouse Ter119 APC-ef780 (clone TER119) eBioscience Cat# 47-5921-80; RRID: AB_1548786 1:200
Streptavidin BV785 BioLegend Cat#405233 1:200
Rat monoclonal Anti-Mouse Sca1 FITC (clone E13-161.7) BioLegend Cat# 122506; RRID: AB_756191 1:200
Rat monoclonal Anti-Mouse CD150 PE-Cy7 (clone mShad150) eBioscience Cat# 25-1502-82 RRID: AB_10805742 1:200
Armenian Hamster monoclonal Anti-Mouse CD48 APC-Cy7 (clone HM48-1) BD Biosciences Cat# 561242; RRID: AB_10644381 1:100
Rat monoclonal Anti-Mouse CD45R/B220 Biotin (clone RA3-6B2) BD Biosciences Cat# 553086; RRID: AB_394615 1:200
Armenian Hamster monoclonal Anti-Mouse CD3e Biotin (clone 145-2C11) BD Biosciences Cat#553060; RRID: AB_394593 1:200
Rat monoclonal Anti-Mouse F4/80 Biotin (clone BM8) eBioscience Cat# 13-4801-85; RRID: AB_466657 1:200
Rat monoclonal Anti-Mouse Gr-1 Biotin (clone RB6-8C5) BD Biosciences Cat# 553125; RRID: AB_394641 1:200
Mouse monoclonal Anti-Mouse Nk1.1 Biotin (clone PK136) Biolegend Cat# 108704; RRID: AB_313391 1:200
Rat monoclonal Anti-Mouse Csf1r Biotin (clone AFS98) BioLegend Cat# 135508; RRID: AB_2085223 1:200
Rat monoclonal Anti-Mouse CD4 Biotin (clone GK1.5) BD Biosciences Cat# 553728; RRID: AB_395012 1:200
Rat monoclonal Anti-Mouse CD8a Biotin (clone 53-6.7) eBioscience Cat# 13-0081-85; RRID: AB_466346 1:200
Armenian Hamster monoclonal CD11c Biotin (clone N418) BioLegend Cat# 117304; RRID: AB_313773 1:200
Rat monoclonal Anti-Mouse Ter119 Biotin (clone TER119) BD Biosciences Cat# 553672; RRID: AB_394985 1:200

IMMUNOFLUORESCENCE ANTIBODIES
Goat polyclonal anti-m/rCD31/PECAM1 R&D systems Cat#AF3628 1:100
Rabbit polyclonal anti-GFP Invitrogen Cat#A11122; RRID: AB_221569 1:300
Chicken polyclonal anti-GFP Invitrogen Cat#A10262; RRID: AB_2534023 1:500
Rat monoclonal anti-mouse CD117 (c-Kit) (clone 2B8) eBioscience Cat# 14-1171-82; RRID: AB_467433 1:200
Rabbit polyclonal anti-RFP Rockland Cat# 600-401-379; 1:500
Armenian hamster monoclonal anti-mouse CD27 (clone LG-7F9) eBioscience Cat#14-0271-82; RRID: AB_467183 1:100
Rabbit anti-mouse/rat/human Runx1, clone EPR3099 Abcam Cat# ab92336; RRID: AB_2049267 1:400
Donkey polyclonal anti-goat Alexa Fluor Plus 647 Invitrogen Cat#A32849; RRID: AB_2762840 1:500
Donkey polyclonal anti-rabbit Alexa Fluor Plus 488 Invitrogen Cat#A32790; RRID: AB_2762833 1:500
Donkey polyclonal anti-rabbit Alexa Fluor Plus 555 Invitrogen Cat#A32794; RRID: AB_2762834 1:500
Donkey polyclonal anti-rat Alexa Fluor 488 Invitrogen Cat#21208; RRID: AB_2535794 1:500
Donkey polyclonal anti-rat CF568 Biotium Cat#20092; RRID: AB_ 1:500
Donkey anti-chicken IGY AF488 Invitrogen Cat#A78948; RRID: AB_2921070 1:500
Goat polyclonal anti-armenian hamster DyLight 649 (clone Poly4055) BioLegend Cat#405505; RRID: AB_1575122 1:500

| Validation | All antibodies have been validated by the manufacturer for the respective specific applications, and each lot has been tested for performance. |
|---|---|

# Eukaryotic cell lines

Policy information about cell lines and Sex and Gender in Research

| Cell line source(s) | OP9: ATCC CRL-2749. OP9-Dl1 were generated in the lab of Juan Carlos Zúniga-Pflücker. |
|---|---|

| Authentication | No new cell lines were generated in this study. No lines were authenticated. |
| --- | --- |
| Mycoplasma contamination | OP9 and OP9-DI1 cells tested negative for mycoplasma contamination during routine checks. |
| Commonly misidentified lines<br>(See ICLAC register) | No commonly misidentified lines in the ICLAC registry were used for the present study. |

# Animals and other research organisms

Policy information about studies involving animals; ARRIVE guidelines recommended for reporting animal research, and Sex and Gender in Research

| Laboratory animals | The transgenic mouse lines used in the present study were the following (indicated are name of the mouse line, source and RRID and/or reference).<br>Cdh5-CreERT2 - Gift from R.Adams (Wang et al., 2010)<br>R26zsGreen - Gift from M.Iannacone - RRID:IMSR_JAX:007906<br>R26tdTomato - The Jackson Laboratory - RRID:IMSR_JAX:007909<br>R26EYFP - The Jackson Laboratory - RRID:IMSR_JAX:006148<br>Csf1r-iCre - (Deng et al., 2010) - RRID:IMSR_JAX:021024<br>Csf1rMerCreMer (Qian et al, 2011) RRID:IMSR_JAX:019098<br>B6.SJL-Ptprca Pepcb/BoyJ (B6 CD45.1) - Gift from L. Naldini - RRID:IMSR_JAX:002014<br><br>All transgenic mouse lines were maintained on a CD45.2 C57BL/6 genetic background, with the exception of females used for timed matings in order to generate adult mice with 4-OHT activation during embryogenesis, which were instead of C57BL/6/FVB mixed background (F1). Mouse age is indicated in the manuscript and was comprised between 3 and 12 weeks at the time of the start of the experiment. Mice were housed in individually ventilated cages or filter top cages with a 12h:12h light-dark cycle (350/450 lux) and unrestricted access to food and water in the animal facilities at the San Raffaele Scientific Institute, University of Oxford or at the University of Milan. Standardized housing conditions included 22 (+/-2) °C temperature and relative humidity of 55% (+/-5%). |
| --- | --- |
| Wild animals | This study does not involve wild animals. |
| Reporting on sex | Sex information was not considered in the study design, and was not collected in embryos analyzed in this study. No sex-based difference was observed in analysis of adult mice. Hence, findings of the present study apply to both sexes. |
| Field-collected samples | This study does not involve field collected samples. |
| Ethics oversight | All mouse experiments were performed in accordance with experimental protocols approved by San Raffaele Scientific Institute and University of Milan Institutional Animal Care and Use Committees (IACUC) and authorized by the Italian Ministry of Health (Authorization numbers n° 503/2019-PR, 753/2023-PR and 351/2022-PR). All procedures carried out at the University of Oxford were in compliance with United Kingdom Home Office regulations and the Oxford University Clinical Medicine Animal Welfare and Ethical Review Committee (PPL number: PP9552402). |

Note that full information on the approval of the study protocol must also be provided in the manuscript.

# Plants

| Seed stocks | N/A |
| --- | --- |
| Novel plant genotypes | N/A |
| Authentication | N/A |

# Flow Cytometry

## Plots

Confirm that:

☒ The axis labels state the marker and fluorochrome used (e.g. CD4-FITC).

☒ The axis scales are clearly visible. Include numbers along axes only for bottom left plot of group (a 'group' is an analysis of identical markers).

☒ All plots are contour plots with outliers or pseudocolor plots.

☒ A numerical value for number of cells or percentage (with statistics) is provided.

## Methodology

**Sample preparation**

Sample preparation is detailed in the Methods section. Briefly, single cell suspensions were obtained from embryonic tissues (yolk sac, embryo/caudal part, thymus, fetal liver) by incubating for 20min at 37°C in calcium/magnesium free PBS supplemented with FBS 10%, 50 U/ml penicillin, 50 mg/ml streptomycin, , EDTA 2mM and collagenase type I (Sigma) 0.12% (w/v), followed by mechanical dissociation by pipetting. Peripheral blood (PB) samples were collected by tail vein bleeding using a scalpel; bone marrow (BM) was obtained by flushing long bones using a syringe and filtered in 40 micron strainers. PB, BM and fetal liver samples were treated with the appropriate amount of RBC Lysis Buffer. For flow cytometry analysis and cell sorting, single cell suspensions were incubated with conjugated antibodies at 4°C in the dark for 15 minutes.

**Instrument**

Flow cytometry data acquisition was performed on a BD LSR Fortessa X-20. Cell sorting was performed using a MoFLO Astrios cell sorter (Beckman Coulter), BD FACSAria II or BD FACSDiscover S8

**Software**

Flow cytometry data was collected with either BD FACSDiva software (version 8.0.2), Summit version 6.3 (Beckman Coulter) or Chorus version 6.1.0 (BD). Flow cytometry data was analyzed using FlowJo software version 10 (BD).

**Cell population abundance**

Cell populations in bulk were sorted to >95% purity, determined by re-analysis of sorted fractions.

**Gating strategy**

Cells were identified by FSC-A and SSC-A parameters, followed by doublet exclusion based on the ratio of FSC-W vs FSC-H and SSC-A vs SSC-W. Dead cells were subsequently excluded based on Hoechst 33258 or 7-AAD incorporation. A representative gating strategy for all experiments is included in main and/or Extended Data figures.

Fig.1B: Cells were gated on Ter119-
Fig. 1E: Cells were gated on Lin- (B220, CD19, CD3e, F4/80, Gr1, Nk1.1, Ter119)
Fig. 5B: Cells were gated on singlets
Extended data Fig. 1B: Cells were gated on Ter119-
Extended data Fig. 1F: Cells were gated on CD45+
Extended data Fig.2A: Cells were gated on Ter119-
Extended data Fig.2C: Cells were gated on Lin- (3220, CD19, CD3e, F4/80, Gr1, Nk1.1, Ter119)
Extended data Fig. 3A:Cells were gated on Live (Hoechst-)
Extended data Fig. 3B: Cells were gated on Live (Hoechst-)
Extended data Fig 3D: Cells were gated on Live (Hoechst-)
Extended data Fig. 4C: Cells were gated on Ter119-
Extended data Fig. 5C: Cells were gated on Ter119-
Extended data Fig 6A: Cells were gated on CD45+
Extended data Fig. 6C: Cells were gated on Live (Hoechst-)
Extended data Fig.6G: Cells were gated on singlets
Extended data Fig.6H: Cells were gated on singlets
Extended data Fig. 10A and B: Cells were gated on Live (Hoechst-)
Extended data Fig. 10E and G: Cells were gated on Live (Hoechst-)

☒ Tick this box to confirm that a figure exemplifying the gating strategy is provided in the Supplementary Information.

