## [Peer Review File · Nature Cardiovascular Research]

Fetal-restricted hematopoietic progenitors arise from hemogenic endothelium in vitelline and umbilical arteries

Corresponding Author: Professor Emanuele Azzoni

Version 0:

Reviewer comments:

Reviewer #1

(Remarks to the Author)

In this study, Barone et al combine inducible lineage fate-mapping with flow cytometry, whole mount imaging, functional assays, and single cell transcriptomics to elucidate the temporal and anatomical emergence of distinct waves of progenitors and HSCs from HE during murine embryonic development. Their studies reveal the distinct anatomical origins, timing, and contributions of HE to fetal and adult hematopoiesis through serial waves of EMP, fetal-restricted HSPC, and adult-type (long-term) HSCs. Overall, the study is well written and carefully executed, contributing to a growing body of recent publications that have defined the complex, layered emergence of fetal and adult hematopoiesis and established a paradigm in which fetal multilineage progenitors and adult HSCs emerge from separate endothelial compartments during embryonic development. Overall, the study merits publication in Nature Cardiovascular Research. However, to improve the manuscript, clarity in some aspects of the experimental approach, interpretation, and data presentation are recommended, as suggested below.

1) In Figure 1, why do E11.5 pre/pro-HSCs label poorly after E10.5 4-OHT treatment (1C) when HSCs/LSK cells in FL/adult BM are labelled efficiently (1E-G)? Pre/pro-HSC populations were previously described by the Medvinsky group based on co-expression of EC marker VE-Cadherin. Why was this marker not used in the current immunophenotypic analysis of pre/pro-HSC? If pre/pro-HSCs are gated as VE-Cad⁺ at E11.5, does this increase the frequency of labelling?

2) Since labelling at E8.5 captures most of the putative “pre/pro-HSCs” in Fig 1C, is it possible that most of these cells represent differentiating progenitors (not pre/pre-HSC) that have lost VE-Cadherin expression subsequent to their labeling at E8.5? Indeed, minimal labelling by E10.5 4-OHT treatment at E11.5 suggests that perhaps the majority of the putative “pre/pro-HSC” at this stage had already downregulated VE-cadherin expression.

3) The authors should be cautious about overstating the conclusions based on labelling in the Cdh5-CreERT2 models at E7.5 vs E8.5 vs E10.5 as entirely separate waves of EHT, given there is likely overlap in emerging progenitors (egs EMP are labelled at E7.5 and E8.5 while early LMP are also labelled at E8.5, arguing labelling at E8.5 captures a mixed population of progenitors). For example, the conclusion that “EMPs do not significantly contribute to fetal lymphomyelopoiesis” (page 6), which is based on E7.5 labelling, seems to ignore potential contribution of later emerging EMP labelled at E8.5, which cannot entirely be distinguished from “fetal-restricted HSPC”. This section should be modified to acknowledge this limitation of the interpretation. Later experiments in the CSF1R-iCRE line likely better address this distinction in regards to EMP contribution. Did the authors look at hematopoietic contribution of cells labelled in this mouse model in the later fetal liver to exclude contribution of EMP to fetal lymphomyelopoiesis at these later stages (or alternatively, cite published studies in this regard using this reporter line)?

4) In Figure 5C, that CFU activity is largely restricted to labelled cells in the E9.5 CP and E10.5 AGM/VU seems inconsistent with data in Figure 3A-C showing kit⁺ cluster cells (which presumably contain CFU activity) are largely unlabeled or heterogeneously labelled by IF analysis at the same stage (Fig 3A-C). If this interpretation is incorrect, it may help present a clearer explanation of how differences in experimental design in 5C and 3A account for this discrepancy. If one were to sort unlabeled cells as kit⁺, would this increase the frequency of CFU in the unlabeled population (given that presumably, the majority of unlabeled cells are going to be non-hemato/endothelial stromal populations, diluting out any CFU activity in this subset when assayed per input cell number)?

5) For the B/T cell assay in Figure 5, it is not clear why this experiment was performed with unsorted populations, rather than sorting as was done for CFU assay. The authors should explain the rationale for this approach, and why a more quantitative analysis (egs limiting dilution of sorted, labelled vs unlabelled cells) was not performed to better quantify T/B cell progenitor frequency in the sorted populations.

6) Given the complexities of anatomical sites, stages, and immunophenotypes examined in the study, it can be difficult to follow at times the distinction between intraembryonic and extraembryonic sites of HSPCs described (egs, in Fig 3D refers to the VU vs DA, but within the AGM). A summary figure describing the anatomical sites, and the temporal localization of different waves of HE identified in the study (potentially with relevant immunophenotype and genetic labelling), would be helpful to summarize the overall findings of the study and guide the reader in reference to anatomical sites, stage, etc for various experiments.

Reviewer #4

(Remarks to the Author)

I was very pleased to review this manuscript by Barone et al.

Major findings of the manuscript include the identification of a transient developmental wave of hematopoiesis which contributes to fetal lympho-myelopoiesis and with limited contribution to adult hematopoiesis

This work builds up on other recent reports showing the presence of HSC-independent waves of hematopoiesis that contribute to mid-gestation and late-gestation and adult hematopoiesis (Ulloa et al 2021; Yokomizo et al., 2022, Ganuza et al., 2022; Patel et al., 2022). Remarkably, Barone et al were able to ascribe anatomically this wave to the c-Kit⁺ clusters present in umbilical and vitelline arteries and not in the dorsal aorta or yolk sac, following endothelial-to-hematopoietic transition (EHT).

Authors employed a lineage tracing approach based on the tamoxifen inducible Cdh5-CreERT line and combination with three fluorescent reporters (R26-tdTomato, R26-zsGreen, and R26YFP). Tamoxifen (TAM) administration at E7.5, E8.5 and E10.5 allows them to with a good degree of specificity label three waves of hematopoiesis: erythroid-myeloid progenitors (EMP) (when treating at E7.5), definitive hematopoiesis (following TAM-treatment at E10.5) and the transient wave with lympho-myeloid potential which seems mostly constrained to the fetal stage. Lineage tracing with the Csf1r-Cre line, served them to label EMPs and confirm the absence of contribution from the yolk sac to the c-Kit⁺ clusters in vitelline and umbilical arteries.

Lineage-tracing followed by transcriptional profiling at single cell level of labelled progenitors allowed to characterize the hematopoietic precursors present in these vitelline and umbilical clusters and identify CD27 as an “exclusive” marker of these clusters.

Finally, serial transplantation of fetal livers (FL) from E8.5 and E10.5 TAM-treated Cdh5-CreERT-R26TdTomato embryos showed that the precursors labelled at E8.5 contributed to the multilineage engraftment of primary recipients but not to HSPCs in the bone marrow and neither to any hematopoietic compartments in secondary recipients.

I consider that this elegant work could significantly advance our knowledge on fetal hematopoiesis. Still, I would like the authors to address the following points as there are some critical technical aspects that I don't understand.

1) Since the activation of Cdh5-Cre-ERT labels endothelium, I really struggle to understand why/how TAM treatment at E7.5 or E8.5 does not result in the later labelling of any other wave of hematopoiesis as Cdh5 is not specific for any specific subtype of endothelium as far as it is known. Thus, EHT from labelled endothelium would contribute to any other later wave. Indeed, authors find labelling of the definitive wave when they label at E10.5. Only if the TAM-labelling of the endothelium happened after the completion of the EHT, one would expect no labelling of hematopoietic derived cells. Can authors show if a specific subfraction of the endothelium is labelled at each time point? Otherwise, where does the specificity on labelling each developmental wave come from? I really struggle here and I may be missing some details.

2-Also, I don't understand the reason on using the many different reporter lines (i.e. R26tdTomato, R26zsGreen and R26EYFP). In line 125 they indicate: “An advantage of this strategy is to use just one Cre mouse line, thus avoiding bias from cell type-specific promoters”. I don't understand the advantage of all these reporters. Can authors indicate if they found any difference among them? Is the efficiency of recombination of one allele different from other ones? Did any of the reporters led to different results which could affect conclusions? Like higher % of labelling of clusters in other locations? Can the authors detail which are the recombination efficiencies for each of the reporters? And justify why each reporter was employed in a particular experiment type if any specific reason.

3) Figure 6 and Figs S4-5.

-Can the authors elaborate on the reason behind pooling YS and VU cells following labelling at E7.5 and E8.5?

-Even though authors classified labelled and unlabelled clusters from different clusters by scRNAseq, it would be helpful if they could validate these results by performing simpler flow cytometry characterization of E9.5 and E10.5 Vitelline and Umbilical clusters from E8.5 TAM-treated embryos rather than a mix of YS+VU. So that they could employ markers for pro-HSC, Pre-type I and pre-type II.

4) Figure 7F-G. Chimerism from transplants should be provided in a more standard format to facilitate the interpretation of results. Knowing the % of total chimerism of any fraction would help to interpret the results. Currently, this is not clear

5) For strain Csf1r-iCRE, detailing some extra background and potential limitations as not 100% specificity, if that is the case should be included and discussed accordingly.

6) Even though authors, detected a good labelling specificity for different waves when labelling at E7.5, E8.5 and E10.5, this is not perfect and the interpretation of limitations and conclusions should be well-discussed.

Reviewer #5

(Remarks to the Author)

Barone et al. examine the origins of hematopoietic stem and progenitor cells (HSPCs) which contribute to the establishment of fetal-restricted lympho-myelopoiesis versus adult hematopoiesis. The authors' strategy was to activate reporter genes using a tamoxifen/4-OHT activated Cre driven by regulatory sequences from the *Cdh5* gene, which encodes vascular endothelial cadherin (VEC), at different times during development. They identify HSPCs labeled by injection of 4-OHT into pregnant dams at embryonic day 8.5 that significantly contributes to hematopoiesis in the fetal liver and developing thymus, and a later wave of HSPCs labeled at E10.5 that contribute more robustly to adult hematopoiesis. These data are consistent with several previous studies that have documented the presence of HSPCs emerging prior to adult repopulating HSCs in the embryo that contribute to fetal hematopoiesis, but minimally to adult hematopoiesis. The novelty of the Barone et al. study is that they claim to have defined the anatomic location of these fetal-restricted HSPCs as the vitelline and umbilical arteries by correlating the timing of their injections to when labeled hematopoietic clusters are present and also based on the location of CD27+ cells by histology.

Their data that early activation (E8.5) of a reporter gene results in more labeled HSPCs and differentiated blood cells in the fetus than activation at E10.5, and activation at E10.5 results in more cells that persist in the adult mouse are convincing. However, the weakness of the study is that the labeling of the different populations of HSPCs is not clean. Some of the data for labeling specific populations are very convincing – for example it appears that labeling endothelial cells at E10.5 completely misses the yolk sac EMP wave. However, it is not clear that later waves, i.e. LMPs, fetal-restricted HSCs, and adult HSCs can be cleanly separated using this approach, as they may all be differentiating from hemogenic endothelial cells throughout the E8.5 -E11.5 window (albeit in different proportions) which compromises the authors' ability to definitively localize these HSPCs anatomically. The authors should at least discuss the caveats to their approach and be mindful of not overinterpreting their data.

Another concern is that the authors used the term "pre-HSC" quite liberally without providing functional data to define these cells. A pre-HSC has historically been defined as a cell that cannot engraft adult mice directly but can be matured *ex vivo* into an adult-repopulating HSC. Molecular profiling of purified pre-HSCs by multiple groups have identified a consistent gene signature. The authors use the term pre-HSC more loosely, based on annotation in their single cell RNA-sequencing data which they don't directly compare to published data, and the presence of cell surface CD27 at E10.5. Moreover, they describe two distinct populations of pre-HSCs that have different molecular signatures, and based on CD27 expression localize one of these pre-HSC populations (pre-HSC 2) to the E10.5 vitelline vasculature. However, neither of the previous studies referenced by the reviewers demonstrate that CD27 marks pre-HSCs at E10.5. In fact, CD27 does not mark type I pre-HSCs at E11.5, so it is unlikely to mark type I pre-HSCs at 10.5 (all functional pre-HSCs at E10.5 are type I pre-HSCs). Unless the authors demonstrate that E10.5 pre-HSCs are CD27+, this marker cannot be used to identify them by histology which is how the authors localize them to the vitelline artery. In general, given that other progenitors are present in the embryo at the times the authors perform their analyses (e.g. LMPs and embryonic multipotent progenitors), it is not clear at all that what they call pre-HSC 2 are in fact pre-HSCs or one of these other progenitor types.

Specific comments:

1. Pre-HSCs in Fig 1A were characterized using CD41, CD43, and CD45. This is a relatively crude set of markers, and better ones, including Kit and CD201 along with endothelial markers have been described (see for example PMID: 27225119). The authors should refine their phenotypic pre-HSC analysis and demonstrate by *ex vivo* maturation and transplant that what they are calling pre-HSCs have pre-HSC activity.
2. Fig S1 H,I: The authors measure labeling of LMPs in the fetal liver at E11.5, but no analysis of LMPs was done prior to colonization of the fetal liver (CD45+ hematopoietic cluster cells at E10.5 by flow or confocal). Demonstrating labeling of LMPs should be shown in the tissues where they arise and have been defined (AGM and U+V arteries) to accurately represent the population.
3. Figure 2: The authors suggest that EMPs do not contribute significantly to fetal lympho-myelopoiesis based on the low contribution of labeled cells when 4-OHT is injected at E7.5. However, in Fig S1 C labeling of EMPs by E8.5 4-OHT treatment is quite high, which makes it difficult to separate the relative contribution of EMPs to FL lympho-myelopoiesis from the contribution of HSPCs in the vitelline arteries, which are also labeled by E8.5 4-OHT treatment.
4. Figure 2B. The authors show that activating Cre at E8.5 labels more T cells in the thymus than labeling at E10.5 and conclude that they have labeled two separate waves of thymus-settling progenitors. But what if it is one continuous wave, and labeling at E8.5 captures it in its entirety, whereas labeling at E10.5 only catches the latter part of the wave? How would their approach discriminate between those two possibilities?
5. Figure 3A,B. These data should be quantified.

6. Figure 4. The authors utilized the Csf1r-iCre line to label EMPs and show significant infiltration of E8.5 iCSF1r-labeled EMPs in the E10.5 fetal liver. Analysis of the E14.5 fetal liver and thymus with these animals could give a better sense of EMP contribution to fetal lympho-myeloopoiesis.

7. Fig 5 D,F: Plotting the outgrowth of lymphoid and myeloid cells as ratios from unsorted populations rather than sorting and plating labeled vs unlabeled cells from the AGM and VA makes interpretation of the data cumbersome. Limiting dilution plating of purified labeled and unlabeled cells would be more valuable and would allow for easier comparison of numbers of LMPs and their expansion in cultures.

8. Figure S4. These data are generated from a relatively impure population, and don't add much to the story. I would suggest not including these data in the manuscript.

9. Fig 7: The authors make claims about labeling different populations of pre-HSCs by injecting 4-OHT into pregnant dams at E8.5 and E10.5. Relatively convincing conclusions can be made about the contribution of HSPCs labeled by 4-OHT at E10.5 to fetal and adult hematopoiesis since HSPCs that form before E10.5 will not be labeled. However, administering 4-OHT at E8.5 may label endothelial cells that do not undergo EHT until E10.5, so overlapping waves of HSPCs may be labeled, making the system messy and difficult to interpret. The authors should quantify the percentage of labeled endothelial cells at E9.5 and E10.5 to determine the extent to which labeled hemogenic endothelial cells are present at these time points. Transplants of E11.5 AGMs should also be done to demonstrate the extent to which HSCs arising in the AGM region are labeled when 4-OHT is administered at E8.5.

10. Figure 7. The authors present engraftment data as "log10 frequency of donor labeled cells in PB (or BM) (normalized on % of donor labeled FL or BM LSK)". It is hard for the reader to wrap their head around what that means. There must be a simpler way to acquire or present these data.

Minor comments:

11. Line 80. Primitive yolk sac progenitors are not unipotent. They also produce megakaryocytes (PMID: 17062726)

12. Line 146. The authors state they examined type I and type II pre-HSCs in both AGM and YS (including VU). The vitelline artery is in both the yolk sac and attached to the dorsal aorta, so which dissected tissue are they saying contains it?

13. Line 194. The authors write "These data show that the hematopoietic wave that emerges from HE between E8.5 and E9.5 is the main contributor of fetal lympho-myeloopoiesis." Wouldn't it be more correct to say that they are measuring a wave between E8.5 and E10.5? To claim the wave occurs between E8.5 and E9.5 would imply that it is almost over between E9.5 and E10.5, but that experiment was not done.

Version 1:

Reviewer comments:

Reviewer #1

(Remarks to the Author)

The authors present substantial new data and updated interpretations that greatly strengthen the conclusions of the manuscript and address my prior concerns. The manuscript should be of broad interest to the field of developmental hematopoiesis.

Reviewer #4

(Remarks to the Author)

Authors have successfully addressed all my comments and provided new experimental evidences supporting their findings. I believe this constitutes a beautiful piece of work that will be of high interest to the field.

Reviewer #5

(Remarks to the Author)

In this revised manuscript, Barone et al. address concerns from the previous reviews with additional experiments. Some of these experiments are strong additions to the manuscript, particularly the limiting dilution T and B progenitor analyses, refinement of pre-HSC markers, comparison of scRNA-seq data to published datasets, and expansion of the whole mount confocal data. Some of the data, particularly the segregation of EMPs, LMPs, and HSCs with their labeling approaches are convincing. The inducible Csf1r-Cre labeling studies are a very helpful addition in this regard. I'm less convinced by the data claiming that fetal-restricted HSCs can be separated from adult HSCs by these labeling methods.

There are many interesting points in this paper, and the comprehensive analyses of labeling strategies will be useful to other investigators in the field. I think the authors overstate some of their conclusions though and have attempted to point these out

in the comments below. Other comments are suggestions to improve the clarity of the presentation.

Specific comments:

1. Lines 131-132. In the sentence “In contrast, tamoxifen metabolization into 4-OHT takes 6 to 12 hours in vivo and remains detectable in serum up to 48 hours post-administration”, is it the tamoxifen or 4-OHT that remains detectable in serum up to 48 hours post-administration”? I assume the authors are referring to tamoxifen, but it would be helpful to the reader if this is clarified.
2. Lines 141-142. In the sentence “We analyzed the labeling of LMPs (CD31+ Kit+ CD45+) in the E10.5 AGM and in YS including VU (YS+VU)”, are the authors referring to the VU within the yolk sac, the VU between the YS and the embryo proper, or the VU within the embryo proper? Please specify in this sentence in the main text, right where this is mentioned.
3. Lines 176-178. The authors write “Taken together, these data suggest that HE lineage tracing in the Cdh5-CreERT2 model (4-OHT at E8.5) can capture a putative population of fetal-restricted HSPCs. In contrast, 4-OHT pulses at E7.5 or E10.5 respectively label either EMPs or adult-type HSCs.”

I'm still struggling with the assertion that the authors are selectively labeling fetal-restricted HSCs at E8.5 and adult HSCs at E10.5 (although I am convinced they are labeling LMPs at E8.5). I still believe it is possible that all the HSCs they are labeling are adult HSCs maturing from pre-HSCs differentiating from hemogenic endothelial cells over the course of several days. Let's imagine that 20% of ECs that give rise to adult HSCs are efficiently labeled at E8.5, and 80% of ECs that give rise to adult HSCs are efficiently labeled at E10.5. In addition, let's imagine that it takes 4-6 days for labeled ECs to undergo EHT and then mature into phenotypic adult HSCs in the fetal liver. That scenario could explain the labeling dynamic of phenotypic HSCs in the fetal liver in Figures 2B-D (more relative contribution by E8.5-labeled ECs to HSCs at E14.5 and less at E16.5 compared to E10.5-labeled ECs). In this scenario, the smaller proportion of HSCs derived from ECs labeled at E8.5 have finished maturing into phenotypic HSCs by E14.5, while the larger bolus of HSCs derived from ECs labeled at 10.5 haven't finished maturing, and their true proportional contribution to fetal liver HSCs is only apparent at E16.5 and in adult animals. In fact, the authors show much later in the manuscript, in Figure 8, that no HSCs have yet matured in the E11.5 AGM region from ECs labeled at E10.5, while pre-HSC to HSC maturation has occurred from ECs labeled at E8.5, consistent with this scenario.

The authors do show, however, that HSPCs labeled at E8.5 can yield multi-lineage engraftment in primary recipients but not in secondary recipients, while E10.5 labeled HSCs can do both (Figure 8F). Therefore, there is some difference in the capacities of the HSCs derived from E8.5 versus E10.5 labeled ECs. However, I don't know whether the inability to self-renew in the context of a secondary transplant qualifies an HSC as “fetal-restricted”.

4. Line 194. The authors write “embryonic gamma/delta T cells were almost exclusively labeled by 4-OHT at E8.5 (Figure 2F). There appears to be a 4-fold difference in labeling efficiency at E8.5 versus E10.5. I think “almost exclusively” is a bit too strong a statement, and stating the fold difference would be more accurate.
5. Lines 198-199. The authors write “Post-natal analysis confirmed absence of peripheral blood (PB) labeling (<5%) in all lineages with activation at E7.5, both at 21 days (Figure S3C) and 2 months (Figure 2G; Figure S3D).” According to the figure legend, Figure S3D shows data from E8.5, not E7.5 labeling.
6. Lines 200-201. The authors write “4-OHT at E10.5 resulted in high levels of recombination in all lineages gradually increasing between 21 days and 2 months (Figure 2G; Figure S3C)”. Are those differences between 21 days and 2 months statistically significant? The averages don't look that different to me. Same question for the E8.5 labeling data – although the variation between mice indeed looks higher at 21 days than in adult mice, the averages don't look all that different.
7. Lines 294-295. The authors write “These data show that Csf1r+ progenitors, including EMPs (defined as CD41+ Kit+ CD16/32+) contribute in vivo to fetal lymphoid cells, confirming a previous report”. I agree the authors demonstrated that Csf1r+ progenitors contribute in vivo to fetal lymphoid cells, but what is their evidence that EMPs contribute to fetal lymphoid cells? They show that Csf1r-iCre also marks non-EMPs in the yolk sac and reference a previous report that yolk sac LMPs express Csf1r. It would be more accurate to delete the phrase “including EMPs (defined as CD41+ Kit+ CD16/32+)”.
8. I'm not sure what Figures 5D,E add to the story. Figure 5F,G contain much better data and are easier to interpret.
9. Lines 327-328. The authors write “The different frequencies of B and T lymphocyte progenitors suggests that developmental uncoupling may already take place in the tissues of emergence of these precursors.” Are these differences in B and T progenitor frequencies significant? Couldn't differences in the efficiency of the B and T cell assays be responsible for the different frequencies of T and B progenitors? This is strong statement without strong evidence to back it.
10. Lines 395-397. The authors write “Remarkably, all recipient mice repopulated with cells traced at E8.5, but none of those with cells traced at E10.5, displayed high levels of donor-derived labeling in myeloid and lymphoid cells of the PB (Figure 8B,C; Figure S10A,C).” But the bar graphs in 8B,C show 1 out of 2 mice transplanted with cells traced at E10.5 were multi-lineage engrafted in PB, and both mice had multi-lineage bone marrow engraftment, so the statement is confusing. The authors should make that statement when referring to Figure 8C and D, which presents the labeling frequency.

11. Lines 818-819. The authors write “4-OHT E8.5 (n=4) and 4-OHT E10.5 (n=2) AGM transplanted recipient mice were analyzed in 4 independent experiments.”. Were there 6 mice in each experiment, or a total of 6 mice were transplanted, and split over 4 independent experiments? If the latter, are 2 transplanted E10.5-labeled AGMs enough samples to reach a remarkable conclusion? It is a startling result, based on not very robust data if only two E10.5 AGM regions were analyzed.

12. Lines 395-397. The authors write “Remarkably, all recipient mice repopulated with cells traced at E8.5, but none of those with cells traced at E10.5, displayed high levels of donor-derived labeling in myeloid and lymphoid cells of the PB (Figure 8B,C; Figure S10A,C). Figure S10C shows pre-HSC labeling, not PB analysis of myeloid and lymphoid cells.

13. Figure 8E, F. I still find the engraftment data presented as Donor cell labeling in PB (or BM) normalized to input (log10) is hard to interpret.

Version 2:

Reviewer comments:

Reviewer #5

(Remarks to the Author)

The authors have done an excellent job in addressing my concerns. I congratulate them on this exciting work.

Dear Dr Azzoni, dear Emanuele,

Please accept my apologies for the delay in processing your work. As you know, we had to replace some referees who failed to provide a report, despite initially agreeing and without answering our queries.

Your manuscript "Homogenic endothelium of the vitelline and umbilical arteries is the major contributor to mouse fetal lympho-myelopoiesis" has now been seen by 3 referees, whose comments are appended below. Although they find your work of potential interest, they have raised substantive concerns which in our view are sufficiently important to preclude publication of the work in Nature Cardiovascular Research, at least in its present form.

In general, we agree with the reviewers that the work would be of interest to the broad audience of Nature Cardiovascular Research. At the same time, the review process identified several opportunities to strengthen the work. We would be happy to look at a revised manuscript that has been revised accordingly (unless something similar has been accepted at Nature Cardiovascular Research or appeared elsewhere in the meantime).

More specifically, it will be particularly important to the editors that you demonstrate that your labelling approach effectively differentiated between distinct endothelial populations. All reviewers highlighted this as a fundamental limitation of the manuscript, and we agree that this must be addressed.

Moreover, you will see that Reviewers #2 and #3 suggested several improvements to more robustly characterise and validate the haematopoietic populations you identified. We encourage you to follow their recommendations, as the manuscript would strongly benefit from additional experimental support for your claims.

Although the above are the most important to us, please respond to all points raised by our referees, ideally with new experiments. If you have any questions or would like to discuss the scope of the revision please do get in touch.

In the event of resubmission, please include a point-by-point response to the referees' comments (in a separate document to any cover letter). At the top of your point-by-point response to the referees' comments, please also include responses to the various editorial points highlighted above as guidance for your revision (this section should be labelled "Response to editors' comments"). To resubmit, please use the following link below:

*** This url links to your confidential home page and associated information about manuscripts you may have submitted or be reviewing for us. If you wish to forward this email to co-authors, please delete the link to your homepage first ***

I should stress, however, that we would be reluctant to trouble our referees again unless we thought their comments had been addressed in full. In the meantime we hope that you find our referees' comments helpful.

Best regards,

Dr. Andrea Tavosanis
Associate Editor
Nature Cardiovascular Research

Response to editors' comments

We thank the editorial team and the reviewers for the thorough and constructive evaluation of our manuscript, which has been very helpful and allowed us to substantially improve our manuscript. We have addressed the major points identified by the editors with new experiments which we included in our revised manuscript.

More specifically, to demonstrate that our labelling approach in *Cdh5-CreER^{T2}* mice can differentiate between distinct endothelial and hemogenic endothelial subpopulations (Editorial point #1), we have:

- Performed new whole-mount imaging experiments of *Cdh5-CreER^{T2}::R26^{EYFP}* embryos with a different marker panel (CD31, Runx1 and EYFP) which enabled us to identify and analyze labeling in hemogenic endothelium in addition to hematopoietic cluster cells (results shown in **Revised Figure 3** and **Figure S4**);
- Performed a new comprehensive flow cytometric analysis with an expanded panel of markers (Ter119, CD31, Kit, CD41, CD45, CD43, CD201 and zsGreen). This allowed us to more precisely quantify the extent of labelling in endothelial and hemogenic endothelial cells with the three activation time points (new data included in **Revised Figure 3** and **Figure S4**);

These new experiments, in addition to the data already included in the manuscript, provide evidence that the three 4-OHT activation modes that we used in this study effectively lead to the labeling of distinct subsets of endothelial and hemogenic endothelial cells, which in turn supports the specificity of capturing the consecutive hematopoietic waves. Indeed, neither 4-OHT activation at E7.5 or E8.5 in *Cdh5-CreER^{T2}* mice label (or label very few) endothelial and hemogenic endothelial cells in the ventral side of the dorsal aorta, providing a strong rationale for the lack of labeling of adult-type HSCs with these activation modes (new data included in **Revised Figure 3** and **Figure S4**). Moreover, it appears very unlikely that activation with 4-OHT at E7.5 labels fetal-restricted HSPCs, which we show to originate from hemogenic endothelium of the vitelline and umbilical arteries between E8.5 and E9.5, given that this activation mode: 1) labels a minority of endothelial cells in the vitelline and umbilical arteries and 2) yields a low to undetectable labeling of Kit⁺/CD27⁺ hematopoietic cluster cells located in these vascular structures. We also made the additional observation that 24-48 hours post-activation we found very few hemogenic endothelial cells labelled in both AGM and YS (as shown by both flow cytometry and whole-mount imaging). In the same tissues, hematopoietic cluster cells and phenotypic pre-HSPCs are labelled at a high frequency. This result is in line with the hypotheses that: i) endothelial-to-hematopoietic transition events

happen very quickly *in vivo*, and ii) virtually no untransitioned hemogenic endothelial cells are “left behind” (see also reply to reviewer 4, point 1 below).

To better characterize and validate, both immunophenotypically and functionally, the hematopoietic populations identified (Editorial point #2), we have:

- Performed a new comprehensive flow cytometric analysis with an expanded panel of markers (Ter119, CD31, Kit, CD41, CD45, CD43, CD201 and zsGreen), which allowed a more accurate quantification of the frequency of labelling in the different phenotypically identified pre-HSC and LMP subsets (new data included in **Revised Figure 1** and **Figure S2**)
- Performed new co-culture assays using sorted cells from both AGM and YS+VU with a limiting dilution approach. These experiments allowed us to assess and precisely quantify B and T lymphoid potential in labelled cell fractions (new data included in **Revised Figure 5** and **Figure S6**). In addition, we have improved our analysis of existing *ex vivo* culture data (now included in **Revised Figure 5** and **Figure S6**);
- Performed new direct transplantation experiments of *Cdh5-CreER^{T2}* E11.5 AGM and YS+VU traced at E8.5 (fetal restricted HSPCs) and E10.5 (HSCs). This new experiment enabled us to determine the engraftment potential of fetal-restricted HSPCs before fetal liver seeding (new data included in **Revised Figure 8** and **Figure S10**).

The most important pieces of information emerging from these new experiments were: i) non-pre-HSC CD201- hematopoietic progenitors in the E11.5 AGM and YS+VU were almost exclusively captured by pulse-labeling of hemogenic endothelium at E8.5, while type I and type II pre-HSC were partially labelled by both activations at E8.5 and E10.5; ii) the new LDA co-culture confirmed that lymphoid potential is virtually confined to E8.5-labelled hematovascular cells in the E10.5 embryo, and showed a higher frequency of T cell progenitors than B cell progenitors within E8.5-labelled cells in the *Cdh5-CreER^{T2}* E10.5 AGM and YS+VU (**Revised Figure 5F,G**), thus supporting the notion that T- and B lymphocyte-producing progenitors during development are functionally uncoupled; iii) adult repopulating potential in the E11.5 AGM was essentially confined to fetal-restricted HSPCs. The latter is perhaps the most striking result and provides support to the recently emerging concept that the fetal liver is an important site of pre-HSC maturation.

Additionally, to conclusively determine the extent of the fetal lympho-myeloid contribution of EMPs, which we could not precisely determine with the *Cdh5-CreER^{T2}* or *Csf1r-iCre* models, we have performed new lineage tracing analyses using a transgenic mouse line not previously included in the manuscript (*Csf1r^{MerCreMer::R26^{tdTomato}}*). Specifically, we have analyzed fetal livers and thymuses of E16.5 embryos activated with 4-OHT at E8.5 or E9.5. We found that in *Csf1r*+ progenitors, lymphoid potential appears only at E9.5. Lympho-myeloid contribution of *Csf1r*+ progenitors was limited in the FL but, surprisingly, more pronounced in the thymus (new data included in **Revised Figure 4** and **Figure S5**).

To more clearly reflect the central novelty and significance of our findings—namely, the precise developmental and anatomical origin of the wave of fetal-restricted hematopoietic stem/progenitor cells (HSPCs) identified in our study, we decided to change the manuscript title to “**Fetal-restricted hematopoietic progenitors arise from hemogenic endothelium in**

vitelline and umbilical arteries”. We hope that this new title will help conveying the main message of the study.

We believe the manuscript to be much stronger and complete in its current version. We are confident that our results, including the new ones added in the revised version, will be of particular interest to the community. We discuss in detail the specific comments to each of the points raised by the Reviewers in the point-by-point reply enclosed here.

Reviewers' expertise:

Reviewer #1: scRNA-Seq (haematopoietic cell profiling)

Reviewer #4: Haematopoietic lineage tracing in mouse

Reviewer #5: Haematopoietic development differentiation

Reviewer #1

In this study, Barone et al combine inducible lineage fate-mapping with flow cytometry, whole mount imaging, functional assays, and single cell transcriptomics to elucidate the temporal and anatomical emergence of distinct waves of progenitors and HSCs from HE during murine embryonic development. Their studies reveal the distinct anatomical origins, timing, and contributions of HE to fetal and adult hematopoiesis through serial waves of EMP, fetal-restricted HSPC, and adult-type (long-term) HSCs. Overall, the study is well written and carefully executed, contributing to a growing body of recent publications that have defined the complex, layered emergence of fetal and adult hematopoiesis and established a paradigm in which fetal multilineage progenitors and adult HSCs emerge from separate endothelial compartments during embryonic development. Overall, the study merits publication in Nature Cardiovascular Research. However, to improve the manuscript, clarity in some aspects of the experimental approach, interpretation, and data presentation are recommended, as suggested below.

We thank this Reviewer for the positive evaluation of our manuscript and for the insightful comments. We are confident that the revision process has addressed the issues raised by this and other Reviewers and has ultimately resulted in an improved manuscript.

1) *In Figure 1, why do E11.5 pre/pro-HSCs label poorly after E10.5 4-OHT treatment (1C) when HSCs/LSK cells in FL/adult BM are labelled efficiently (1E-G)? Pre/pro-HSC populations were previously described by the Medvinsky group based on co-expression of EC marker VE-Cadherin. Why was this marker not used in the current immunophenotypic analysis of pre/pro-HSC? If pre/pro-HSCs are gated as VE-Cad⁺ at E11.5, does this increase the frequency of labelling?*

We thank the Reviewer for this useful comment. The Reviewer was right that in the previous version of the manuscript what we called immunophenotypic “pre-HSC” consisted in fact of a heterogeneous population which included a significant fraction of hematopoietic progenitors other than pre-HSCs. To better define immunophenotypic type I and type II pre-

HSCs in the AGM and YS+VU of E11.5 embryos, we implemented a more refined flow cytometry panel with the use of additional surface markers. Based on the work of Zhou et al., Nature 2016 (PMID: 27225119), we performed a new analysis which included the markers Ter119, CD31, Kit, CD41, CD45, CD43, CD201 and zsGreen. Regarding the specific use of an anti-VE-Cadherin antibody, we decided against using this specific marker because in our experience it usually yielded a highly variable labeling, particularly dependent on experimental conditions. More specifically, our tissue dissociation protocol involves the use of enzymes such as collagenase, which can degrade the epitope recognized by commercially available anti-VE-Cadherin antibodies, thereby compromising the accurate detection of this marker. Therefore, we used the CD31 as a vascular marker instead.

In our new panel, immunophenotypically defined type I pre-HSCs were defined as: CD31+ Kit+ CD41^{low} CD45- CD43+ CD201+ and type II pre-HSCs as CD31+ Kit+ CD41^{low} CD45+ CD43+ CD201+. Hematopoietic progenitors other than pre-HSCs were defined as CD31+ Kit+ CD41^{low} CD45+/- CD43+ CD201-. As shown in the new data included in **Revised Figure 1** and described at lines 149-163, in E11.5 embryos both type I and type II pre-HSCs are now labelled with both E8.5 and E10.5 activations, while the E7.5 activation still labels a minority of these subsets. As correctly mentioned by the Reviewer, this is in agreement with the labeling of LSK and HSCs observed later on in FL with the same activation modes. We believe that the discrepancy of labelling in the previous version of our manuscript was essentially due to the fact that our previous pre-HSC definition identified a much less pure population and included many progenitors other than pre-HSC. Indeed, non-HSC hematopoietic progenitors defined as above in both E11.5 AGM and YS+VU exhibited a much more selective labelling pattern, and, strikingly, were almost exclusively labelled by the E8.5 activation in AGM and YS+VU (thus likely driving the result shown in the previous version of the manuscript). This was a particularly relevant result, especially in light of the new transplant experiments shown in **Revised Figure 8**, which showed that in the E11.5 AGM we could detect donor-derived labelled cells only when donor AGMs were traced with the E8.5 activation – and thus donor-derived engraftment potential is likely coming from the CD201- progenitor population in this specific tissue and developmental stage. See also response to point 2 of this Reviewer.

2) *Since labelling at E8.5 captures most of the putative “pre/pro-HSCs” in Fig 1C, is it possible that most of these cells represent differentiating progenitors (not pre/pre-HSC) that have lost VE-Cadherin expression subsequent to their labeling at E8.5? Indeed, minimal labelling by E10.5 4-OHT treatment at E11.5 suggests that perhaps the majority of the putative “pre/pro-HSC” at this stage had already downregulated VE-cadherin expression.*

We thank the Reviewer for this comment. The Reviewer was correct in his/her hypothesis. As also mentioned in the response to Point 1 of this Reviewer, based on the results that we obtained with the new flow cytometry panel (which are now shown in **Revised Figure 1**), in our previous analysis most of the phenotypic type I/type II pre-HSC that were found labelled with the E8.5 activation probably represented hematopoietic progenitors other than pre-HSCs. Our new results show that while Type I and Type II pre-HSC are partially labelled with both E8.5 and E10.5 activations, non-preHSC hematopoietic progenitors in both E11.5 AGM and YS+VU exhibit a much more selective labelling pattern, and, strikingly, are almost exclusively labelled by the E8.5 activation in AGM and YS+VU. The E10.5 4-OHT activation

labels very few of these progenitors. Since the non-preHSC subset constitutes a numerically much more abundant population than pre-HSCs in the E11.5 AGM and YS+VU, and given that our previous gating strategy could not discriminate between pre-HSCs and these progenitors, our previous result was likely driven by the labeling frequency that we can now more reliably attribute to the non-preHSC progenitor subset. For a detailed answer to this point including immunophenotypic definitions of each subset, please also refer to the reply to Point 1 of this Reviewer.

3) *The authors should be cautious about overstating the conclusions based on labelling in the Cdh5-CreERT2 models at E7.5 vs E8.5 vs E10.5 as entirely separate waves of EHT, given there is likely overlap in emerging progenitors (egs EMP are labelled at E7.5 and E8.5 while early LMP are also labelled at E8.5, arguing labelling at E8.5 captures a mixed population of progenitors). For example, the conclusion that “EMPs do not significantly contribute to fetal lympho-myelopoiesis” (page 6), which is based on E7.5 labelling, seems to ignore potential contribution of later emerging EMP labelled at E8.5, which cannot entirely be distinguished from “fetal-restricted HSPC”. This section should be modified to acknowledge this limitation of the interpretation. Later experiments in the CSF1R-iCRE line likely better address this distinction in regards to EMP contribution. Did the authors look at hematopoietic contribution of cells labelled in this mouse model in the later fetal liver to exclude contribution of EMP to fetal lympho-myelopoiesis at these later stages (or alternatively, cite published studies in this regard using this reporter line)?*

We thank the reviewer for this useful comment. It is true that 4-OHT at E8.5 in the *Cdh5-CreERT²* model labels YS EMPs (see **Revised Figure S1**) and therefore the contribution of the EMPs labelled between E8.5 and E9.5 cannot be discriminated from the contribution of fetal-restricted HSPCs labelled with the same activation mode. EMPs are also labelled with 4-OHT at E7.5 in the same model, but their later hematopoietic contribution is very limited (see **Revised Figure 2**). Our reasoning was that although YS EMPs were labelled with both E7.5 and E8.5 activations to near 100% efficiency, we observed a major contribution to fetal lympho-myelopoiesis only with the activation at E8.5; therefore, we deduced that this must have come from something other than EMPs.

The constitutive *Csfr-iCre* allele, in addition to labeling EMPs, traces the myeloid progeny of HSPCs. Furthermore, *Csf1r* is expressed in LMPPs and B cell progenitors in the FL (Boiers et al Cell Stem Cell 2013 PMID: 24054998; Zriwil et al Blood 2016 PMID: 27207794). Therefore, although this mouse model was important to demonstrate that hematopoietic clusters in the major arteries arise independently of EMPs (**Revised Figure 4A-E, Figure S5A**), it would have been difficult to discriminate an EMP vs HSPC origin at developmental stages later than E10.5 with the same model.

To conclusively determine the extent of the fetal lympho-myeloid contribution of EMPs, we have now performed new lineage tracing analyses using an inducible transgenic mouse line not previously included in the manuscript (*Csf1r^{MerCreMer}::R26^{tdTomato}*). Previous results using this approach showed that the late gestation (E16.5) lympho-myeloid contribution of EMPs labelled with 4-OHT at E9.25 was limited (~5%) (Atkins et al., J Exp Med 2022 PMID: 34928315). We have chosen both the E8.5 and E9.5 activation time points as both of these have been previously used to trace EMPs in the absence of HSC labeling (Gomez-Perdiguero

et al, Nature 2015 PMID: 25470051; Hoeffel et al Immunity 2015 PMID:25902481; Iturri et al Immunity 2021, PMID: 34062116, Soares-da-Silva et al., JEM 2021 PMID: 33566111; Atkins et al., J Exp Med 2022 PMID: 34928315; and others). Using this mouse line, we analyzed labelling frequencies of HSPCs and lympho-myeloid cells in the E16.5 fetal liver, in addition to brain microglia and TSP in the thymus at the same stage. The E8.5 activation yielded high labeling of microglia and 20-25% of FL macrophages, a labeling pattern essentially overlapping with the E7.5 activation in *Cdh5-CreER^{T2}* mice. In contrast, the E9.5 activation labelled around a subset of myeloid and B cells (<20%) and 30-40% T cells in the E16.5 FL and thymus, with the highest labelling detected in more differentiated thymocytes (CD4+ CD8+ DP, 50% labeling). Microglia were still labelled at a very high frequency with the E9.5 activation. LSK, including HSCs and MPPs were not labelled with either activation. Labelling specificity was also confirmed by flow cytometry analysis of E9.5-E10.5 caudal part/AGM and YS/YS+VU, which confirmed that, in *Csf1r^{MerCreMer::R26^{tdTomato}}* embryos, most of the hematopoietic progenitors labelled with E8.5 and E9.5 activations exhibited an immunophenotype consistent with EMPs. Still, we cannot completely exclude a contribution from LMPs which were shown to emerge in the YS vascular plexus starting from E9.5 and to express *Csf1r* (Boiers et al Cell Stem Cell 2013 PMID: 24054998; Luis et al Nature Immunology 2016 PMID: 27695000). Thus, in *Csf1r*+ hematopoietic progenitors, which largely comprise EMPs, lymphoid potential appears only at E9.5.

These data allowed us to draw the following conclusions:

- i) Comparison with the *Cdh5-CreER^{T2}* data suggests that in the E16.5 FL, the majority of lympho-myeloid cells originate independently from EMPs or *Csf1r*+ progenitors arising between E8.5 and E10.5. Thus, most lympho-myeloid cells in the E16.5 FL likely originate from fetal-restricted HSPCs;
- ii) The higher labeling frequencies observed in the thymus with the activation at E9.5 in *Csf1r^{MerCreMer}* mice emphasize the HSC-independent origin of the first TSPs and reinforce the notion that T- and B-myeloid-producing progenitors during development are functionally uncoupled (Kawamoto et al., 2000 PMID: 10795742; Ramond et al., 2014 PMID: 24317038; Berthault et al., 2017 PMID: 28825702). This point is further supported by our new LDA assay showing a higher frequency of T cell progenitors than B cell progenitors within E8.5-labelled cells in the *Cdh5-CreER^{T2}* E10.5 AGM and YS+VU (**Revised Figure 5F,G**).

These new data are included in **Revised Figure 4** and **Figure S5** (Results section lines 281-299). In addition, we have removed the statement “*EMPs do not significantly contribute to fetal lympho-myelopoiesis*” and we have updated the Discussion section to argue these points (lines 480-493). We trust that the Reviewer will agree that the new data and discussion have satisfactorily addressed the issue.

4) *In Figure 5C, that CFU activity is largely restricted to labelled cells in the E9.5 CP and E10.5 AGM/VU seems inconsistent with data in Figure 3A-C showing kit+ cluster cells (which presumably contain CFU activity) are largely unlabeled or heterogeneously labelled by IF analysis at the same stage (Fig 3A-C). If this interpretation is incorrect, it may help present a clearer explanation of how differences in experimental design in 5C and 3A account for this discrepancy. If one were to sort unlabeled cells as kit+, would this increase the frequency of CFU in the unlabeled population (given that presumably, the majority of unlabeled cells are*

going to be non-hemato/endothelial stromal populations, diluting out any CFU activity in this subset when assayed per input cell number)?

The vast majority of CFU-C activity (CFU-E, CFU-G/M/GM, and CFU-GEMM) in the E9.5 embryo is associated with Kit⁺ CD41^{low} CD16/32⁺ EMPs (McGrath et al., Cell Reports 2015. PMID: 26095363), which we find labelled at high frequency at both E7.5 and E8.5 activation time points as shown by immunofluorescence and flow cytometry (**Revised Figure S1**). Kit⁺ clusters located in the VA of CP at E9.5 and in the VU of the AGM, as well as the large arteries of the YS at E10.5, are more prominently labeled with activation at E8.5 than at E7.5 in the *Cdh5-CreER^{T2}* model and contain pre-HSPCs with lymphoid potential (**Revised Figure 5D-G**), as also supported by scRNA-seq data (**Revised Figure 6, Figure S7 and S8**) and CD27 protein expression, absent in EMP clusters (**Revised Figure 7 and Figure S9**). One important consideration to make here is that those VA clusters contain cells other than EMPs (**Revised Figure 4A,C and Figure S5**) and therefore would likely not generate CFU in these assays. Instead, the majority of EMPs are labelled at similar levels with E7.5 and E8.5 activations. We believe that this explains the fact that we have observed CFU-C activity to be largely restricted to labelled cells at both activations. Notably, in the E9.5 CP those CFU-Cs are very few and likely originating by cells circulating from the YS and not from the VA+UA clusters. Indeed, it was previously demonstrated that all CFU-C activity at E9.5 originates from the YS (Lux et al., Blood 2008; PMID: 17932251)

One difference in experimental design between the data reported in **Figure 3A** and **Figure 5C** that could also potentially account, at least in part, for the discrepancy observed by the reviewer is the different reporter line used in the two experiments. In **Figure 3A** we used *R26^{EYFP}*, while in **Figure 5C** we used *R26^{tdTomato}/R26^{zsGreen}*. The latter line has a higher recombination efficiency as previously reported (Patel et al., Nature 2022, PMID: 35705805). The reason for the usage of different reporter lines is explained below (see response to Point 2 of Reviewer #4) and mainly relates to technical reasons.

Nevertheless, to further clarify this point, we have now changed our analysis and normalized the CFU-C assay data by reporting the number of colonies relative to the number of Kit⁺ cells present in 1000 cells in both the positive and negative fractions. This allowed us to estimate the virtual number of colonies that would be obtained if we had assayed 1000 Ter119- Kit⁺ tdTomato/zsGreen⁺ or tdTomato/zsGreen⁻ cells. When normalization was performed as outlined above, the results and relative statistical significance did not change. This new analysis is now included in **Revised Figure 5C**.

5) *For the B/T cell assay in Figure 5, it is not clear why this experiment was performed with unsorted populations, rather than sorting as was done for CFU assay. The authors should explain the rationale for this approach, and why a more quantitative analysis (egs limiting dilution of sorted, labelled vs unlabelled cells) was not performed to better quantify T/B cell progenitor frequency in the sorted populations.*

We thank the reviewer for this comment. We previously performed OP9 co-cultures for B/T cell potential on sorted cells by using the same cell isolation strategy shown in **Figure 5B** of the manuscript. These experiments essentially yielded the same result as the data shown in **Figure 5D,E** (see **Figure 1** of this rebuttal); however, this experiment was only done on E10.5

YS+VU. We subsequently performed the experiment on unfractionated cells from both AGM and YS+VU (included in the revised manuscript) to maximize cell viability and culture efficiency. We reasoned that post-culture normalization of B/T labelling on the initial labelling of Kit+ cells would still allow us to draw meaningful conclusions. We were encouraged by the fact that the results on unsorted cells normalization were consistent with our previous experiments done with sorted YS+VU cells.

Figure 1. OP9 co-cultures for B and T cell potential from sorted cells.

OP9 (B cell) or OP9-DI1 (T cell) co-culture assays. Values shown represent absolute numbers of B and T cells obtained after culture, normalized on 1000 Live Ter119- labeled or unlabeled cells sorted from E10.5 *Cdh5-CreERT2::R26^{tdTomato/zsGreen}* YS+VU activated with 4-OHT at E7.5 (left) or E8.5 (right). Each data point results from the average of a technical duplicate. N = 6 (4-OHT E7.5 B cells), N = 3 (4-OHT E7.5 T cells), N = 3 (4-OHT E8.5 B cells), N = 3 (4-OHT E8.5 T cells) different samples were analyzed in 4 independent experiments. Error bars represent mean ± SD. *p<0.05 (one-tailed unpaired Student's t-test).

In our revised manuscript, to streamline and simplify the presentation of the results, we have now reanalysed the T cell bulk OP9 co-culture data, which was previously shown separately for each progenitor subpopulation (DN1, DN2, etc.). Since the results were very similar and consistent among subpopulations and therefore not very informative, we now show the post-culture labelling of T cells “as a whole”, similar to what we did for B cells. This new analysis is presented in **Revised Figure 5E**; new gating strategies for OP9 (B cell) and OP9-DI1 (T cell) co-culture analysis are shown in **Revised Figure S6A,B**. Myeloid cell analysis was moved to **Revised Figure S6C**.

We agree with the reviewer that the previous version of our manuscript was missing a quantitative approach. Therefore, to precisely establish the frequency of T and B cell progenitors in labelled versus unlabelled E10.5 hemato-vascular fractions (4-OHT at E8.5), we have performed new OP9/OP9-DI1 B/T co-culture assays using sorted cells from both AGM and YS+VU, this time by adopting a limiting dilution experimental setting. The cell isolation strategy for OP9 (B cell) and OP9-DI1 (T cell) LDA co-culture analysis is shown in **Revised Figure S6D,E** and it is the same that was also used for the isolation of hemato-vascular cells for scRNA-seq.

The results of the OP9 LDA analysis are presented in **Revised Figure 5F,G** (lines 320-324) and indicate that T cell progenitors are present in labelled hemato-vascular fractions at a 1/81

frequency in the AGM and slightly less frequent in the YS+VU (1/123), while they are absent or very rare in nonlabelled hemato-vascular fractions. A similar scenario was seen in B cell LDA, although with lower progenitor frequencies, indicating that the 4-OHT activation captures virtually the entirety of B and T cell progenitors in the E10.5 AGM and YS+VU. Of note, as mentioned, our new LDA assay showed a higher frequency of T cell progenitors than B cell progenitors within E8.5-labelled cells in the *Cdh5-CreER^{T2}* E10.5 AGM and YS+VU (**Revised Figure 5F,G**), thus supporting the notion that T- and B lymphocyte-producing progenitors during development are functionally uncoupled (Kawamoto et al., 2000 PMID: 10795742; Ramond et al., 2014 PMID: 24317038; Berthault et al., 2017 PMID: 28825702). The different frequencies of B and T lymphocyte progenitors further suggest that developmental uncoupling may already take place in the tissues of emergence of these precursors.

6) *Given the complexities of anatomical sites, stages, and immunophenotypes examined in the study, it can be difficult to follow at times the distinction between intraembryonic and extraembryonic sites of HSPCs described (egs, in Fig 3D refers to the VU vs DA, but within the AGM). A summary figure describing the anatomical sites, and the temporal localization of different waves of HE identified in the study (potentially with relevant immunophenotype and genetic labelling), would be helpful to summarize the overall findings of the study and guide the reader in reference to anatomical sites, stage, etc for various experiments.*

We thank the Reviewer for this comment. We agree that the distinction between intraembryonic and extraembryonic sites, especially when referring to vitelline and umbilical arteries, could be difficult to follow. We have clarified this distinction throughout the text in the revised manuscript (see description to Figure 3 and S4, lines 212-261). To avoid confusion and to help distinguishing anatomical sites, we have moved most of the YS immunofluorescence data – including new data- to the supplementary figures. For instance, **Revised Figures S4, S5, S9** show labeling in hematopoietic clusters within the YS arteries. When referring to the intraembryonic portion of the VU as part of the AGM, this is clearly indicated in the figures and the related text. The rationale for this mainly relates to the fact that the initial emergence of fetal-restricted HSPCs takes place at E9.5 in the intraembryonic portion of the VU within the AGM (**Revised Figure 3, Figure 4 and Revised Figure S4**).

We are also thankful for the suggestion to include a summary figure. We have generated a graphical schematic and included it in **Revised Figure 8G** (also shown below in **Figure 2** of this Rebuttal). We hope this will be helpful to the reader to summarize and fix the main concepts and findings of the study.

HE Timeline	E7.5-E8.5	E8.5-E9.5		E10.5-E11.5
Anatomical Localization Hematopoietic Progenitor Output				
Labeling Mode Cdh5-CreER^{T2}	4-OHT E7.5	4-OHT E8.5	4-OHT E8.5	4-OHT E10.5
Csf1^{MerCreMer}	4-OHT E8.5	4-OHT E9.5	No	No
Csf1r-iCre	Yes	Yes	No	No
Fetal Contribution Macrophages	Yes	Yes	Yes*	Low
Myeloid	No	Low	Yes	Low
B cell	No	Low	Yes	Low
T cell	No	Yes	Yes*	Yes

Figure 2. Summary schematic.

Conceptual schematic summarizing the timeline and the anatomical sites of localization of the subsets of HE identified within this and other studies, along with the labeling modes used to target these subsets or their immediate progenitors, and the observed contribution to fetal macrophages, myeloid cells, B and T lymphocytes (shown with E16.5 as a reference timepoint). (*): The asterisk indicates that the contribution has been evaluated using multiple Cre lines; i.e., macrophages and T cells are substantially contributed by *Csf1r*+ EMPs and/or LMPs (*Csf1^{MerCreMer}*, 4-OHT at E9.5). Because the combined contribution of EMPs plus fetal-restricted HSPCs (*Cdh5-CreER^{T2}*, 4-OHT at E8.5) is higher than the EMP contribution alone, an exclusive contribution of fetal-restricted HSPCs can be inferred.

Reviewer #4 (Remarks to the Author):

I was very pleased to review this manuscript by Barone et al. Major findings of the manuscript include the identification of a transient developmental wave of hematopoiesis which contributes to fetal lympho-myelopoiesis and with limited contribution to adult hematopoiesis. This work builds up on other recent reports showing the presence of HSC-independent waves of hematopoiesis that contribute to mid-gestation and late-gestation and adult hematopoiesis (Ulloa et al 2021; Yokomizo et al., 2022, Ganuza et al., 2022; Patel et al., 2022). Remarkably, Barone et al were able to ascribe anatomically this wave to the c-Kit+ clusters present in umbilical and vitelline arteries and not in the dorsal aorta or yolk sac, following endothelial-to-hematopoietic transition (EHT).

Authors employed a lineage tracing approach based on the tamoxifen inducible Cdh5-CreERT line and combination with three fluorescent reporters (R26-tdTomato, R26-zsGreen, and R26YFP). Tamoxifen (TAM) administration at E7.5, E8.5 and E10.5 allows them to with a good degree of specificity label three waves of hematopoiesis: erythroid-myeloid progenitors (EMP) (when treating at E7.5), definitive hematopoiesis (following TAM-treatment at E10.5) and the transient wave with lympho-myeloid potential which seems mostly constrained to the fetal stage. Lineage tracing with the Csf1r-Cre line, served them to label EMPs and confirm the absence of contribution from the yolk sac to the c-Kit+ clusters in vitelline and umbilical arteries.

Lineage-tracing followed by transcriptional profiling at single cell level of labelled progenitors allowed to characterized the hematopoietic precursors present in these vitelline and umbilical clusters and identify CD27 as an “exclusive” marker of these clusters. Finally, serial transplantation of fetal livers (FL) from E8.5 and E10.5 TAM-treated Cdh5-CreERT-R26TdTomato embryos showed that the precursors labelled at E8.5 contributed to the multilineage engraftment of primary recipients but not to HSPCs in the bone marrow and neither to any hematopoietic compartments in secondary recipients.

I consider that this elegant work could significantly advance our knowledge on fetal hematopoiesis. Still, I would like the authors to address the following points as there are some critical technical aspects that I don't understand.

We thank this Reviewer for the thorough and positive evaluation of our work. We trust that the revised manuscript will clarify the issues.

1) Since the activation of Cdh5-Cre-ERT labels endothelium, I really struggle to understand why/how TAM treatment at E7.5 or E8.5 does not result in the later labelling of any other wave of hematopoiesis as Cdh5 is not specific for any specific subtype of endothelium as far as it is known. Thus, EHT from labelled endothelium would contribute to any other later wave. Indeed, authors find labelling of the definitive wave when they label at E10.5. Only if the TAM-labelling of the endothelium happened after the completion of the EHT, one would expect no labelling of hematopoietic derived cells. Can authors show if a specific subfraction of the endothelium is labelled at each time point? Otherwise, where does the specificity on labelling each developmental wave come from? I really struggle here and I may be missing some details.

We thank the Reviewer for raising this important point. The *Cdh5-CreER^{T2}* model had already been used to selectively fate map YS and AGM hematopoiesis (see Gentek et al., Immunity 2018 PMID:29858009; Soares-da-Silva et al., J Exp Med 2021 PMID: 33566111). Our working hypothesis was that the specificity in the labelling of different hematopoietic waves in this model arises because hemogenic endothelium is essentially a transient population and most importantly it is not bipotent (Swiers et al., Nature Communications 2013. PMID: 24326267). Additionally, hemogenic endothelium was shown to represent a lineage distinct from arterial endothelium (Ditadi et al., Nature Cell Biology 2015; PMID: 25915127). Thus, the hemogenic endothelium expressing *Cdh5* (VE-Cadherin) that is labelled in our system within the 24-hour window provided by the persistence of 4-OHT in the organism undergoes endothelial-to-hematopoietic transition (EHT) and does not contribute to structural endothelial cells. Theoretically, there would still be the possibility of labeling hemogenic endothelial cells that do not immediately undergo EHT and persist in the organism after the 4-OHT labeling time window, but we considered this to be very unlikely because previous data suggest that EHT is accomplished in vivo in a rapid time window of a few hours. Time lapse imaging studies have suggested that a single event of EHT is completed in approximately 6-12 hours (Boisset et al., Nature 2010, PMID: 20154729). In our revised manuscript, we have confirmed in our model that this is indeed the case (see below)

In our whole-mount imaging experiments at E9.5 and E10.5, already included in the previous version of our manuscript, we had observed important differences in the labeling of endothelial cells. Interestingly, neither 4-OHT activation at E7.5 or E8.5 label (or label very few) endothelial cells in the ventral side of the aorta, providing a strong rationale for the lack of labeling of adult type HSC (known to arise from HE in this location) with these activation modes. In addition, 4-OHT at E7.5 labels a minority of endothelial cells and Kit⁺ hematopoietic clusters in the VU arteries, supporting the fact that this activation mode does not mark fetal-restricted HSPCs that here we demonstrate originating from hemogenic endothelial cells located in these vascular structures (See **Revised Figure 3C** and **3E**; **Revised Figure 7A**; additional **Figure 3** of this Rebuttal). This is now clearly mentioned in the revised manuscript text (lines 238-242). Moreover, the contribution of E7.5 labeled progenitors in the *Cdh5-CreER^{T2}* model to lympho-myelopoiesis at later developmental stages is comparable to what we observed with 4-OHT activation at E8.5 in the *Csf1r^{MerCreMer}* model (see **Revised Figure 4F**), supporting the idea that this activation mode well discriminates an early wave of EMPs.

Figure 3. Whole mount imaging of E10.5 *Cdh5-CreER^{T2}::R26^{EYFP}* AGM regions highlighting differences in endothelial labelling with 4-OHT activations at E7.5 or E8.5.

Confocal whole mount immunofluorescence analysis of E10.5 *Cdh5-CreER^{T2}::R26^{EYFP}* embryos. For both activations (4-OHT E7.5 and 4-OHT E8.5), left panels show maximum intensity 3D projections and right panels show a single 2.5μm slice. Boxed area in the upper merged image is magnified in the lower panels. da: dorsal aorta; va: vitelline artery; d:dorsal; v:ventral. Empty arrows indicate lack of EYFP labelling; white arrow indicates a labelled (EYFP+) endothelial cell.

For a more systematic and quantitative analysis of whether our pulse-labelling approach in *Cdh5-CreER^{T2}* embryos could differentiate between distinct endothelial subsets in hematopoietic sites and label discrete hemogenic endothelial subpopulations, we have:

- Performed new whole-mount imaging experiments of *Cdh5-CreER^{T2}::R26^{EYFP}* embryos with a different marker panel (CD31, Runx1 and EYFP) which enabled us to identify and analyze labelling in hemogenic endothelium in addition to hematopoietic cluster cells (results shown in **Revised Figure 3** and **Figure S4**);
- Performed a new comprehensive flow cytometric analysis with an expanded panel of markers (Ter119, CD31, Kit, CD41, CD45, CD43, CD201 and zsGreen). This allowed us to more precisely quantify the extent of labelling in endothelial and hemogenic endothelial cells with the three activation time points (new data included in **Revised Figure 3** and **Figure S4**);

These new experiments, in addition to the data already included in the manuscript, provided evidence that the three 4-OHT activation modes that we used in this study effectively lead to the labeling of distinct subsets of endothelial and hemogenic endothelial cells, which in turn supports the specificity of capturing the consecutive hematopoietic waves. Our new whole mount images confirmed that very few hemogenic endothelial cells in the DA are labelled with the E7.5 and E8.5 activations (**Revised Figure 3E-F**). We made the additional observation that 24-48 hours post-4-OHT activation hemogenic endothelium is generally not labelled in both AGM and YS, as shown by both flow cytometry and whole-mount imaging (**Revised Figure 3E-G** and **Figure S4F-H**). In the same tissues, hematopoietic cluster cells and phenotypic pre-HSPCs are labelled at a high frequency. This is in line with the hypotheses that: i) endothelial-to-hematopoietic transition events happen very quickly *in vivo*, and ii) virtually no untransitioned hemogenic endothelial cells are “left behind”, thus confirming our working hypothesis outlined above. Additionally, the fact that we detected low labeling of non-hemogenic endothelium in hematopoietic sites even with the activation models that label hematopoietic cells in the same locations (**Revised Figure 3G** and **Figure S4C,H**) raises the intriguing possibility that a significant portion of the early embryonic endothelium of these regions may in fact be hemogenic, as recently proposed. These points are now mentioned and discussed in the revised manuscript (lines 255-261; discussion section lines 463-479). We believe that these new data not only provide support for the specificity in the labelling of different hematopoietic waves in our model but also offer important novel biological insight.

2-Also, I don't understand the reason on using the many different reporter lines (i.e. R26tdTomato, R26zsGreen and R26EYFP). In line 125 they indicate: "An advantage of this strategy is to use just one Cre mouse line, thus avoiding bias from cell type-specific promoters". I don't understand the advantage of all these reporters. Can authors indicate if they found any difference among them? Is the efficiency of recombination of one allele different from other ones? Did any of the reporters led to different results which could affect conclusions? Like higher % of labelling of clusters in other locations? Can the authors detail which are the recombination efficiencies for each of the reporters? And justify why each reporter was employed in a particular experiment type if any specific reason.

The reason behind the use of the different reporter lines is mainly technical and explained below. The *R26^{tdTomato}* and *R26^{zsGreen}* are essentially the same transgenic line containing different fluorescent proteins (Madisen et al. 2010) and, accordingly, as shown in **Revised Figure S2G**, we observed no significant difference in their recombination efficiency.

We additionally used the *R26^{EYFP}* line mainly because in our experience it represented a valid option for whole-mount imaging of AGM and caudal parts, which are cleared with a non-aqueous solution (benzyl alcohol-benzyl benzoate, BABB). It is not possible to reliably detect fluorescent proteins in this solution without the use of an antibody. As a well-performing anti-GFP antibody that also recognizes EYFP was available and we had prior experience with it (Azzoni et al., *Development* 2014 PMID: 24757004; Neo et al. *Nature Cardiovascular Research* 2025 PMID: 41219569), we conducted most of our embryo imaging studies using this line. The exception was the whole-mount imaging of yolk sacs, as in this case sample preparation does not require the use of the BABB solution and therefore zsGreen could be visualized

without the use of an antibody (**Revised Figure S1D; S4D**). However, the $R26^{EYFP}$ line has a lower recombination efficiency than the $R26^{tdTomato}$ and $R26^{zsGreen}$ lines, as mentioned above and also previously reported (Patel et al., *Nature* 2022, PMID: 35705805).

A direct comparison of the efficiency of $R26^{tdTomato}$ and $R26^{EYFP}$ reporters in our experimental setting ($Cdh5-CreER^{T2}$) is shown in **Figure 4** of this Rebuttal. Here, we show that with the $R26^{tdTomato}$ reporter the labeling of E10.5 YS EMPs reaches >90% efficiency by flow cytometry, whereas with the $R26^{EYFP}$ line, which was mainly used for immunofluorescence analyses in this study, the labeling rate is lower, but still around 70-75%.

Figure 4. Recombination efficiency of E10.5 EMPs in $Cdh5-CreER^{T2}$ mice assessed by flow cytometry with two different reporter lines ($R26^{EYFP}$ and $R26^{tdTomato}$)

Quantification of flow cytometric analysis of labeled EMPs (Ter119- CD41lo Kit+ CD16-32+) in E10.5 $Cdh5-CreER^{T2}::R26^{EYFP}$ YS (n=12) and E10.5 $Cdh5-CreER^{T2}::R26^{tdTomato}$ YS (n=10). A single pulse of 4-OHT was administered at E8.5. Samples were analysed in 3 independent experiments.

Thus, we preferred $R26^{tdTomato}$ and $R26^{zsGreen}$ for most analyses other than whole-mount imaging (flow cytometry analyses and cell sorting for transcriptional and functional assays). In general, our results were always reproducible with the different reporter lines and aside from the expected lower efficiency of the EYFP reporter we did not observe any bias in our results. Indeed, in the new whole mount immunofluorescences included in the revised manuscript we have used the $R26^{EYFP}$ for the YS as well, and we could see no notable differences in labeling pattern as compared to the previous data obtained with the $R26^{zsGreen}$ line (see **Revised Figure S4F-G** and compare with **S4D-E**).

3) *Figure 6 and Figs S4-5.*

-Can the authors elaborate on the reason behind pooling YS and VU cells following labelling at E7.5 and E8.5?

-Even though authors classified labelled and unlabelled clusters from different clusters by scRNAseq, it would be helpful if they could validate these results by performing simpler flow

cytometry characterization of E9.5 and E10.5 Vitelline and Umbilical clusters from E8.5 TAM-treated embryos rather than a mix of YS+VU. So that they could employ markers for pro-HSC, Pre-type I and pre-type II.

The reason behind pooling YS and VU at E10.5 was largely derived from the whole-mount imaging data – specifically, the distribution of labelled hematopoietic clusters. At E10.5, both the YS arteries and the VU arteries in *Cdh5-CreER^{T2}::R26^{EYFP}* and *Cdh5-CreER^{T2}::R26^{ZsGreen}* embryos contain large Kit⁺ CD27⁺ clusters which are labelled with the E8.5 activation (and not labelled/much less so at E7.5), while the DA does not. The majority of cells within those clusters were not labeled with the *Csf1r-iCre* line, suggesting an origin independent from EMPs (**Revised Figure 4 and Figure S5A**). It is nearly impossible to cleanly dissect the YS plexus away from the YS arteries. Therefore, we pooled YS cells together with the vitelline and umbilical arteries because both of these tissues contain those large clusters that exhibit a similar immunophenotype and labeling dynamics.

At the same time, it is also very difficult, if not impossible, to cleanly dissect the VU away from the DA at E10.5, because at this stage the vitelline artery (VA) is connected to the DA through vascular ramifications that often contain Kit⁺ hematopoietic clusters (these can be seen in **Revised Figure 3C and 3E and Figure 4** of this Rebuttal; see also Yokomizo et al., Blood 2011, PMID: 21505195). Therefore, it is only with WM imaging that we can precisely assign an anatomical localization to the labelled cells. Based on the labeling pattern and the extra-aortic localization of these clusters, we considered them as part of the VA. In our interpretation, this is the most probable reason why in our scRNA-seq analysis we observe labelled pre-HSPCs in the AGM: those are likely to be located in the ramifications of the VA that was impossible to separate from the DA (given the scarce labeling of aortic clusters with 4-OHT at E8.5). We made this point clearer in the revised manuscript.

We performed the experiment suggested by the Reviewer, although we did this at E11.5 because: i) we wanted to compare these results with those shown in Figure 1 and ii) the VU at this stage is much easier to dissect and there are more cells to analyze, which makes data on relatively uncommon subpopulations such as pre-HSCs much more reliable. We dissected the VU separately from the YS and AGM and performed a flow cytometry analysis with our improved pre-HSC panel (See also response to Reviewer #1 point 1). The result of this analysis is shown in **Revised Figure S2F** and discussed in the revised manuscript (lines 160-163). We observed no significant differences in the labelling frequency of hematopoietic progenitors (highly labelled at this stage) or Type I and II pre-HSCs in the YS as compared to the VU. Thus, we think this provides a rationale for the pooling of these tissues in other experiments.

Given this result, and the limited availability of mice, we decided not to perform the experiment at E10.5. Moreover, we excluded the E9.5 stage because at this stage of development it is technically unfeasible to cleanly dissect the vitelline and umbilical arteries (VU) separately from the dorsal aorta, due to the fact that these vascular structures are very close and connected to each other (see **Revised Figure 3A, 4A** and Yzaguirre and Speck *Developmental Dynamics* 2016, PMID: 27389484).

4) *Figure 7F-G. Chimerism from transplants should be provided in a more standard format to*

facilitate the interpretation of results. Knowing the % of total chimerism of any fraction would help to interpret the results. Currently, this is not clear

The data previously shown in the submitted manuscript did not show levels of chimerism. Instead, they represented a longitudinal analysis of the percentages of labeling in PB myeloid, B and T cells and in BM progenitor subsets of transplant recipients. These labelling frequencies are shown within donor cells (CD45.2+) and have been normalized to the initial level of labeling in donor tissues (FL or BM). The main reason for normalizing is because of the intrinsic variability of the labeling observed in our experimental setting. Therefore, we were confident that this way of presenting data would facilitate the interpretation of our results instead of complicating it.

However, we acknowledge that the interpretation of our data was probably not immediate for the reader and needed improvement. We also realise that another reviewer made a similar comment (Please also see response to Point 10 of reviewer #5). Therefore, we now added chimerism information in the revised figure (**Revised Figure 8E,F**), and simplified the wording currently used in the figure to label the Y axis of the graphs (which now reads “Donor cell labelling in PB/BM normalized on input”). We also simplified the experimental schematic of transplant experiments (**Revised Figure 8A**), now included in the same figure with the new E11.5 AGM transplant experiments (**Revised Figure 8**). We trust that these changes will help the interpretation of our results and thus improve conveying the main message of our work.

In addition, complete transplantation data, including the percentages of chimerism and labeling in each fraction, are also included in the manuscript as Supplementary Tables (**Tables S4, S5 and S6**). Of note, chimerism levels were comparable for samples activated with 4-OHT at E8.5 or E10.5 and generally high (>90%) in PB myeloid and B cells as well as in BM HSPCs of primary transplants, while, as expected, more dynamic but still above threshold for T cells. Secondary transplants followed the same trend, except for chimerism in BM HSCs which was more variable and generally lower, but above threshold and comparable among samples (See Revised **Figure 8**).

*5) For strain *Csf1r-iCRE*, detailing some extra background and potential limitations as not 100% specificity, if that is the case should be included and discussed accordingly.*

We were not sure of what exactly the Reviewer was asking here; we thought that he/she may refer to the fact that the *Csfr-iCre* mouse line is not specific for EMPs at all developmental stages. Indeed, the constitutive *Csfr-iCre* strain, in addition to labeling EMPs, traces the myeloid progeny of HSPCs. Furthermore, *Csf1r* is expressed in LMPPs and B cell progenitors in the FL (Boiers et al Cell Stem Cell 2013 PMID: 24054998; Zriwil et al Blood 2016 PMID: 27207794). Therefore, although this mouse model was important to demonstrate that hematopoietic clusters in the major arteries arise independently of EMPs (**Revised Figure 4A-E, Figure S5A**), it would have been difficult to discriminate an EMP vs HSPC origin at developmental stages later than E10.5 with the same model.

To trace the progeny of EMPs in FL and thymus at late gestational stages, we thought that the best possible approach would involve using the inducible *Csf1r^{MerCreMer}* transgenic mouse line.

Thus, to conclusively determine the extent of the fetal lympho-myeloid contribution of EMPs, we have now performed new lineage tracing analyses using this inducible mouse line not previously included in the manuscript (*Csf1r^{MerCreMer}::R26^{tdTomato}*). Previous results using this approach showed that the late gestation (E16.5) lympho-myeloid contribution of EMPs labelled with 4-OHT at E9.25 was limited (~5%) (Atkins et al., J Exp Med 2022 PMID: 34928315). We have chosen both the E8.5 and E9.5 activation time points as both of these have been previously used to trace EMPs in the absence of HSC labeling (Gomez-Perdiguero et al, Nature 2015 PMID: 25470051; Hoeffel et al Immunity 2015 PMID:25902481; Iturri et al Immunity 2021, PMID: 34062116, Soares-da-Silva et al., JEM 2021 PMID: 33566111; Atkins et al., J Exp Med 2022 PMID: 34928315; and others). Using this mouse line, we analyzed labelling frequencies of HSPCs and lympho-myeloid cells in the E16.5 fetal liver, in addition to brain microglia and TSP in the thymus at the same stage. The E8.5 activation yielded high labeling of microglia and 20-25% of FL macrophages, a labeling pattern essentially overlapping with the E7.5 activation in *Cdh5-CreER^{T2}* mice. In contrast, the E9.5 activation labelled around a subset of myeloid and B cells (<20%) and 30-40% T cells in the E16.5 FL and thymus, with the highest labelling detected in more differentiated thymocytes (CD4+ CD8+ DP, 50% labeling). Microglia were still labelled at a very high frequency with the E9.5 activation. LSK, including HSCs and MPPs were not labelled with either activation. Labelling specificity was also confirmed by flow cytometry analysis of E9.5-E10.5 caudal part/AGM and YS/YS+VU, which confirmed that, in *Csf1r^{MerCreMer}::R26^{tdTomato}* embryos, most of the hematopoietic progenitors labelled with E8.5 and E9.5 activations exhibited an immunophenotype consistent with EMPs. Still, we cannot completely exclude a contribution from LMPs which were shown to emerge in the YS vascular plexus starting from E9.5 and to express *Csf1r* (Boiers et al Cell Stem Cell 2013 PMID: 24054998; Luis et al Nature Immunology 2016 PMID: 27695000). Thus, in *Csf1r*+ hematopoietic progenitors, which largely comprise EMPs, lymphoid potential appears only at E9.5.

These data allowed us to draw the following conclusions:

- i) Comparison with the *Cdh5-CreER^{T2}* data suggests that in the E16.5 FL, the majority of lympho-myeloid cells originate independently from EMPs or *Csf1r*+ progenitors arising between E8.5 and E10.5. Thus, most lympho-myeloid cells in the E16.5 FL likely originate from fetal-restricted HSPCs;
- ii) The higher labeling frequencies observed in the thymus with the activation at E9.5 in *Csf1r^{MerCreMer}* mice emphasize the HSC-independent origin of the first TSPs and reinforce the notion that T- and B-myeloid-producing progenitors during development are functionally uncoupled (Kawamoto et al., 2000 PMID: 10795742; Ramond et al., 2014 PMID: 24317038; Berthault et al., 2017 PMID: 28825702). This point is further supported by our new LDA assay showing a higher frequency of T cell progenitors than B cell progenitors within E8.5-labelled cells in the *Cdh5-CreER^{T2}* E10.5 AGM and YS+VU (**Revised Figure 5F,G**).

These new data are included in **Revised Figure 4** and **Figure S5**. In addition, we have removed the statement “*EMPs do not significantly contribute to fetal lympho-myelopoiesis*” and we have updated the Discussion section to argument these points (lines 480-493). Please also see Response to Reviewer 1, point #3.

6) Even though authors, detected a good labelling specificity for different waves when labelling at E7.5, E8.5 and E10.5, this is not perfect and the interpretation of limitations and conclusions should be well-discussed.

We thank the reviewer for this comment. We agree that in the *Cdh5-CreER^{T2}* model there may be some degree of overlap in the labelled EHT waves, in particular regarding YS EMPs, which are labelled at both E7.5 and E8.5 activations (as shown in **Revised Figure S1**). Thus, the later contribution of the EMPs labelled between E8.5 and E9.5 cannot be discriminated from the contribution of fetal-restricted HSPCs labelled with the same activation mode. EMPs are also labelled with 4-OHT at E7.5 in the same model, but their later hematopoietic contribution is limited (see **Revised Figure 2**). Our reasoning was that although YS EMPs were labelled with both E7.5 and E8.5 activations to near 100% efficiency, we observed a major contribution to fetal lympho-myelopoiesis only with the activation at E8.5; therefore, we deduced that this must have come from something other than EMPs.

We trust that the new experiments employing the *Csf1r^{MerCreMer}* transgenic mouse line that we included in the revised manuscript (presented in **Revised Figure 4** and **Figure S5**) and discussed in detail in the answer to the previous point of this Reviewer and in the reply to Point 3 of Reviewer #1, helped overcoming the limitations of other models and allowed us to draw meaningful conclusions. In addition, we are confident that the new imaging and flow cytometry experiments and analysis, discussed in detail in the answer to point 1 of this Reviewer, helped strengthening the rationale for the labeling specificity of different hematopoietic waves in the *Cdh5-CreER^{T2}* model and ultimately supported a better interpretation of our lineage tracing data.

Considering the Reviewers' suggestions and the new data included in the Revised manuscript, we have rewritten the Discussion section. We hope that the Reviewer will find it improved. In addition, in the revised manuscript we have included a summary visual schematic (**Revised Figure 8G** and shown in **Figure 3** of this Rebuttal), which we hope will be helpful to the reader.

Reviewer #5 (Remarks to the Author):

Barone et al. examine the origins of hematopoietic stem and progenitor cells (HSPCs) which contribute to the establishment of fetal-restricted lympho-myelopoiesis versus adult hematopoiesis. The authors' strategy was to activate reporter genes using a tamoxifen/4-OHT activated Cre driven by regulatory sequences from the Cdh5 gene, which encodes vascular endothelial cadherin (VEC), at different times during development. They identify HSPCs labeled by injection of 4-OHT into pregnant dams at embryonic day 8.5 that significantly contributes to hematopoiesis in the fetal liver and developing thymus, and a later wave of HSPCs labeled at E10.5 that contribute more robustly to adult hematopoiesis. These data are consistent with several previous studies that have documented the presence of HSPCs emerging prior to adult repopulating HSCs in the embryo that contribute to fetal hematopoiesis, but minimally to adult hematopoiesis. The novelty of the Barone et al. study is that they claim to have defined the anatomic location of these fetal-restricted HSPCs as the vitelline and umbilical arteries by correlating the timing of their injections to when labeled hematopoietic clusters are present and also based on the location of CD27+ cells by histology.

Their data that early activation (E8.5) of a reporter gene results in more labeled HSPCs and differentiated blood cells in the fetus than activation at E10.5, and activation at E10.5 results in more cells that persist in the adult mouse are convincing. However, the weakness of the study is that the labeling of the different populations of HSPCs is not clean. Some of the data for labeling specific populations are very convincing – for example it appears that labeling endothelial cells at E10.5 completely misses the yolk sac EMP wave. However, it is not clear that later waves, i.e. LMPs, fetal-restricted HSCs, and adult HSCs can be cleanly separated using this approach, as they may all be differentiating from hemogenic endothelial cells throughout the E8.5 -E11.5 window (albeit in different proportions) which compromises the authors' ability to definitively localize these HSPCs anatomically. The authors should at least discuss the caveats to their approach and be mindful of not overinterpreting their data.

We thank this Reviewer for the thorough reading and evaluation of our manuscript. We are aware of the potential caveats to our experimental system. Indeed, as pointed out by the Reviewer, and as mentioned in our manuscript, our interpretation is that the E8.5 4OHT activation in *Cdh5-CreER^{T2}* mice labels a mixed population of HSPCs, which hematopoietic contribution is largely restricted to fetal life. In the E11.5 AGM and YS+VU, we found that the labelling of non-preHSC hematopoietic progenitors (CD31+ Kit+ CD41low CD45+/- CD43+ CD201-) was exclusive to the E8.5 activation (**Revised Fig.1**). The level of labelling of adult-type HSCs with this activation mode is much inferior, as showed in multiple ways in our manuscript (e.g. **Revised Figure 2 and Figure 8**). In particular, our transplantation data, including the new E11.5 AGM transplants presented in **Revised Figure 8**, argue that very few, if any, functional adult-type HSCs are labelled with this activation mode.

Some degree of overlap between the hemogenic endothelial subsets labelled at different time point is inevitable due to the fact that, as in all inducible Cre-LoxP transgenic mouse lines, when performing pre-natal activation, it is not possible to precisely control the timing of the activation and intra-litter variability in embryo development also contributes to the variance. Still, we are confident that the different proportions of labelling mentioned by the

reviewer, in addition to the new data included in the revised manuscript, allowed us to draw strong conclusions.

In the revised manuscript, as suggested by the Reviewer, we have 1) added new experiments and 2) improved the discussion of the caveats of our experimental approach.

In particular, one issue we experimentally addressed involved clarifying the contribution of YS EMPs. 4-OHT at E8.5 in the *Cdh5-CreER^{T2}* model labels YS EMPs (see **Revised Figure S1**) and therefore the contribution of the EMPs labelled between E8.5 and E9.5 could not be discriminated from the contribution of fetal-restricted HSPCs labelled with the same activation mode. EMPs are also labelled with 4-OHT at E7.5 in the same model, but their later hematopoietic contribution is very limited (see **Revised Figure 2**). Our reasoning was that although YS EMPs were labelled with both E7.5 and E8.5 activations to near 100% efficiency, we observed a major contribution to fetal lympho-myelopoiesis only with the activation at E8.5; therefore, we deduced that this must have come from something other than EMPs. To address this issue and to generate further insight into the lympho-myeloid contribution of EMPs, we have now performed new lineage tracing analyses using an inducible transgenic mouse line not previously included in the manuscript (*Csf1r^{MerCreMer::R26^{tdTomato}}*). Previous results using this approach showed that the late gestation (E16.5) lympho-myeloid contribution of EMPs labelled with 4-OHT at E9.25 was limited (~5%) (Atkins et al., J Exp Med 2022 PMID: 34928315). We have chosen both the E8.5 and E9.5 activation time points as both of these have been previously used to trace EMPs in the absence of HSC labeling (Gomez-Perdiguero et al, Nature 2015 PMID: 25470051; Hoeffel et al Immunity 2015 PMID:25902481; Iturri et al Immunity 2021, PMID: 34062116, Soares-da-Silva et al., JEM 2021 PMID: 33566111; Atkins et al., J Exp Med 2022 PMID: 34928315; and others). Using this mouse line, we analyzed labelling frequencies of HSPCs and lympho-myeloid cells in the E16.5 fetal liver, in addition to brain microglia and TSP in the thymus at the same stage. The E8.5 activation yielded high labeling of microglia and 20-25% of FL macrophages, a labeling pattern essentially overlapping with the E7.5 activation in *Cdh5-CreER^{T2}* mice. In contrast, the E9.5 activation labelled around a subset of myeloid and B cells (<20%) and 30-40% T cells in the E16.5 FL and thymus, with the highest labelling detected in more differentiated thymocytes (CD4+ CD8+ DP, 50% labeling). Microglia were still labelled at a very high frequency with the E9.5 activation. LSK, including HSCs and MPPs were not labelled with either activation. Labelling specificity was also confirmed by flow cytometry analysis of E9.5-E10.5 caudal part/AGM and YS/YS+VU, which confirmed that, in *Csf1r^{MerCreMer::R26^{tdTomato}}* embryos, most of the hematopoietic progenitors labelled with E8.5 and E9.5 activations exhibited an immunophenotype consistent with EMPs. Still, we cannot exclude a contribution from LMPs which were shown to emerge in the YS vascular plexus starting from E9.5 and to express *Csf1r* (Boiers et al Cell Stem Cell 2013 PMID: 24054998; Luis et al Nature Immunology 2016 PMID: 27695000). Thus, in *Csf1r*+ hematopoietic progenitors, which largely comprise EMPs, lymphoid potential appears only at E9.5.

These data allowed us to draw the following conclusions:

- i) Comparison with the *Cdh5-CreER^{T2}* data suggests that in the E16.5 FL, the majority of lympho-myeloid cells originate independently from EMPs or *Csf1r*+ progenitors arising between E8.5 and E10.5. Thus, most lympho-myeloid cells in the E16.5 FL likely originate from fetal-restricted HSPCs;

- ii) The higher labeling frequencies observed in the thymus with the activation at E9.5 in *Csf1r^{MerCreMer}* mice emphasize the HSC-independent origin of the first TSPs and reinforce the notion that T- and B-myeloid-producing progenitors during development are functionally uncoupled (Kawamoto et al., 2000 PMID: 10795742; Ramond et al., 2014 PMID: 24317038; Berthault et al., 2017 PMID: 28825702). This point is further supported by our new LDA assay showing a higher frequency of T cell progenitors than B cell progenitors within E8.5-labelled cells in the *Cdh5-CreER^{T2}* E10.5 AGM and YS+VU (**Revised Figure 5F,G**).

These new data are included in **Revised Figure 4** and **Figure S5**. In addition, we have removed the statement “*EMPs do not significantly contribute to fetal lympho-myelopoiesis*” and we have updated the Discussion section to argue these points (lines 480-493). We are confident that the Reviewer will find these new data informative (See also reply to Point 3 of Reviewer #1).

Additionally, to improve the anatomical localization of the HSPC waves identified, we have investigated in more detail the labelling dynamics of endothelial and hemogenic endothelial cells at each time point of activation. To demonstrate that our labelling approach in *Cdh5-CreER^{T2}* mice can differentiate between distinct endothelial and hemogenic endothelial subpopulations, we have:

- Performed new whole-mount imaging experiments of *Cdh5-CreER^{T2}::R26^{EYFP}* embryos with a different marker panel (CD31, Runx1 and EYFP) which enabled us to identify and analyze labeling in hemogenic endothelium in addition to hematopoietic cluster cells (results shown in **Revised Figure 3** and **Figure S4**);
- Performed a new comprehensive flow cytometric analysis with an expanded panel of markers (Ter119, CD31, Kit, CD41, CD45, CD43, CD201 and zsGreen). This allowed us to more precisely quantify the extent of labelling in endothelial and hemogenic endothelial cells with the three activation time points (new data included in **Revised Figure 3** and **Figure S4**);

These new experiments, in addition to the data already included in the manuscript, provide evidence that the three 4-OHT activation modes that we used in this study effectively lead to the labeling of distinct subsets of endothelial and hemogenic endothelial cells, which in turn supports the specificity of capturing the consecutive hematopoietic waves. Indeed, neither 4-OHT activation at E7.5 or E8.5 label (or label very few) endothelial and hemogenic endothelial cells in the ventral side of the dorsal aorta, providing a strong rationale for the lack of labeling of HSCs with these activation modes. Moreover, it appears very unlikely that activation with 4-OHT at E7.5 labels fetal-restricted HSPCs, which we show to originate from hemogenic endothelium of the vitelline and umbilical arteries between E8.5 and E9.5, given that this activation mode: 1) labels a minority of endothelial cells in the vitelline and umbilical arteries and 2) yields a low labeling of Kit⁺ hematopoietic cluster cells located in these vascular structures. We also made the additional observation that 24-48 hours post-activation we found very few hemogenic endothelial cells labelled in both AGM and YS (as shown by both flow cytometry and whole-mount imaging). In the same tissues, hematopoietic cluster cells and phenotypic pre-HSPCs are labelled at a high frequency. This result is in line with the hypotheses that: i) endothelial-to-hematopoietic transition events happen very quickly *in*

vivo, and ii) virtually no untransitioned hemogenic endothelial cells are “left behind” (see also reply to Reviewer #4, point 1).

Another concern is that the authors used the term “pre-HSC” quite liberally without providing functional data to define these cells. A pre-HSC has historically been defined as a cell that cannot engraft adult mice directly but can be matured ex vivo into an adult-repopulating HSC. Molecular profiling of purified pre-HSCs by multiple groups have identified a consistent gene signature. The authors use the term pre-HSC more loosely, based on annotation in their single cell RNA-sequencing data which they don’t directly compare to published data, and the presence of cell surface CD27 at E10.5. Moreover, they describe two distinct populations of pre-HSCs that have different molecular signatures, and based on CD27 expression localize one of these pre-HSC populations (pre-HSC 2) to the E10.5 vitelline vasculature. However, neither of the previous studies referenced by the reviewers demonstrate that CD27 marks pre-HSCs at E10.5. In fact, CD27 does not mark type I pre-HSCs at E11.5, so it is unlikely to mark type I pre-HSCs at 10.5 (all functional pre-HSCs at E10.5 are type I pre-HSCs). Unless the authors demonstrate that E10.5 pre-HSCs are CD27+, this marker cannot be used to identify them by histology which is how the authors localize them to the vitelline artery. In general, given that other progenitors are present in the embryo at the times the authors perform their analyses (e.g. LMPs and embryonic multipotent progenitors), it is not clear at all that what they call pre-HSC 2 are in fact pre-HSCs or one of these other progenitor types.

We apologize for the use of the “pre-HSC” term, which we agree was too “liberal” in absence of a functional connotation. We are not claiming that all or most phenotypically or transcriptionally identified pre-HSCs that are labelled with the E8.5 activation in the *Cdh5-CreER^{T2}* mouse model are also pre-HSCs as defined functionally. What our data instead suggests is that fetal-restricted HSPCs labelled at E8.5 may emerge from cells that express markers that have been previously used to identify progenitors and/or pre-HSCs (see **Revised Figure 1** and **Figure S2**) – but are not necessarily functional pre-HSCs as defined by *ex vivo* culture followed by transplantation assays.

Indeed, it was already known that even with the best available definition of markers, not all phenotypic pre-HSCs also correspond to functional HSCs; at least two third are not (Zhou et al., Nature 2016 (PMID: 27225119)) and the population of intra-aortic clusters and more in general of pre-HSCs is very heterogeneous (Barone et al, Cells 2022 PMID: 35326511).

Regarding the comments related to CD27: in addition to marking type II pre-HSCs, this marker was shown to highly enrich for progenitors with lymphoid potential at both E10.5 and E11.5 emerging prior to HSCs (Li et al, Blood 2017 PMID:28588017). Given that Type I pre-HSCs do not express CD27, it is likely that at E10.5 progenitors with lymphoid potential represent a significant portion, if not the majority of fetal-restricted HSPCs (as also suggested by the analysis shown in **Figure 5** of this Rebuttal and included in the revised manuscript). Therefore, we think that this marker can be used to reliably identify fetal-restricted pre-HSPC by immunofluorescence at this developmental stage, as shown by our analysis.

To address the points raised here by reviewer, we have:

1) Performed an improved flow cytometry analysis of pre-HSCs and hematopoietic progenitors using an expanded panel of markers (Ter119, CD31, Kit, CD41, CD45, CD43, CD201 and zsGreen), now included in **Revised Figure 1** (see also response to points 1 and 2 of Reviewer #1);

2) Rephrased the terminology in our manuscript to avoid potentially misleading claims regarding pre-HSCs. More specifically: we have changed the term “pre-HSC” to “pre-HSPC” throughout the manuscript when referring to the population of cells labelled with the E8.5 activation in the *Cdh5-CreER^{T2}* model. We believe this is a more accurate terminology, given the fact that this cell population probably contains a mixture of progenitors and fetal HSCs (see new E11.5 transplantation data in **Revised Figure 8**). Additionally, when referring to immunophenotypically identified pre-HSC we have used the “Type I pre-HSC” and “Type II pre-HSC” denominations (e.g. **Revised Figure 1**), to avoid confusion with clusters 29 and 30 of our scRNAseq dataset, which we named pre-HSPC 1 and pre-HSPC 2. We believe that these changes will be helpful for improving clarity and avoiding confusion in the field.

3) We thank the reviewer for the useful suggestion of comparing our scRNA-seq data to published datasets. We have now done this analysis. We explored different annotations and found the most informative to be Vink et al., Cell Reports 2020 (PMID: 32402290) for HSC signature and the “uncommitted progenitors” (Prog) and “lymphoid-like cells” (Ly) annotations from Patel et al., Nature 2022 (PMID: 35705805).

As shown in **Figure 5** of this rebuttal, cluster #29 (pre-HSPC 1) in our dataset exhibits similarity to HSCs and uncommitted progenitors, and this signature is particularly expressed in unlabelled AGM cells. On the other hand, the lymphoid progenitor signature is more expressed in cluster #30 (pre-HSPC 2) and in particular in labelled cells of YS+VU. We included this new analysis in the revised manuscript (lines 345-350; **Revised Figure 6E-F** and **S7E-F**).

Figure 5. Mapping of our scRNAseq data to existing datasets.

Gene expression of published signatures for HSC (Vink et al., *Cell Reports* 2020), AGM-derived progenitors (“Prog”) and lymphoid progenitors (“Ly”) (Patel et al., *Nature* 2022). The average expression of top50 genes for each signature was calculated on single cells using the AddModuleScore function in Seurat.

Specific comments:

1. Pre-HSCs in Fig 1A were characterized using CD41, CD43, and CD45. This is a relatively crude set of markers, and better ones, including Kit and CD201 along with endothelial markers have been described (see for example PMID: 27225119). The authors should refine their phenotypic pre-HSC analysis and demonstrate by *ex vivo* maturation and transplant that what they are calling pre-HSCs have pre-HSC activity.

We agree with this reviewer (and the other reviewer who made a similar comment) that the panel of antibodies we previously used for the flow cytometric analysis of pre/pro-HSCs in the E11.5 AGM and YS+VU included a limited set of markers, and thus may have led to skewed results, due to the fact that what we previously identified as pre-HSC II was in fact a mixed population, also including progenitors. To better define immunophenotypic type I and type II pre-HSCs in the AGM and YS+VU of E11.5 embryos, we implemented a more refined flow cytometry panel with the use of additional surface markers. Based on the work of Zhou et al., Nature 2016 (PMID: 27225119), we performed a new analysis which included the markers Ter119, CD31, Kit, CD41, CD45, CD43, CD201 and zsGreen. Regarding the specific use of an anti-VE-Cadherin antibody, we decided against using this specific marker because in our experience it usually yielded a highly variable labeling, particularly dependent on experimental conditions. More specifically, our tissue dissociation protocol involves the use of enzymes such as collagenase, which can degrade the epitope recognized by commercially available anti-VE-Cadherin antibodies, thereby compromising the accurate detection of this marker. Therefore, we used the CD31 as a vascular marker instead.

In our new panel, immunophenotypically defined type I pre-HSCs were defined as: CD31+ Kit+ CD41^{low} CD45- CD43+ CD201+ and type II pre-HSCs as CD31+ Kit+ CD41^{low} CD45+ CD43+ CD201+. Hematopoietic progenitors other than pre-HSCs were defined as CD31+ Kit+ CD41^{low} CD45+/- CD43+ CD201-. As shown in the new data included in **Revised Figure 1** (commented at lines 149-163), in E11.5 embryos both type I and type II pre-HSCs are now labelled with both E8.5 and E10.5 activations, while the E7.5 activation still labels a minority of these subsets. This is in agreement with the labeling of LSK and HSCs observed later on in FL with the same activation modes. In contrast, non-HSC hematopoietic progenitors (defined as above) in both E11.5 AGM and YS+VU exhibited a much more selective labelling pattern, and, strikingly, were almost exclusively labelled by the E8.5 activation in AGM and YS+VU. This was a particularly relevant result, especially in light of the new transplant experiments shown in **Revised Figure 8**, which showed that in the E11.5 AGM we could detect donor-derived labelled cells only when donor AGMs were traced with the E8.5 activation – and thus donor-derived engraftment potential is likely coming from the CD201- progenitor population in this specific tissue and developmental stage (see also response to Reviewer #1 points 1-2).

Regarding *ex vivo* maturation and transplantation assays of AGM or YS+VU explants or co-aggregates, for the reasons outlined above (i.e. we are not claiming that all or most phenotypically or transcriptionally identified pre-HSCs that are labelled with the E8.5 activation in the *Cdh5-CreER^{T2}* mouse are also pre-HSCs as defined functionally. What our data instead suggests is that fetal-restricted HSPCs labelled at E8.5 emerge from cells that express markers of pre-HSCs, but are not necessarily functional pre-HSPC as defined by transplantation assays; see response to the previous comment of this Reviewer) we did not consider those technically challenging and lengthy experiments as essential for what concerned the primary scope of this manuscript, and thus in the interest of optimizing time and resources we so far did not perform them. Instead, we performed new direct transplantation experiments of *Cdh5-CreER^{T2}* E11.5 AGM and YS+VU traced at E8.5 (fetal restricted HSPCs) and E10.5 (HSCs). This new experiment enabled us to determine the engraftment potential of fetal-restricted HSPCs before fetal liver seeding (new data included in **Revised Figure 8 and Figure S10**). Our data suggests that adult repopulating potential in the E11.5 AGM is essentially confined to fetal-restricted HSPCs. This provides support to the recently emerging concept that the fetal liver is an important site of pre-HSC maturation.

2. Fig S1 H,I: The authors measure labeling of LMPs in the fetal liver at E11.5, but no analysis of LMPs was done prior to colonization of the fetal liver (CD45+ hematopoietic cluster cells at E10.5 by flow or confocal). Demonstrating labeling of LMPs should be shown in the tissues where they arise and have been defined (AGM and U+V arteries) to accurately represent the population.

Follow-up comment to the pre-revision rebuttal:

Hematopoietic cluster cells in the dorsal aorta at E10.5 contain CD45+ and CD45- cells. All the pre-HSCs are in the CD45- population at E10.5. The CD45+ hematopoietic cluster population at E10.5 will robustly produce lymphoid and myeloid cells ex vivo. So the experiment suggested by the reviewer is feasible – it is possible to measure LMP activity in the CD45+ hematopoietic cluster population in E10.5 embryos.

We thank the Reviewer for the useful comments. As suggested by the Reviewer, we performed a flow cytometric analysis of LMPs and type I pre-HSCs in the E10.5 AGM and YS+VU. The results are shown in **Revised Figure S2A-B** and commented at lines 141-144. LMPs were defined as CD31+ Kit+ CD45+. While in the AGM LMP labeling was partial with both activation time points, 4-OHT at E8.5 traced the majority of LMPs in the YS+VU, consistent with LMP emergence at/around E9.5 in the YS (Boiers et al., Cell Stem Cell 2013, PMID: 24054998).

3. Figure 2: The authors suggest that EMPs do not contribute significantly to fetal lympho-myelopoiesis based on the low contribution of labeled cells when 4-OHT is injected at E7.5. However, in Fig S1 C labeling of EMPs by E8.5 4-OHT treatment is quite high, which makes it difficult to separate the relative contribution of EMPs to FL lympho-myelopoiesis from the contribution of HSPCs in the vitelline arteries, which are also labeled by E8.5 4-OHT treatment.

We thank the reviewer for this comment. The issue was also raised from another Reviewer. Indeed, 4-OHT at E8.5 in the *Cdh5-CreER^{T2}* model labels YS EMPs (see **Revised Figure S1**) and therefore the contribution of the EMPs labelled between E8.5 and E9.5 cannot be directly discriminated from the contribution of fetal-restricted HSPCs labelled with the same activation mode. EMPs are also labelled with 4-OHT at E7.5 in the same model, but their later hematopoietic contribution is very limited (see **Revised Figure 2**). Our reasoning was that although YS EMPs were labelled with both E7.5 and E8.5 activations to near 100% efficiency, we observed a major contribution to fetal lympho-myelopoiesis only with the activation at E8.5; therefore, we deduced that this must have come from something other than EMPs.

To conclusively determine the extent of the fetal lympho-myeloid contribution of EMPs and thus discriminating it from the non-EMP contribution, we have now performed new lineage tracing analyses using an inducible transgenic mouse line not previously included in the manuscript (*Csf1r^{MerCreMer}::R26^{tdTomato}*). Previous results using this approach showed that the late gestation (E16.5) lympho-myeloid contribution of EMPs labelled with 4-OHT at E9.25 was limited (~5%) (Atkins et al., J Exp Med 2022 PMID: 34928315). We have chosen both the E8.5 and E9.5 activation time points as both of these have been previously used to trace EMPs in

the absence of HSC labeling (Gomez-Perdiguero et al, Nature 2015 PMID: 25470051; Hoeffel et al Immunity 2015 PMID:25902481; Iturri et al Immunity 2021, PMID: 34062116, Soares-da-Silva et al., JEM 2021 PMID: 33566111; Atkins et al., J Exp Med 2022 PMID: 34928315; and others). Using this mouse line, we analyzed labelling frequencies of HSPCs and lympho-myeloid cells in the E16.5 fetal liver, in addition to brain microglia and TSP in the thymus at the same stage. The E8.5 activation yielded high labeling of microglia and 20-25% of FL macrophages, a labeling pattern essentially overlapping with the E7.5 activation in *Cdh5-CreER^{T2}* mice. In contrast, the E9.5 activation labelled around a subset of myeloid and B cells (<20%) and 30-40% T cells in the E16.5 FL and thymus, with the highest labelling detected in more differentiated thymocytes (CD4+ CD8+ DP, 50% labeling). Microglia were still labelled at a very high frequency with the E9.5 activation. LSK, including HSCs and MPPs were not labelled with either activation. Labelling specificity was also confirmed by flow cytometry analysis of E9.5-E10.5 caudal part/AGM and YS/YS+VU, which confirmed that, in *Csf1r^{MerCreMer::R26^{tdTomato}}* embryos, most of the hematopoietic progenitors labelled with E8.5 and E9.5 activations exhibited an immunophenotype consistent with EMPs. Still, we cannot completely exclude a contribution from LMPs which were shown to emerge in the YS vascular plexus starting from E9.5 and to express *Csf1r* (Boiers et al Cell Stem Cell 2013 PMID: 24054998; Luis et al Nature Immunology 2016 PMID: 27695000). Thus, in *Csf1r*+ hematopoietic progenitors, which largely comprise EMPs, lymphoid potential appears only at E9.5.

These data allowed us to draw the following conclusions:

- i) Comparison with the *Cdh5-CreER^{T2}* data suggests that in the E16.5 FL, the majority of lympho-myeloid cells originate independently from EMPs or *Csf1r*+ progenitors arising between E8.5 and E10.5. Thus, most lympho-myeloid cells in the E16.5 FL likely originate from fetal-restricted HSPCs;
- ii) The higher labeling frequencies observed in the thymus with the activation at E9.5 in *Csf1r^{MerCreMer}* mice emphasize the HSC-independent origin of the first TSPs and reinforce the notion that T- and B-myeloid-producing progenitors during development are functionally uncoupled (Kawamoto et al., 2000 PMID: 10795742; Ramond et al., 2014 PMID: 24317038; Berthault et al., 2017 PMID: 28825702). This point is further supported by our new LDA assay showing a higher frequency of T cell progenitors than B cell progenitors within E8.5-labelled cells in the *Cdh5-CreER^{T2}* E10.5 AGM and YS+VU (**Revised Figure 5F,G**).

These new data are included in **Revised Figure 4** and **Figure S5** and described at lines 281-299. In addition, we have removed the statement “*EMPs do not significantly contribute to fetal lympho-myelopoiesis*” and we have updated the Discussion section to argument these points (lines 480-493). We trust that the Reviewer will agree that the new data and discussion have satisfactorily addressed the issue. (See also reply to Point 3 of Reviewer #1).

4. Figure 2B. The authors show that activating Cre at E8.5 labels more $\gamma\delta$ T cells in the thymus than labeling at E10.5 and conclude that they have labeled two separate waves of thymus-settling progenitors. But what if it is one continuous wave, and labeling at E8.5 captures it in its entirety, whereas labeling at E10.5 only catches the latter part of the wave? How would their approach discriminate between those two possibilities?

The existence of two separate waves of thymus-settling progenitors (TSP) is well supported by multiple evidences (Luc et al., Nature Immunology 2012, PMID: 22344248; Ramond et al., Nature Immunology 2014, PMID: 24317038; Luis et al., Nature Immunology 2016, PMID: 27695000; Elsaïd et al., Blood 2020, PMID: 33025012 and others). In particular, it is the first TSP wave that retains the ability of generating $\gamma\delta$ T cells, lost by TSP of the second wave (Ramond et al., Nature Immunology 2014, PMID: 24317038). The differential labeling of $\gamma\delta$ T cells in the fetal thymus with 4-OHT activation at E8.5 and E10.5 is not the only evidence supporting the labelling of two separate waves of TSPs in our system. 4-OHT at E8.5 labels more differentiated TSP (DN2 to DN4 and DP) at higher frequency than 4-OHT at E10.5, whereas with the latter activation mode the highest levels of labelling are observed in DN1 (and higher than 4-OHT at E8.5). If, as the reviewer is suggesting, there was one continuous wave captured in its entirety by the activation at E8.5, then we would most probably observe higher levels of labelling in DN1 with this activation mode, which is not the case.

All these things considered, we cannot conclusively prove that the labelling of two waves of TSP in our system is mutually exclusive in our system; at the same time, this is not the major focus of the present manuscript. We acknowledge that it is still possible that there is some overlap between the two waves labelled at E8.5 and E10.5. The conclusions that we can draw are based on the levels of labelling observed in the distinct TSP subpopulations captured by the different activations, which in our opinion are in support of the widely accepted two-wave model. For these reasons, in the revised version of the manuscript we have removed the statement currently claiming that “our tracing system can differentially label the two separate waves of thymic-settling progenitors (TSP)” and tuned it down to “Thus, our data is consistent with the previously reported model suggesting the existence of two separate waves of thymus-settling progenitors (TSP)” (lines 195-197).

Notably, the new data generated with the *Csf1r^{MerCreMer::R26^{tdTomato}}* mouse line, included in **Revised Figure 4F** and discussed in the reply to the previous point of this Reviewer is also in support of the two-wave TSP model.

5. Figure 3A,B. These data should be quantified.

The whole mount imaging data in 3A are already quantified in Figure 3B as the % of labelled Kit⁺ clusters in the E9.5 vitelline artery (VA).

*6. Figure 4. The authors utilized the *Csf1r-iCre* line to label EMPs and show significant infiltration of E8.5 *iCsf1r*-labeled EMPs in the E10.5 fetal liver. Analysis of the E14.5 fetal liver and thymus with these animals could give a better sense of EMP contribution to fetal lymphomyelopoiesis.*

The constitutive *Csf1r-iCre* allele, in addition to labeling EMPs, traces the myeloid progeny of HSPCs. Furthermore, *Csf1r* is expressed in LMPPs and B cell progenitors in the FL (Boiers et al Cell Stem Cell 2013 PMID: 24054998; Zriwil et al Blood 2016 PMID: 27207794). Therefore, although this mouse model was important to demonstrate that hematopoietic clusters in the major arteries arise independently of EMPs (**Revised Figure 4A-E, Figure S5A**), it would have been difficult to discriminate an EMP vs HSPC origin at developmental stages later than E10.5 with the same model.

For these reasons, we addressed this request with new lineage tracing experiments which were performed using inducible *Csf1^{MerCreMer}::R26^{tdTomato}* mice, as already discussed in the replies to points 3 and 4 of this Reviewer. These new data are included in **Revised Figure 4F** and **Figure S5B-D**. See also reply to Point 3 of Reviewer #1.

7. Fig 5 D,F: Plotting the outgrowth of lymphoid and myeloid cells as ratios from unsorted populations rather than sorting and plating labeled vs unlabeled cells from the AGM and VA makes interpretation of the data cumbersome. Limiting dilution plating of purified labeled and unlabeled cells would be more valuable and would allow for easier comparison of numbers of LMPs and their expansion in cultures.

We thank the reviewer for this useful comment – again, which raised an issue also pointed out by another Reviewer. We previously performed OP9 co-cultures for B/T cell potential on sorted cells by using the same cell isolation strategy shown in **Figure 5B** of the manuscript. These experiments essentially yielded the same result as the data shown in **Figure 5D,E** (see **Figure 2** of this rebuttal); however, this experiment was only done on E10.5 YS+VU. We subsequently performed the experiment on unfractionated cells from both AGM and YS+VU (included in the revised manuscript) to maximize cell viability and culture efficiency. We reasoned that post-culture normalization of B/T labelling on the initial labelling of Kit⁺ cells would still allow us to draw meaningful conclusions. We were encouraged by the fact that the results on unsorted cells+ normalization were consistent with our previous experiments done with sorted YS+VU cells.

In our revised manuscript, to streamline and simplify the presentation of the results, we have now reanalysed the T cell bulk OP9 co-culture data, which was previously shown separately for each progenitor subpopulation (DN1, DN2, etc.). Since the results were very similar and consistent among subpopulations and therefore not very informative, we now show the post-culture labelling of T cells “as a whole”, similar to what we did for B cells. This new analysis is presented in **Revised Figure 5E**; new gating strategies for OP9 (B cell) and OP9-DI1 (T cell) co-culture analysis are shown in **Revised Figure S6A,B**. Myeloid cell analysis was moved to **Revised Figure S6C**.

We agree with the reviewer that the previous version of our manuscript was missing a quantitative approach for *ex vivo* B and T differentiations. Therefore, to precisely establish the frequency of T and B cell progenitors in labelled versus unlabelled E10.5 hemato-vascular fractions (4-OHT at E8.5), we have performed new OP9/OP9-DL1 B/T co-culture assays using sorted cells from both AGM and YS+VU, this time by adopting a limiting dilution experimental setting (LDA). The cell isolation strategy for OP9 (B cell) and OP9-DI1 (T cell) LDA co-culture analysis is shown in **Revised Figure S6D,E** and it is the same that was also used for the isolation of hemato-vascular cells for scRNA-seq. The results of the OP9 LDA analysis are presented in **Revised Figure 5F-G** (lines 320-324) and indicate that T cell progenitors are present in labelled hemato-vascular fractions at a 1/81 frequency in the AGM and slightly less frequent in the YS+VU (1/123), while they are absent or very rare in nonlabelled hemato-vascular fractions. A similar scenario was seen in B cell LDA, although in the latter case the overall progenitor frequency was lower. These data indicate that the 4-OHT activation captures virtually the entirety of B and T cell progenitors in the E10.5 AGM and YS+VU. Of note, as mentioned, our

new LDA assay showed a higher frequency of T cell progenitors than B cell progenitors within E8.5-labelled cells in the *Cdh5-CreER^{T2}* E10.5 AGM and YS+VU (**Revised Figure 5F,G**), thus supporting the notion that T- and B lymphocyte-producing progenitors during development are functionally uncoupled, consistent with previous observations (Kawamoto et al., 2000 PMID: 10795742; Ramond et al., 2014 PMID: 24317038; Berthault et al., 2017 PMID: 28825702). The different frequencies of B and T lymphocyte progenitors further suggest that developmental uncoupling may already take place in the tissues of emergence of these precursors.

8. Figure S4. These data are generated from a relatively impure population, and don't add much to the story. I would suggest not including these data in the manuscript.

We agree with the Reviewer and we thank him/her for the suggestion. We have removed the data previously included in Figure S4 from the revised version of the manuscript.

9. Fig 7: The authors make claims about labeling different populations of pre-HSCs by injecting 4-OHT into pregnant dams at E8.5 and E10.5. Relatively convincing conclusions can be made about the contribution of HSPCs labeled by 4-OHT at E10.5 to fetal and adult hematopoiesis since HSPCs that form before E10.5 will not be labeled. However, administering 4-OHT at E8.5 may label endothelial cells that do not undergo EHT until E10.5, so overlapping waves of HSPCs may be labeled, making the system messy and difficult to interpret. The authors should quantify the percentage of labeled endothelial cells at E9.5 and E10.5 to determine the extent to which labeled hemogenic endothelial cells are present at these time points. Transplants of E11.5 AGMs should also be done to demonstrate the extent to which HSCs arising in the AGM region are labeled when 4-OHT is administered at E8.5.

We thank the Reviewer for this comment. The *Cdh5-CreER^{T2}* model has been previously used to selectively fate map YS and AGM hematopoiesis (see Gentek et al., Immunity 2018 PMID:29858009; Soares-da-Silva et al., J Exp Med 2021 PMID: 33566111). Our working hypothesis was that the specificity in the labelling of different hematopoietic waves in this model arises because hemogenic endothelium (HE) is essentially a transient, hematopoietic-committed population and, importantly, it is not bipotent (Swiers et al., Nature Communications 2013. PMID: 24326267). Additionally, HE was shown to represent a lineage distinct from arterial endothelium (Ditadi et al., Nature Cell Biology 2015; PMID: 25915127). Thus, the HE expressing *Cdh5* (VE-Cadherin) that is labelled in our system within the 24-hour window provided by the persistence of 4-OHT in the organism undergoes endothelial-to-hematopoietic transition (EHT) and does not contribute to structural endothelial cells. Theoretically, as the Reviewer is pointing out, there would still be the possibility of labeling hemogenic endothelial cells that do not immediately undergo EHT and persist in the organism after the 4-OHT labeling time window, but we considered this to be unlikely because previous data suggest that EHT is accomplished *in vivo* in a rapid time window of a few hours. Indeed, *ex vivo* time lapse imaging studies have suggested that a single event of EHT is completed in approximately 6-12 hours (Boisset et al., Nature 2010, PMID: 20154729). In our revised manuscript, we have confirmed in our model that this is also the case *in vivo* (see below).

We agree with the Reviewer regarding the importance of determining the extent of endothelial/hemogenic endothelial labelling the *Cdh5-CreER^{T2}* model in our experimental setting. Thus, we have:

- Performed new whole-mount imaging experiments of *Cdh5-CreER^{T2}::R26^{EYFP}* embryos with a different marker panel (CD31, Runx1 and EYFP) which enabled us to identify and analyze labeling in HE in addition to hematopoietic cluster cells (results shown in **Revised Figure 3** and **Figure S4**);
- Performed a new flow cytometric analysis with an expanded panel of markers (Ter119, CD31, Kit, CD41, CD45, CD43, CD201 and zsGreen). This allowed us, in addition to identifying Type I/Type II pre-HSC and non-preHSC progenitors labeling (**Revised Figure 1**), to more precisely quantify the extent of labelling in endothelial and hemogenic endothelial cells with the three activation time points. In this panel, endothelial cells could be identified as CD31+ Ter119- Kit- CD41- CD45- CD43- CD201- and hemogenic endothelial cells can also be identified as CD31+ Ter119- Kit+ CD41- CD45- CD43- CD201-. Indeed, emerging evidence is suggesting that Kit expression is initiated at the hemogenic endothelium stage (reviewed in Hou et al., *Blood Sci* 2024, PMID: 39027902; Neo et al. *Nature Cardiovascular Research* 2025 PMID: 41219569) (new data included in **Revised Figure 3** and **Figure S4**);

These new experiments, in addition to the data already included in the manuscript, provided evidence that the three 4-OHT activation modes that we used in this study effectively lead to the labeling of distinct subsets of endothelial and hemogenic endothelial cells, which in turn supports the specificity of capturing the consecutive hematopoietic waves. Our new whole mount images showed that very few CD31+Runx1+ hemogenic endothelial cells in the DA are labelled with the E7.5 and E8.5 activations (**Revised Figure 3E-F**). We made the additional observation that 24-48 hours post-4-OHT activation HE is generally not labelled in both AGM and YS, as shown by both flow cytometry and whole-mount imaging (**Revised Figure 3E-G** and **Figure S4F-H**). However, in the same tissues, hematopoietic cluster cells and phenotypic pre-HSPCs are labelled at a high frequency, and the high efficiency of the system is demonstrated by the labeling of reference populations, e.g. brain microglia (**Revised Figure S1**) and HSCs (**Revised Figure 2** and **S2**). Thus, our data are in line with the hypotheses that: i) EHT events happen very quickly *in vivo*, and ii) virtually no untransitioned hemogenic endothelial cells are “left behind”, thus confirming our working hypothesis outlined above. Additionally, the fact that we detected low labeling of non-hemogenic endothelium in hematopoietic sites even with the activation models that label hematopoietic cells in the same locations (**Revised Figure 3G** and **Figure S4C,H**) raises the intriguing possibility that a significant portion of the early embryonic endothelium of these regions may in fact be hemogenic, as recently proposed (Randolph et al., 2025 PMID: 40221343). We believe that these new data not only provide support for the specificity in the labelling of different hematopoietic waves in our model but also offer important biological insight. These points are now mentioned and discussed in the revised manuscript (Results section lines 235-242 and 255-261; discussion section lines 463-479). See also reply to Reviewer #4 point 1 regarding endothelial labeling.

We also thank the Reviewer for the second part of the request. As suggested, we have now performed direct transplantation experiments of *Cdh5-CreER^{T2}* E11.5 AGM and YS+VU traced at E8.5 (fetal restricted HSPCs) and E10.5 (HSCs). This new experiment enabled us to

determine the engraftment potential of fetal-restricted HSPCs before fetal liver seeding. We could analyze 4 mice transplanted with cells traced at E8.5 of which 3 were repopulated, and 2 transplanted with cells traced at E10.5 (both repopulated). Unfortunately, we could not obtain a higher numerosity is because in the first experiment which we performed many mice died due to technical issues related to the functioning of the irradiator in the animal house, which was delivering a radiation dose higher than expected. For this reason, we had to exclude this first experiment. We could not add more experiments due to time and mouse availability constraints. Nevertheless, we think that despite the small numbers, this experiment still allowed us to draw meaningful conclusions, especially because the result in this case was clearly a “black-and-white” one. Remarkably, all recipient mice repopulated with cells traced at E8.5, but none of those with cells traced at E10.5, displayed high levels of donor-derived cell labeling in myeloid and lymphoid cells of the PB (**Revised Figure 8B,C; Figure S10A,C**). Analysis of BM at 16 weeks post-transplantation confirmed PB data, with differentiated cells labeled only with E8.5 activation. Within the BM progenitor cell compartment, LSK, MPP and LK were only labeled in mice transplanted with AGMs traced at E8.5. Notably, immunophenotypic HSCs were not found labelled with either activation time point (**Revised Figure 8D; Figure S10B**). Within transplanted E11.5 AGM pools, a similar percentage of type I and type II pre-HSCs were labeled with both activations, while CD31+ Kit+ CD41low CD45+/- CD43+ CD201- non-preHSC progenitors were highly labeled with the E8.5 activation at and not with the activation at E10.5 (**Revised Figure S10C**), consistent with our new lineage tracing data (**Revised Figure 1**). Therefore, these new data suggest that: 1) adult repopulating potential in the E11.5 AGM is virtually confined to fetal-restricted HSPCs and 2) Adult-type HSCs labelled at E10.5 in *Cdh5-CreER^{T2}* mice are not competent for engraftment in the E11.5 AGM, and thus likely require further maturation in the fetal liver. This result provides support to the recently emerging concept that the fetal liver is an important site of pre-HSC maturation (Cain et al., 2025 PMID: 39256623; Ganuza et al. 2022 PMID 36202972).

10. Figure 7. The authors present engraftment data as “log10 frequency of donor labeled cells in PB (or BM) (normalized on % of donor labeled FL or BM LSK)”. It is hard for the reader to wrap their head around what that means. There must be a simpler way to acquire or present these data.

We thank the Reviewer for this comment. We acknowledge that the interpretation of these data was not immediate for the reader and needed improvement. Indeed, another reviewer also made a similar comment (Please also see response to Point 4 of reviewer #4).

The data previously shown in the submitted manuscript (and now shown in **Revised Figure 8E and 8F**, bottom panels) report a longitudinal analysis of the percentages of labeling in PB myeloid, B and T cells and in BM progenitor subsets of transplant recipients. These labelling frequencies are shown within donor cells (CD45.2+) and have been normalized to the initial level of labeling in donor tissues (FL or BM). The main reason for normalizing relates to the intrinsic variability of the labeling observed in our experimental setting and therefore we thought that this way of presenting data would facilitate the interpretation of our results instead of complicating it. We have now simplified the wording currently used in the figure

to label the Y axis of these graphs which now reads “Donor cell labelling in PB/BM normalized on input”.

Moreover, we now added absolute chimerism information in the revised figure (**Revised Figure 8E and 8F**, top panels) and simplified the experimental schematic of transplant experiments (**Revised Figure 8A**), now included in the same figure with the new E11.5 AGM transplant experiments (**Revised Figure 8B-D**). In addition, complete transplantation data, including the percentages of chimerism and labeling in each fraction, are also included in the manuscript as Supplementary Tables (**Tables S4, S5 and S6**).

We trust that these changes will help the interpretation of our results and thus improve conveying the main message of our work.

Minor comments:

11. Line 80. Primitive yolk sac progenitors are not unipotent. They also produce megakaryocytes (PMID: 17062726)

We have addressed this by revising the text accordingly; we have removed the “unipotent” wording. (line 83)

12. Line 146. The authors state they examined type I and type II pre-HSCs in both AGM and YS (including VU). The vitelline artery is in both the yolk sac and attached to the dorsal aorta, so which dissected tissue are they saying contains it?

The reason behind pooling YS and VU at E10.5 was largely derived from the whole-mount imaging data – specifically, the distribution of labelled hematopoietic clusters. At E10.5, both the YS arteries and the VU arteries in *Cdh5-CreER^{T2}::R26^{EYFP}* and *Cdh5-CreER^{T2}::R26^{ZsGreen}* embryos contain large Kit⁺ CD27⁺ clusters which are labelled with the E8.5 activation (and not labelled/much less so at E7.5), while the DA does not. The majority of cells within those clusters were not labeled with the *Csf1r-iCre* line, suggesting an origin independent from EMPs (**Revised Figure 4 and Figure S5A**). It is nearly impossible to cleanly dissect the YS plexus away from the YS arteries. Therefore, we pooled YS cells together with the vitelline and umbilical arteries because both of these tissues contain those large clusters that exhibit a similar immunophenotype and labeling dynamics.

At the same time, it is also very difficult, if not impossible, to cleanly dissect the VU away from the DA at E10.5, because at this stage the vitelline artery (VA) is connected to the DA through vascular ramifications that often contain Kit⁺ hematopoietic clusters (these can be seen in **Revised Figure 3C and 3E and Figure 4** of this Rebuttal; see also Yokomizo et al., Blood 2011, PMID: 21505195). Therefore, it is only with WM imaging that we can precisely assign an anatomical localization to the labelled cells. Based on the labeling pattern and the extra-aortic localization of these clusters, we considered them as part of the VA. In our interpretation, this

is the most probable reason why in our scRNA-seq analysis we observe labelled pre-HSPCs in the AGM: those are likely to be located in the ramifications of the VA that was impossible to separate from the DA (given the scarce labeling of aortic clusters with 4-OHT at E8.5). We made this point clearer in the revised manuscript.

To determine the labeling frequencies of pre-HSCs and hematopoietic progenitors in the VU separately from the YS and AGM, we dissected the VU separately from the YS and AGM and performed a flow cytometry analysis with our improved pre-HSC panel (See also response to Reviewer #1 point 1). The result of this analysis is shown in **Revised Figure S2F** and discussed in the Revised manuscript (lines 160-163). We observed no significant differences in the labelling frequency of hematopoietic progenitors (highly labelled at this stage) or Type I and II pre-HSCs in the YS as compared to the VU. Thus, we think this provides support for the pooling of these tissues in other experiments. (Please see also response to Reviewer #4, point #3)

13. Line 194. The authors write “These data show that the hematopoietic wave that emerges from HE between E8.5 and E9.5 is the main contributor of fetal lympho-myelopoiesis.” Wouldn’t it be more correct to say that they are measuring a wave between E8.5 and E10.5? To claim the wave occurs between E8.5 and E9.5 would imply that it is almost over between E9.5 and E10.5, but that experiment was not done.

The reason why we stated that we are measuring the contribution of a HE wave from E8.5 to E9.5 is that 4-OHT does not persist *in vivo* in the mouse serum for more than 12 hours, and most probably not more than 24 hours (Martinez-Corral et al., *Methods Mol Biol* 2018 PMID: 30242751). (Z)-4-OHT, which we used in this work, has a *in vivo* half-life of <3 hours, peaks in the serum almost immediately after administration and is largely cleared to undetectable levels within a 12-hour window as shown by mass spectrometry (Fowler et al., *Dev Cell* 2024 PMID: 38569552) therefore, we don’t think we can say that we are labelling HE between E9.5 and E10.5.

Moreover, we have amended this statement to “These data show that the hematopoietic wave that emerges from HE between E8.5 and E9.5 is a *major* contributor to fetal lympho-myelopoiesis” (lines 204-205).

Response to editors' comments

Dear Professor Azzoni, dear Emanuele,

Your manuscript "Fetal-restricted hematopoietic progenitors arise from hemogenic endothelium in vitelline and umbilical arteries" has now been seen by 3 referees, whose comments are appended below. You will see that they all agree your manuscript has improved, and two of them are satisfied with the current version.

However, our reviewer #5 raised a few more points that need to be addressed in the text before we can make a decision on publication. We therefore invite you to revise and resubmit your manuscript, taking into account the remaining requests.

Best regards,

Andrea

*Andrea Tavosanis, Ph.D.
Senior Editor
Nature Cardiovascular Research*

We thank the Editor for the opportunity to further revise our manuscript. We are pleased that Reviewers #1 and #4 are satisfied with the revised version and appreciate Reviewer #5's additional comments, which helped us further improve the clarity and interpretation of the work.

We have now addressed all remaining points raised by Reviewer #5. In particular, we have moderated our conclusions where appropriate, clarified terminology and figure references, strengthened statistical analyses, removed redundant data, and revised sections of the text to explicitly acknowledge alternative interpretations. We believe these changes improve both the rigor and transparency of the manuscript.

We hope that the revised version and detailed point-by-point response enclosed here satisfactorily address the remaining concerns.

Response to reviewers' comments

Reviewers comments:

Reviewer #1 (Remarks to the Author):

The authors present substantial new data and updated interpretations that greatly strengthen the conclusions of the manuscript and address my prior concerns. The manuscript should be of broad interest to the field of developmental hematopoiesis.

Reviewer #4 (Remarks to the Author):

Authors have successfully addressed all my comments and provided new experimental evidences supporting their findings. I believe this constitutes a beautiful piece of work that will be of high interest to the field.

Reviewer #5 (Remarks to the Author):

In this revised manuscript, Barone et al. address concerns from the previous reviews with additional experiments. Some of these experiments are strong additions to the manuscript, particularly the limiting dilution T and B progenitor analyses, refinement of pre-HSC markers, comparison of scRNA-seq data to published datasets, and expansion of the whole mount confocal data. Some of the data, particularly the segregation of EMPs, LMPs, and HSCs with their labeling approaches are convincing. The inducible Csf1r-Cre labeling studies are a very helpful addition in this regard. I'm less convinced by the data claiming that fetal-restricted HSCs can be separated from adult HSCs by these labeling methods.

There are many interesting points in this paper, and the comprehensive analyses of labeling strategies will be useful to other investigators in the field. I think the authors overstate some of their conclusions though and have attempted to point these out in the comments below. Other comments are suggestions to improve the clarity of the presentation.

We thank Reviewer #5 for their careful reading of the revised manuscript and for their constructive and thoughtful comments. We have addressed all points raised, either by revising the text to improve clarity and precision, moderating interpretations where appropriate, or by removing or restructuring data presentation as suggested. Detailed responses to each of the specific comments are provided below.

Note that line numbers mentioned in our response are related to the manuscript file without track changes.

Specific comments:

1. Lines 131-132. In the sentence “In contrast, tamoxifen metabolization into 4-OHT takes 6 to 12 hours in vivo and remains detectable in serum up to 48 hours post-administration”, is it the tamoxifen or 4-OHT that remains detectable in serum up to 48 hours post-administration”? I assume the authors are referring to tamoxifen, but it would be helpful to the reader if this is clarified.

We agree that this sentence required clarification. We have revised the text to explicitly state that it is tamoxifen to remain detectable in serum for up to 48 hours post-administration. The revised sentence now reads: “In contrast, tamoxifen requires hepatic metabolization into 4-hydroxytamoxifen (4-OHT), a process that takes approximately 6–12 hours in vivo; tamoxifen itself remains detectable in serum for up to 48 hours post-administration” (lines 116-119)

2. Lines 141-142. In the sentence “We analyzed the labeling of LMPs (CD31+ Kit+ CD45+) in the E10.5 AGM and in YS including VU (YS+VU)”, are the authors referring to the VU within the yolk sac, the VU between the YS and the embryo proper, or the VU within the embryo proper? Please specify in this sentence in the main text, right where this is mentioned.

We thank the reviewer for highlighting this ambiguity. We have now specified in the main text that YS+VU refers to the yolk sac together with the extra-embryonic portions of the vitelline and umbilical arteries connecting the yolk sac to the embryo proper. The text now reads: “We analyzed the labeling of LMPs (CD31+ Kit+ CD45+) in the E10.5 AGM and in YS including VU connecting the yolk sac to the embryo proper (YS+VU)” (lines 128-129).

3. Lines 176-178. The authors write “Taken together, these data suggest that HE lineage tracing in the Cdh5-CreERT2 model (4-OHT at E8.5) can capture a putative population of fetal-restricted HSPCs. In contrast, 4-OHT pulses at E7.5 or E10.5 respectively label either EMPs or adult-type HSCs.”

I'm still struggling with the assertion that the authors are selectively labeling fetal-restricted HSCs at E8.5 and adult HSCs at E10.5 (although I am convinced they are labeling LMPs at E8.5). I still believe it is possible that all the HSCs they are labeling are adult HSCs maturing from pre-HSCs differentiating from hemogenic endothelial cells over the course of several days. Let's imagine that 20% of ECs that give rise to adult HSCs are efficiently labeled at E8.5, and 80% of ECs that give rise to adult HSCs are efficiently labeled at E10.5. In addition, let's imagine that it takes 4-6 days for labeled ECs to undergo EHT and then mature into phenotypic adult HSCs in the fetal liver. That scenario could explain the labeling dynamic of phenotypic HSCs in the fetal liver in Figures 2B-D (more relative contribution by E8.5-labeled ECs to HSCs at E14.5 and less at E16.5 compared to E10.5-labeled ECs). In this scenario, the smaller proportion of HSCs derived from ECs labeled at E8.5 have finished maturing into phenotypic HSCs by E14.5, while the larger bolus of HSCs derived from ECs labeled at 10.5 haven't finished maturing, and their true proportional contribution to fetal liver HSCs is only apparent at E16.5 and in adult animals. In fact, the authors show much later in the manuscript, in Figure 8, that no HSCs have yet matured in the E11.5 AGM region from ECs labeled at E10.5, while pre-HSC to HSC maturation has occurred from ECs labeled at E8.5, consistent with this scenario.

The authors do show, however, that HSPCs labeled at E8.5 can yield multi-lineage engraftment in primary recipients but not in secondary recipients, while E10.5 labeled HSCs can do both (Figure 8F). Therefore, there is some difference in the capacities of the HSCs derived from E8.5 versus E10.5 labeled ECs. However, I don't know whether the inability to self-renew in the context of a secondary transplant qualifies an HSC as “fetal-restricted”.

We thank the reviewer for this important conceptual point and agree that differences in maturation timing could be theoretically considered as a possible explanation for the observed labeling dynamics. However, we believe that several of ours and others' observations argue against a scenario solely based on delayed maturation of adult-type HSCs.

More specifically:

- 1) First, in our experimental conditions, at E14.5 immunophenotypic HSCs originating from hemogenic endothelial cells (HEC) pulsed at E10.5 are labelled at a high frequency (>75%; Revised Figure 1F), and those originating from HEC pulsed at E8.5 are also labelled but with a significantly lower frequency and much greater variability, which would be consistent with an ongoing differentiation process of the latter. Thus, both phenotypic HSCs originating from HEC pulsed at E8.5 and E10.5 are seen in the E14.5 FL, which is the main reason why we chose this time point for the transplants included in Figure 8. Our transplantation data clearly indicates that 4 days (E10.5>E14.5) are sufficient for HEC to mature into functional HSCs: the progeny of HSCs labelled at E10.5 can be traced into primary and secondary transplant recipients. Thus, the higher labelling frequency of phenotypic HSCs observed in the E14.5 FL with the E10.5 activation as compared to the E8.5 activation is not easily reconciled with the scenario proposed by the reviewer.
- 2) Second, our “fetal-restricted” definition was given not only based on the HSPC labeling dynamics now shown in Figure 1, but also on the lympho-myeloid contribution dynamics reported in Figure 2. E8.5-derived contributions consistently decline after birth, whereas E10.5-labeled cells progressively increase in postnatal hematopoiesis (Please see new Figure 2D and response to point 6 below). We think that this scenario is more consistent with the presence of temporally distinct hematopoietic stem/progenitor populations with different potential and functional outputs, rather than a single homogeneous HSC population labeled at different maturation stages.
- 3) Third, as mentioned by the reviewer, our FL transplantation data provides important evidence of functional differences in the potential of the two subsets of HSPCs originating from HEC labelled at E8.5 and E10.5. Although we agree that the result of our AGM transplantations (i.e. Presence of donor derived labeling in recipients reconstituted with E11.5 AGMs labelled at E8.5 but absence of donor derived labeling in recipients reconstituted with E11.5 AGMs labelled at E10.5) can be explained by the fact that HEC labelled at E10.5 have not yet matured into functional HSCs at E11.5 in the AGM, as we clearly mention in our manuscript, by E14.5 both HSPCs labelled at E8.5 and E10.5 have had enough time for maturation (see point 1 above). In our view, the greater reconstitution and superior self-renewal potential of E14.5 HSPCs labelled at E10.5 are consistent with the scenario that adult-fated HSCs are only captured by the E10.5 activation, which is also in agreement with our lineage tracing data. Moreover, these findings raise the possibility that HSCs labelled at E10.5 require transit through the FL niche to achieve full functional maturation, in contrast to HSPCs labelled at E8.5, which (at least in part) already display multilineage engraftment capacity when transplanted from the AGM at E11.5. This suggests that the latter population may be intrinsically primed toward differentiation at earlier developmental stages.
- 4) Fourth, proof of existence of fetal HSCs and eMPPs was also provided by other investigators (Beaudin et al 2016, PMID: 27666010; Patel et al 2022 PMID: 35705805; Kobayashi et al 2023 PMID: 36906851; Shang et al 2025 PMID: 41284889, etc.). Recent barcoding studies showed that eMPPs are generated in the embryo prior to E14.5 and independently from HSCs (Shang et al 2025). Our data is also consistent with previous reports suggesting functional heterogeneity within the E11.5 pre-HSC pool (Hadland et al. 2017 PMID: 28479303). Perhaps the strongest evidence arguing against a “single wave” of adult HSCs is the study by Yokomizo et al. showing that HSCs and other progenitors during fetal hematopoiesis originate independently from each other, and that adult-fated HSCs contribute minimally to other hematopoietic progenitors during gestation (Yokomizo et al Nature 2022, PMID: 36104564), a finding that was also confirmed by others and using different models. Thus, our data fits with the bulk of previous studies providing direct support for the existence of fetal HSCs and a non-HSC source of multilineage hematopoiesis in the embryo, rather than one single type of HSCs that emerges over the course of several days during mouse development.
- 5) Fifth, when referring to “fetal-restricted HSPCs” in our data we are not only referring to fetal HSCs. Instead, what our data suggests is that the output of the HEC labelled at E8.5 is heterogeneous and includes fetal HSCs, eMPPs and possibly other progenitors (as clearly mentioned in our manuscript, lines 198-204). Indeed, figure 1 shows that labeling in the CD201- progenitor fraction at E11.5 is almost exclusive to the E8.5 activation. As correctly mentioned by the reviewer, LMPs are also labelled with the activation at E8.5 in the Cdh5-CreERT2 model. Mouse LMPs express Csf1r (refs. 20 and 25 and Camiolo et al., Leukemia 2026 PMID: 41545699). However, we do not believe that LMPs alone can explain the substantial lympho-myeloid contribution observed later in development. This is supported by our Csf1r^{MerCreMer} lineage tracing data showing limited contribution to B and myeloid cells in the FL (and no contribution to LSK) and agrees with the study by Boiers et al. (PMID: 24054998) which showed that LMPs yield little contribution to fetal myelopoiesis. Indeed, the percentages found by us and the Boiers study which employed a different tracing strategy (Rag1-Cre) are remarkably similar, consistent with the lymphoid contribution which we observed with the Csf1r^{MerCreMer} model may indeed largely originate from LMPs.
- 6) Sixth, our imaging data clearly demonstrate that HEC at E8.5 largely label HSPCs emerging in the vitelline and umbilical arteries. We acknowledge we have not directly investigated the location of clusters labelled at E10.5 by whole mount imaging. However, based on our HSC labeling, the timing of emergence of aortic clusters and the cartography imaging data from Yokomizo and Dzierzak, 2010 (PMID: 20876651) which showed that at E11.5 the DA still contains a significant number of hematopoietic clusters whereas those in VA and UA substantially decrease, we can infer a predominant localization in the dorsal aorta for the HEC labelled at E10.5. While spatial origin alone does not constitute definitive proof of functional divergence, multiple studies indicate that distinct anatomical sites provide different niche environments that can influence hematopoietic output and fate decisions. A clear example is provided by erythro-myeloid progenitors (EMPs), which originate primarily from hemogenic endothelium within the yolk sac vascular plexus and give rise to progeny with functional properties that are distinct from those of aortic-derived HSCs. These differences have been attributed, at least

in part, to site-specific endothelial programs, local signaling environments, and differences in timing of emergence. Similarly, hemogenic endothelium located in the vitelline/umbilical arteries versus the dorsal aorta is exposed to distinct hemodynamic forces, stromal interactions, and molecular cues, all of which have been shown to influence endothelial-to-hematopoietic transition and subsequent lineage potential. Therefore, although we do not claim that anatomical origin alone defines HSPC identity, we propose that spatially and temporally distinct hemogenic niches may contribute to the functional heterogeneity observed between HSPCs labeled at E8.5 and E10.5, in combination with differences in timing of emergence and maturation.

- 7) Finally, we would like to suggest an alternative interpretation of our data. 4-OHT labeling at E10.5 recombined a large fraction of E14.5 FL phenotypic HSCs (~75%), which -similar to the those labelled at E8.5- also exhibited a progressive decrease to ~54% at E16.5 and ~40% in the adult bone marrow (Figure 1E–G and shown below). Intriguingly, the magnitude of this decrease closely matches the proportion of lympho-myeloid cells (excluding macrophages) labeled in the E16.5 fetal liver following E10.5 activation (~30%; Figure 2A). This quantitative correspondence raises the possibility that 4-OHT administration at E10.5 may label not only adult-fated HSCs (which we believe to be exclusively labeled with this activation mode), but also a subset of fetal-restricted HSPCs, whose progeny contribute substantially to fetal lympho-myeloid output but do not persist long term. This would also be supported by the fact that the sum of the averages of phenotypic E14.5 HSCs labelled at E8.5 and E10.5 exceeds 100%, indicating possible overlap. In this scenario, the decline in labeled HSCs between fetal and adult stages would reflect the selective loss of fetal-restricted HSPCs from the phenotypic HSC compartment, whereas adult-fated HSCs would progressively dominate postnatal hematopoiesis. This interpretation supports a model in which overlapping hemogenic endothelial populations give rise to functionally heterogeneous HSPCs with distinct persistence and output. These observations are also consistent with recent work by Ganuza et al., showing that after a transient expansion phase between E12.5 and E14.5, most fetal liver HSCs preferentially undergo differentiation at the expense of self-renewal. Together, our data support the view that the majority of fetal lympho-myeloid production arises from HSPCs with limited long-term persistence, whereas a smaller subset of adult-fated HSCs progressively assumes dominance after birth. Further experiments will be needed to definitely confirm this hypothesis.

To acknowledge the Reviewer's point, we have included the following sentence in the discussion: (lines 502-506): "Although we cannot formally exclude that part of the observed dynamics reflects differences in the timing of pre-HSC emergence and maturation, the combined evidence from lineage output, transplantation, postnatal contribution, and spatial origin argues that our labeling strategy captures functionally heterogeneous HSPCs rather than a single homogeneous HSC pool." If the Reviewer and/or the Editor deems it necessary, we can also include our alternative interpretation described at the point 7 of our response above. It was previously included in the originally submitted manuscript, but we removed it from the revised version due to space constraints.

4. Line 194. The authors write "embryonic gamma/delta T cells were almost exclusively labeled by 4-OHT at E8.5 (Figure 2F). There appears to be a 4-fold difference in labeling efficiency at E8.5 versus E10.5. I think "almost exclusively" is a bit too strong a statement, and stating the fold difference would be more accurate.

We agree and have revised the sentence to quantitatively describe the approximately four-fold higher labeling efficiency at E8.5 compared to E10.5, removing the phrase "almost exclusively". Revised text now reads: "In contrast to E10.5 activation, double-positive (CD4+ CD8+ DP) T cells displayed the highest labeling frequency, and embryonic $\gamma\delta$ T cells were preferentially labeled by 4-OHT administration at E8.5, showing an approximately four-fold higher labeling efficiency compared to E10.5 (Figure 2B)" (lines 180-183)

5. Lines 198-199. The authors write "Post-natal analysis confirmed absence of peripheral blood (PB) labeling (<5%) in all lineages with activation at E7.5, both at 21 days (Figure S3C) and 2 months (Figure 2G; Figure S3D)." According to the figure legend, Figure S3D shows data from E8.5, not E7.5 labeling.

We thank the reviewer for noticing this inaccuracy. The figure reference has now been corrected.

6. Lines 200-201. The authors write “4-OHT at E10.5 resulted in high levels of recombination in all lineages gradually increasing between 21 days and 2 months (Figure 2G; Figure S3C)”. Are those differences between 21 days and 2 months statistically significant? The averages don't look that different to me. Same question for the E8.5 labeling data – although the variation between mice indeed looks higher at 21 days than in adult mice, the averages don't look all that different.

We are grateful to the reviewer for pointing this out. We agree that it is important to include statistics for the comparison of B-T-myeloid labelling at different time points of analysis. This comment prompted us to include a new longitudinal analysis of these data, now presented in Figure 2D. To make space for this, Figure 1 and Figure 2 have now been restructured: Figure 1 now contains pre-HSC and HSPC labelling data, whereas Figure 2 contains B-T-myeloid labelling data only. We think that this change makes the flow much clearer and is also helped by a better correspondence between paragraphs and figures.

As the reviewer was rightfully pointing out, the differences between 21 days and 2 months are not statistically significant for any of the cell subsets (Figure 2D) and therefore we removed the claim relative to a gradual increase. However, as now shown in Figure 2D, the interesting change happens between E16.5 and postnatal (21 days), in which the E8.5 contribution significantly decreases for all the cell subsets and the E10.5 exhibits a concomitant increase, statistically significant for myeloid, T cells and CD45+. For B cells, this increase does not reach statistical significance, however the trend is evident and the difference between E16.5 and adult is indeed highly significant ($p < 0.008$, not shown in figure). This was a central point of the story that was not clearly highlighted in the previous version of the manuscript. Thus, we have now added a few sentences in the text:

Lines 185-197: Post-natal analysis confirmed absence of peripheral blood (PB) labeling (<5%) in all lineages with activation at E7.5, both at 21 days (Figure S3C) and 2 months (Figure 2C,D), while E10.5 activation resulted in high levels of recombination in all lineages (Figure 2C,D; Figure S3C). Conversely, activation at E8.5 showed highly variable labeling at 21 days (average 30-35%) (Figure S3C), which exhibited a decreasing trend at 2 months (20-25%; Figure 2C,D; Figure S3D). Notably, comparison of fetal and postnatal stages revealed divergent dynamics for the two labeling windows. Labeling obtained with 4-OHT activation at E8.5, which was highest at E16.5, declined markedly after birth, indicating that progeny of this HE wave progressively lose representation in postnatal hematopoiesis (Figure 2D). In contrast, labeling following 4-OHT activation at E10.5 increased from E16.5 to postnatal stages (Figure 2D), consistent with the gradual takeover of hematopoiesis by adult-type definitive HSCs. Together, these reciprocal trends indicate a developmental handover between a predominantly fetal hematopoietic program emerging from E8.5–E9.5 HE and a later-arising program specified from E10.5 HE that sustains postnatal blood production.

7. Lines 294-295. The authors write “These data show that *Csf1r*+ progenitors, including EMPs (defined as *CD41+* *Kit+* *CD16/32+*) contribute in vivo to fetal lymphoid cells, confirming a previous report”. I agree the authors demonstrated that *Csf1r*+ progenitors contribute in vivo to fetal lymphoid cells, but what is their evidence that EMPs contribute to fetal lymphoid cells? They show that *Csf1r-iCre* also marks non-EMPs in the yolk sac and reference a previous report that yolk sac LMPs express *Csf1r*. It would be more accurate to delete the phrase “including EMPs (defined as *CD41+* *Kit+* *CD16/32+*)”.

We agree with this. We have removed the phrase “including EMPs (defined as *CD41+* *Kit+* *CD16/32+*)”, and now more accurately state that *Csf1r*+ progenitors contribute to fetal lymphoid populations, without attributing this activity specifically to EMPs (line 344).

8. I'm not sure what Figures 5D,E add to the story. Figure 5F,G contain much better data and are easier to interpret.

We agree with the reviewer that the bulk OP9 co-culture data previously shown in Figures 5D and 5E was redundant with the new limiting dilution assays. Therefore, we have followed the reviewer's suggestion and we have removed this data as well as all corresponding references in the text and figure legends. The conclusions in the paragraph titled “B- and T-lymphoid potential appears in intra- and extra-embryonic HE between E8.5 and E9.5”, based on Figure 5 data, now rely on CFU-C and LDA, which were both generated through culture of sorted cell fractions, and we agree provide clearer and more quantitative evidence.

9. Lines 327-328. The authors write “The different frequencies of B and T lymphocyte progenitors suggests that developmental uncoupling may already take place in the tissues of emergence of these precursors.” Are these differences in B and T progenitor frequencies significant? Couldn't differences in the efficiency of the B and T cell assays be responsible for the different frequencies of T and B progenitors? This is strong statement without strong evidence to back it.

We thank the reviewer for raising this point. We acknowledge the fact that we made this claim without specifying whether the differences in B and T progenitor frequencies were significant. Indeed, these differences are both significant according to the ELDA tool. We now added p-values to the text, lines 307-309: T progenitors were significantly more

frequent than B progenitors in both AGM (1:81 versus 1:204, $p=0.016$) and YS+VU (1:123 versus 1:314, $p=0.012$) (Figure 5D,E).

We would like to add that we think that it is unlikely that differences in the efficiency of the B and T cell assays alone could explain the higher frequency of T cell progenitors observed in both AGM and YS+VU. In this respect, the YS+VU zsG- fraction serves as a useful control, as in this subset the observed frequency is higher for B cell progenitors than T cell progenitors (1:968 versus 1:2190). If the observed differences depended exclusively on a higher efficiency of the T cell assay, we would have observed lower frequencies for B cell progenitors in all fractions, which was not the case.

We have nevertheless changed and softened the statement to make it less strong, which now reads: The different frequencies of B and T lymphocyte progenitors observed in AGM and YS+VU are consistent with a model in which developmental uncoupling may already take place in the tissues of emergence of these precursors (lines 312-314).

10. Lines 395-397. The authors write "Remarkably, all recipient mice repopulated with cells traced at E8.5, but none of those with cells traced at E10.5, displayed high levels of donor-derived labeling in myeloid and lymphoid cells of the PB (Figure 8B,C; Figure S10A,C)." But the bar graphs in 8B, C show 1 out of 2 mice transplanted with cells traced at E10.5 were multi-lineage engrafted in PB, and both mice had multi-lineage bone marrow engraftment, so the statement is confusing. The authors should make that statement when referring to Figure 8C and D, which presents the labeling frequency.

We thank the reviewer for bringing up this lack of clarity in the references to figure panels. This statement refers exclusively to donor-derived labeling (tdTomato+/zsGreen+) shown in Figure 8C and D as correctly pointed out by the reviewer, not to overall donor chimerism (CD45.2+), which is instead shown in Figure 8B.

Figure 8B shows that both mice transplanted with AGM cells traced at E10.5 (2 e.e./recipient) were multi-lineage engrafted. In both these mice, the contribution of donor derived labeled cells in PB and BM was minimal, in contrast to recipients transplanted with E8.5-traced cells (Figure 8C and 8D). To avoid confusion, we have amended the figure references in the text. We now refer to panel 8B in the previous sentence: "we directly transplanted E11.5 AGM and YS+VU from Cdh5-CreERT2::R26tdTomato/zsGreen pulsed with 4-OHT at E8.5 (fetal-restricted HSPCs) or E10.5 (adult-type HSCs) into lethally irradiated syngeneic CD45.1 recipients (Figure 8A,B)" while we only refer to figure 8C and to the supplementary figure showing the flow cytometric analysis in the sentence mentioned by the reviewer, at lines 381-383: "Remarkably, all recipient mice repopulated with cells traced at E8.5, but none of those with cells traced at E10.5, displayed high levels of donor-derived labeling in myeloid and lymphoid cells of the PB (Figure 8C; Figure S10A)"

11. Lines 818-819. The authors write "4-OHT E8.5 (n=4) and 4-OHT E10.5 (n=2) AGM transplanted recipient mice were analyzed in 4 independent experiments.". Were there 6 mice in each experiment, or a total of 6 mice were transplanted, and split over 4 independent experiments? If the latter, are 2 transplanted E10.5-labeled AGMs enough samples to reach a remarkable conclusion? It is a startling result, based on not very robust data if only two E10.5 AGM regions were analyzed.

The reviewer refers to the legend of Figure 8C. The latter case is the correct one (i.e. a total of 6 mice were transplanted and split over 4 independent experiments). However, as mentioned in the Methods section, each mouse was transplanted using pooled AGM regions from two donor embryos. This brings the number of transplanted AGMs to a total of four for the E10.5 activation. We agree that the wording in the figure legend required clarification. Thus, we have revised the relevant legends as follows:

(B) Percentage (%) of chimerism (% of donor CD45.2+) in adult lethally irradiated mice transplanted with E11.5 Cdh5-CreERT2::R26tdTomato/zsGreen AGM (2 e.e./recipient), in PB (left) and BM (right) 16 weeks after transplantation. Each bar indicates a single recipient mouse. The dashed line indicates 5%, the threshold we considered for repopulation.

(C) Longitudinal analysis showing the frequency of labeled (zsGreen+ or tdTomato+) donor myeloid (CD45.2+ CD11b+), B cells (CD45.2+ B220+), T cells (CD45.2+ CD3e+) in PB of mice transplanted with E11.5 AGM. Gating strategy shown in S10A. Four (4-OHT E8.5) and two (4-OHT E10.5) mice were analyzed in four independent transplant experiments. Data are mean \pm SD. A summary of E11.5 transplantation data is provided in Table S4.

We acknowledge the limited sample size for the E10.5 condition and have therefore moderated the strength of the conclusions. Lines 391-393: "These data suggest that, in the E11.5 AGM, adult repopulation potential may be confined to fetal-restricted HSPCs and is likely to reside in the CD201- subfraction, while adult-type HSCs labeled at E10.5 are not yet competent for engraftment and most probably require further maturation". We believe to have avoided categorical language and to have framed this finding as supportive rather than conclusive. Nevertheless, given the consistency of our results despite the limited numbers, we also believe that this finding is suggestive of an important biological

difference between HSPC labelled at E8.5 – which are endowed with repopulation potential already in their tissue of emergence- and those labelled at E10.5, which are not. We don't think that this is simply a difference that can be attributed to the timing of maturation, but rather we are looking at different progenitor subsets with distinct properties. We agree that further experiments will be needed to confirm this result and we have mentioned this in the discussion (lines 496-497, "Further experiments will be needed to validate this finding").

We would like to point out that we have already explained the reasons for the limited sample size of this experiment in the previous point-by-point response to the reviewers (Reviewer 5, point 9). We report here the relevant text:

As suggested, we have now performed direct transplantation experiments of Cdh5-CreERT2 E11.5 AGM and YS+VU traced at E8.5 (fetal restricted HSPCs) and E10.5 (HSCs). This new experiment enabled us to determine the engraftment potential of fetal-restricted HSPCs before fetal liver seeding. We could analyze 4 mice transplanted with cells traced at E8.5 of which 3 were repopulated, and 2 transplanted with cells traced at E10.5 (both repopulated). Unfortunately, we could not obtain a higher numerosity is because in the first experiment which we performed many mice died due to technical issues related to the functioning of the irradiator in the animal house, which was delivering a radiation dose higher than expected. For this reason, we had to exclude this first experiment. We could not add more experiments due to time and mouse availability constraints. Nevertheless, we think that despite the small numbers, this experiment still allowed us to draw meaningful conclusions, especially because the result in this case was highly consistent across recipients. Remarkably, all recipient mice repopulated with cells traced at E8.5, but none of those with cells traced at E10.5, displayed high levels of donor-derived cell labeling in myeloid and lymphoid cells of the PB (Revised Figure 8C; Figure S10A). Analysis of BM at 16 weeks post-transplantation confirmed PB data, with differentiated cells labeled only with E8.5 activation. Within the BM progenitor cell compartment, LSK, MPP and LK were only labeled in mice transplanted with AGMs traced at E8.5. Notably, immunophenotypic HSCs were not found labelled with either activation time point (Revised Figure 8D; Figure S10B). Within transplanted E11.5 AGM pools, a similar percentage of type I and type II pre-HSCs were labeled with both activations, while CD31+ Kit+ CD41low CD45+/- CD43+ CD201- non-preHSC progenitors were highly labeled with the E8.5 activation at and not with the activation at E10.5 (Revised Figure S10C), consistent with our new lineage tracing data (Revised Figure 1). Therefore, these new data suggest that: 1) adult repopulating potential in the E11.5 AGM is virtually confined to fetal-restricted HSPCs and 2) Adult-type HSCs labelled at E10.5 in Cdh5-CreERT2 mice are not competent for engraftment in the E11.5 AGM, and thus likely require further maturation in the fetal liver. This result provides support to the recently emerging concept that the fetal liver is an important site of pre-HSC maturation (Cain et al., 2025 PMID: 39256623; Ganuza et al. 2022 PMID 36202972).

12. Lines 395-397. The authors write "Remarkably, all recipient mice repopulated with cells traced at E8.5, but none of those with cells traced at E10.5, displayed high levels of donor-derived labeling in myeloid and lymphoid cells of the PB (Figure 8B,C; Figure S10A,C). Figure S10C shows pre-HSC labeling, not PB analysis of myeloid and lymphoid cells.

We thank the reviewer for noting this error. The reference to Figure S10C has been removed.

13. Figure 8E, F. I still find the engraftment data presented as Donor cell labeling in PB (or BM) normalized to input (log10) is hard to interpret.

We appreciate the reviewer's concern. In these experiments, overall donor engraftment is shown directly as chimerism (% of donor CD45.2+ on total CD45+, shown in the top graphs in Figure 8E), while normalization to input labelling is intended to specifically assess the relative contribution of traced (labelled) versus untraced cells, accounting for differences in labelling efficiency at the time of transplantation.

Because in our opinion no clearer alternative representation would faithfully convey this information, we have retained this normalization strategy but have revised the text to more explicitly explain how these values should be interpreted and how they relate to overall engraftment. The revised text now reads: "Chimerism levels and labeling frequencies within donor-derived fractions were followed over time in PB. To account for variability in labeling efficiency, donor-derived labeling was normalized to the percentage of labeled LSK in each donor sample prior to transplantation (Figure S10D)" (lines 396-399)

The figure legends for panels 8E and F have also been amended for clarity.

Response to editors' comments

Dear Dr. Azzoni, dear Emanuele,

Thank you for submitting your revised manuscript "Fetal-restricted hematopoietic progenitors arise from hemogenic endothelium in vitelline and umbilical arteries" (NCVR-2024-07-0609B). It has now been seen by the original referees and their comments are below. The reviewers find that the paper has improved in revision, and therefore we'll be happy in principle to publish it in Nature Cardiovascular Research, pending minor revisions to comply with our editorial and formatting guidelines.

We have also checked with our referee, and they are happy with your suggestion to re-introduce the data in the manuscript. They liked your proposal to reduce the emphasis on these findings and reserve them for the extended data or supplementary materials. Please feel free to implement this change when uploading the final version of the paper.

Sincerely,

Andrea

*Andrea Tavosanis, Ph.D.
Senior Editor
Nature Cardiovascular Research*

We thank the Editor for the positive response, and we are delighted that our paper has been accepted in principle for publication in *Nature Cardiovascular Research*. We are pleased that Reviewer #5 is satisfied with the latest revised version of our manuscript.

Please note that in the final version of the manuscript enclosed here we have reintroduced the dataset that we had removed in the previous version; an explanation is provided below.

Response to reviewers' comments

Reviewers comments:

Reviewer #5 (Remarks to the Author):

The authors have done an excellent job in addressing my concerns. I congratulate them on this exciting work.

We would like to thank the Reviewer for the positive response and for the appreciation of our work.

Following up on one of the Reviewer's suggestions (Point 8, reviewer 5), in the second round of revision we had removed the OP9 bulk co-cultures previously included in Figure 5D and 5E. In those OP9 bulk co cultures, we used both E7.5 and E8.5 4-OHT activations, and we observed that both AGM and YS+VU cells labelled at E8.5 generated B and T lymphocytes at greater frequency than those labelled at E7.5.

In the new LDA experiments included in Figure 5, we confirmed that B and T potential is observed in the labelled fraction at much greater frequency than the unlabelled fraction. However, although the LDA experiments had the significant advantage over the bulk co-cultures that we could precisely quantify the frequency of B and T cell progenitors in the labelled fractions, in this experimental setting we used only the E8.5 activation.

These two datasets (bulk OP9 co-cultures and LDA OP9 co-cultures) are not redundant with each other, because the unlabelled cell subset with activation at E8.5 does not equal the labelled subset at E7.5. More specifically: a) EMPs are efficiently labelled with both E7.5 and E8.5 4-OHT activations and b) LMP labelling, although higher with the E8.5 activation, is also seen with the E7.5 activation. The frequency of B/T lymphoid cells generated in culture is very low with activation at E7.5, implying that lymphoid potential is scarce in the cell fraction labelled with this activation mode, which is consistent with our in vivo lineage tracing results. However, due to the overlap in EMP labelling, with the E8.5 activation only we cannot attribute the ex vivo B/T potential to EMPs or non-EMPs. It is conceivable that the lymphoid potential is

attributable to non-EMPs, but we cannot conclude this based only on the LDA dataset included here. Although not extensively commented on in the manuscript, we feel that this is an important point.

Thus, we have decided to re-include the OP9 bulk co-culture data in the final version of the manuscript. Nevertheless, following up from the previous comment of the Reviewer, we have now reduced the emphasis on this dataset and removed it from the main figure to include it in Extended Data Figure 6A-D. We feel that the inclusion of this dataset in the manuscript directly supports the conclusion that titles the corresponding paragraph "B- and T-lymphoid potential appears in intra- and extra-embryonic HE between E8.5 and E9.5".